# On the Theory of Transfer Learning: The Importance of Task Diversity

**Nilesh Tripuraneni**
University of California, Berkeley
nilesh_tripuraneni@berkeley.edu

**Michael I. Jordan**
University of California, Berkeley
jordan@cs.berkeley.edu

**Chi Jin**
Princeton University
chij@princeton.edu

## Abstract

We provide new statistical guarantees for transfer learning via representation learning–when transfer is achieved by learning a feature representation shared across different tasks. This enables learning on new tasks using far less data than is required to learn them in isolation. Formally, we consider $t + 1$ tasks parameterized by functions of the form $f_j \circ h$ in a general function class $\mathcal{F} \circ \mathcal{H}$, where each $f_j$ is a task-specific function in $\mathcal{F}$ and $h$ is the shared representation in $\mathcal{H}$. Letting $C(\cdot)$ denote the complexity measure of the function class, we show that for diverse training tasks (1) the sample complexity needed to learn the shared representation across the first $t$ training tasks scales as $C(\mathcal{H}) + tC(\mathcal{F})$, despite no explicit access to a signal from the feature representation and (2) with an accurate estimate of the representation, the sample complexity needed to learn a new task scales only with $C(\mathcal{F})$. Our results depend upon a new general notion of task diversity–applicable to models with general tasks, features, and losses–as well as a novel chain rule for Gaussian complexities. Finally, we exhibit the utility of our general framework in several models of importance in the literature.

## 1  Introduction

Transfer learning is quickly becoming an essential tool to address learning problems in settings with *small* data. One of the most promising methods for multitask and transfer learning is founded on the belief that multiple, differing tasks are distinguished by a small number of task-specific parameters, but often share a common low-dimensional representation. Undoubtedly, one of the most striking successes of this idea has been to only re-train the final layers of a neural network on new task data, after initializing its earlier layers with hierarchical representations/features from ImageNet (i.e., ImageNet pretraining) [Donahue et al., 2014, Gulshan et al., 2016]. However, the practical purview of transfer learning has extended far beyond the scope of computer vision and classical ML application domains such as deep reinforcement learning [Baevski et al., 2019], to problems such as protein engineering and design [Elnaggar et al., 2020].

In this paper, we formally study the composite learning model in which there are $t + 1$ tasks whose responses are generated noisily from the function $f_j^\star \circ \mathbf{h}^\star$, where $f_j^\star$ are task-specific parameters in a function class $\mathcal{F}$ and $\mathbf{h}^\star$ an underlying shared representation in a function class $\mathcal{H}$. A large empirical literature has documented the performance gains that can be obtained by transferring a jointly learned representation $\mathbf{h}$ to new tasks in this model [Yosinski et al., 2014, Raghu et al., 2019, Lee et al., 2019].

There is also a theoretical literature that dates back at least as far as [Baxter, 2000]. However, this progress belies a lack of understanding of the basic statistical principles underlying transfer learning[1]:

**How many samples do we need to learn a feature representation shared across tasks and use it to improve prediction on a new task?**

In this paper we study a simple two-stage empirical risk minimization procedure to learn a new, $j = 0$th task which shares a common representation with $t$ different training tasks. This procedure first learns a representation $\hat{\mathbf{h}} \approx \mathbf{h}^\star$ given $n$ samples from each of $t$ different training tasks, and then uses $\hat{\mathbf{h}}$ alongside $m$ fresh samples from this new task to learn $\hat{f}_0 \circ \hat{\mathbf{h}} \approx f_0^\star \circ \mathbf{h}^\star$. Informally, our main result provides an answer to our sampling-complexity question by showing that the excess risk of prediction of this two-stage procedure scales (on the new task) as[2],

$$\tilde{O}\left( \frac{1}{\nu}\left( \sqrt{\frac{C(\mathcal{H}) + tC(\mathcal{F})}{nt}}\right) + \sqrt{\frac{C(\mathcal{F})}{m}}\right),$$

where $C(\mathcal{H})$ captures the complexity of the shared representation, $C(\mathcal{F})$ captures the complexity of the task-specific maps, and $\nu$ encodes a problem-agnostic notion of task diversity. The latter is a key contribution of the current paper. It represents the extent to which the $t$ training tasks $f_j^\star$ cover the space of the features $\mathbf{h}^\star$. In the limit that $n, t \to \infty$ (i.e., training task data is abundant), to achieve a fixed level of constant prediction error on the new task only requires the number of fresh samples to be $m \approx C(\mathcal{F})$. Learning the task in isolation suffers the burden of learning both $\mathcal{F}$ and $\mathcal{H}$—requiring $m \approx C(\mathcal{F} \circ \mathcal{H})$—which can be significantly greater than the transfer learning sample complexity.

Maurer et al. [2016] present a general, uniform-convergence based framework for obtaining generalization bounds for transfer learning that scale as $O(1/\sqrt{t}) + O(1/\sqrt{m})$ (for clarity we have suppressed complexity factors in the numerator). Perhaps surprisingly, the leading term capturing the complexity of learning $\mathbf{h}^\star$ decays only in $t$ but not in $n$. This suggests that increasing the number of samples per training task cannot improve generalization on new tasks. Given that most transfer learning applications in the literature collect information from only a few training tasks (i.e., $n \gg t$), this result does not provide a fully satisfactory explanation for the practical efficacy of transfer learning methods.

Our principal contributions in this paper are as follows:

- We introduce a problem-agnostic definition of task diversity which can be integrated into a uniform convergence framework to provide generalization bounds for transfer learning problems with general losses, tasks, and features. Our framework puts this notion of diversity together with a common-design assumption across tasks to provide guarantees of a fast convergence rate, decaying with *all of the samples* for the transfer learning problem.

- We provide general-purpose bounds which decouple the complexity of learning the task-specific structure from the complexity of learning the shared feature representation. Our results repose on a novel user-friendly chain rule for Gaussian processes which may be of independent interest (see Theorem 7). Crucially, this chain rule implies a form of modularity that allows us to exploit a plethora of existing results from the statistics and machine learning literatures to individually bound the sample complexity of learning task and feature functions.

- We highlight the utility of our framework for obtaining end-to-end transfer learning guarantees for several different multi-task learning models including (1) logistic regression, (2) deep neural network regression, and (3) robust regression for single-index models.

## 1.1 Related Work

The utility of multitask learning methods was observed at least as far back as Caruana [1997]. In recent years, representation learning, transfer learning, and meta-learning have been the subject of extensive empirical investigation in the machine learning literature (see [Bengio et al., 2013],

[Hospedales et al., 2020] for surveys in these directions). However, theoretical work on transfer learning—particularly via representation learning—has been much more limited.

A line of work closely related to transfer learning is gradient-based meta-learning (MAML) [Finn et al., 2017]. These methods have been analyzed using techniques from online convex optimization, using a (potentially data-dependent) notion of task similarity which assumes that tasks are close to a global task parameter [Finn et al., 2019, Khodak et al., 2019a, Denevi et al., 2019a,b, Khodak et al., 2019b]. Additionally, Ben-David and Borbely [2008] define a different notion of distributional task similarity they use to show generalization bounds. However, these works do not study the question of transferring a common representation in the generic composite learning model that is our focus.

In settings restricted to linear task mappings and linear features, Lounici et al. [2011], Pontil and Maurer [2013], and Cavallanti et al. [2010] have provided sample complexity bounds for the problem of transfer learning via representation learning. Lounici et al. [2011] and Obozinski et al. [2011] also address sparsity-related issues that can arise in linear feature learning.

To our knowledge, Baxter [2000] is the first theoretical work to provide generalization bounds for transfer learning via representation learning in a general setting. The formulation of Baxter [2000] assumes a generative model over tasks which share common features; in our setting, this task generative model is replaced by the assumption that training tasks are diverse (as in Definition 3) and that there is a common covariate distribution across different tasks. In follow-up work, Maurer et al. [2016] propose a general, uniform-convergence-based framework for obtaining transfer learning guarantees which scale as $O(1/\sqrt{t}) + O(1/\sqrt{m})$ [Maurer et al., 2016, Theorem 5]. The second term represents the sample complexity of learning in a lower-dimensional space given the common representation. The first term is the bias contribution from transferring the representation—learned from an aggregate of $nt$ samples across different training tasks—to a new task. Note this leading term decays only in $t$ and not in $n$: implying that increasing the number of samples per training task cannot improve generalization on new tasks. Unfortunately, under the framework studied in that paper, this $\Omega(1/\sqrt{t})$ cannot be improved Maurer et al. [2016].

Recent work in Tripuraneni et al. [2020] and Du et al. [2020] has shown that in specific settings leveraging (1) common design assumptions across tasks and (2) a particular notion of task diversity, can break this barrier and yield rates for the leading term which decay as $O(\text{poly}(1/(nt)))$. However, the results and techniques used in both of these works are limited to the squared loss and linear task maps. Moreover, the notion of diversity in both cases arises purely from the linear-algebraic conditioning of the set of linear task maps. It is not clear from these works how to extend these ideas/techniques beyond the case-specific analyses therein.

## 2 Preliminaries

**Notation:** We use bold lower-case letters (e.g., $\mathbf{x}$) to refer to vectors and bold upper-case letters (e.g., $\mathbf{X}$) to refer to matrices. The norm $\| \cdot \|$ appearing on a vector or matrix refers to its $\ell_2$ norm or spectral norm respectively. We use the bracketed notation $[n] = \{1, \ldots, n\}$ as shorthand for integer sets. Generically, we will use "hatted" vectors and matrices (e.g, $\hat{\boldsymbol{\alpha}}$ and $\hat{\mathbf{B}}$) to refer to (random) estimators of their underlying population quantities. $\sigma_1(\mathbf{A}), \ldots, \sigma_r(\mathbf{A})$ will denote the sorted singular values (in decreasing magnitude) of a rank $r$ matrix $\mathbf{A}$. Throughout we will use $\mathcal{F}$ to refer to a function class of tasks mapping $\mathbb{R}^r \to \mathbb{R}$ and $\mathcal{H}$ to be a function class of features mapping $\mathbb{R}^d \to \mathbb{R}^r$. For the function class $\mathcal{F}$, we use $\mathcal{F}^{\otimes t}$ to refer its $t$-fold Cartesian product, i.e., $\mathcal{F}^{\otimes t} = \{\mathbf{f} \equiv (f_1, \ldots, f_t) \mid f_j \in \mathcal{F} \text{ for any } j \in [t]\}$. We use $\tilde{O}$ to denote an expression that hides polylogarithmic factors in all problem parameters.

### 2.1 Transfer learning with a shared representation

In our treatment of transfer learning, we assume that there exists a generic nonlinear feature representation that is shared across all tasks. Since this feature representation is shared, it can be utilized to transfer knowledge from existing tasks to new tasks. Formally, we assume that for a particular task $j$, we observe multiple data pairs $\{(\mathbf{x}_{ji}, y_{ji})\}$ (indexed over $i$) that are sampled i.i.d from an *unknown* distribution $\mathbb{P}_j$, supported over $\mathcal{X} \times \mathcal{Y}$ and defined as follows:

$$\mathbb{P}_j(\mathbf{x}, y) = \mathbb{P}_{f_j^\star \circ \mathbf{h}^\star}(\mathbf{x}, y) = \mathbb{P}_{\mathbf{x}}(\mathbf{x})\mathbb{P}_{y|\mathbf{x}}(y|f_j^\star \circ \mathbf{h}^\star(\mathbf{x})). \tag{1}$$

Here, $\mathbf{h}^\star : \mathbb{R}^d \to \mathbb{R}^r$ is the shared feature representation, and $f_j^\star : \mathbb{R}^r \to \mathbb{R}$ is a task-specific mapping. Note that we assume that the marginal distribution over $\mathcal{X}$—$\mathbb{P}_{\mathbf{x}}$—is common amongst all the tasks.

We consider transfer learning methods consisting of two phases. In the first phase (the training phase), $t$ tasks with $n$ samples per task are available for learning. Our objective in this phase is to learn the shared feature representation using the entire set of $nt$ samples from the first $j \in [t]$ tasks. In the second phase (the test phase), we are presented with $m$ fresh samples from a new task that we denote as the 0th task. Our objective in the test phase is to learn this new task based on both the fresh samples and the representation learned in the first phase.

Formally, we consider a two-stage Empirical Risk Minimization (ERM) procedure for transfer learning. Consider a function class $\mathcal{F}$ containing task-specific functions, and a function class $\mathcal{H}$ containing feature maps/representations. In the training phase, the empirical risk for $t$ training tasks is:

$$\hat{R}_{\text{train}}(\mathbf{f}, \mathbf{h}) := \frac{1}{nt} \sum_{j=1}^{t} \sum_{i=1}^{n} \ell(f_j \circ \mathbf{h}(\mathbf{x}_{ji}), y_{ji}), \tag{2}$$

where $\ell(\cdot, \cdot)$ is the loss function and $\mathbf{f} := (f_1, \ldots, f_t) \in \mathcal{F}^{\otimes t}$. Our estimator $\hat{\mathbf{h}}(\cdot)$ for the shared data representation is given by $\hat{\mathbf{h}} = \operatorname{argmin}_{\mathbf{h} \in \mathcal{H}} \min_{\mathbf{f} \in \mathcal{F}^{\otimes t}} \hat{R}_{\text{train}}(\mathbf{f}, \mathbf{h})$.

For the second stage, the empirical risk for learning the new task is defined as:

$$\hat{R}_{\text{test}}(f, \mathbf{h}) := \frac{1}{m} \sum_{i=1}^{m} \ell(f \circ \mathbf{h}(\mathbf{x}_{0i}), y_{0i}). \tag{3}$$

We estimate the underlying function $f_0^\star$ for task 0 by computing the ERM based on the feature representation learned in the first phase. That is, $\hat{f}_0 = \operatorname{argmin}_{f \in \mathcal{F}} \hat{R}_{\text{test}}(f, \hat{\mathbf{h}})$. We gauge the efficacy of the estimator $(\hat{f}_0, \hat{\mathbf{h}})$ by its excess risk on the new task, which we refer to as the *transfer learning risk*:

$$\text{Transfer Learning Risk} = R_{\text{test}}(\hat{f}_0, \hat{\mathbf{h}}) - R_{\text{test}}(f_0^\star, \mathbf{h}^\star). \tag{4}$$

Here, $R_{\text{test}}(\cdot, \cdot) = \mathbb{E}[\hat{R}_{\text{test}}(\cdot, \cdot)]$ is the population risk for the new task and the population risk over the $t$ training tasks is similarly defined as $R_{\text{train}}(\cdot, \cdot) = \mathbb{E}[\hat{R}_{\text{train}}(\cdot, \cdot)]$; both expectations are taken over the randomness in the training and test phase datasets respectively. The transfer learning risk measures the expected prediction risk of the function $(\hat{f}_0, \hat{\mathbf{h}})$ on a new datapoint for the 0th task, relative to the best prediction rule from which the data was generated—$f_0^\star \circ \mathbf{h}^\star$.

## 2.2 Model complexity

A well-known measure for the complexity of a function class is its Gaussian complexity. For a generic vector-valued function class $\mathcal{Q}$ containing functions $\mathbf{q}(\cdot) : \mathbb{R}^d \to \mathbb{R}^r$, and $N$ data points, $\bar{\mathbf{X}} = (\mathbf{x}_1, \ldots, \mathbf{x}_N)^\top$, the empirical Gaussian complexity is defined as

$$\hat{\mathfrak{S}}_{\bar{\mathbf{X}}}(\mathcal{Q}) = \mathbb{E}_{\mathbf{g}}[\sup_{\mathbf{q} \in \mathcal{Q}} \frac{1}{N} \sum_{k=1}^{r} \sum_{i=1}^{N} g_{ki} q_k(\mathbf{x}_i)], \qquad g_{ki} \sim \mathcal{N}(0,1) \ i.i.d.,$$

where $\mathbf{g} = \{g_{ki}\}_{k \in [r], i \in [N]}$, and $q_k(\cdot)$ is the $k$-th coordinate of the vector-valued function $\mathbf{q}(\cdot)$. We define the corresponding population Gaussian complexity as $\mathfrak{S}_N(\mathcal{Q}) = \mathbb{E}_{\bar{\mathbf{X}}}[\hat{\mathfrak{S}}_{\bar{\mathbf{X}}}(\mathcal{Q})]$, where the expectation is taken over the distribution of data samples $\bar{\mathbf{X}}$. Intuitively, $\mathfrak{S}_N(\mathcal{Q})$ measures the complexity of $\mathcal{Q}$ by the extent to which functions in the class $\mathcal{Q}$ can correlate with random noise $g_{ki}$.

# 3  Main Results

We now present our central theoretical results for the transfer learning problem. We first present statistical guarantees for the training phase and test phase separately. Then, we present a problem-agnostic definition of task diversity, followed by our generic end-to-end transfer learning guarantee. Throughout this section, we make the following standard, mild regularity assumptions on the loss function $\ell(\cdot, \cdot)$, the function class of tasks $\mathcal{F}$, and the function class of shared representations $\mathcal{H}$.

**Assumption 1** (Regularity conditions). *The following regularity conditions hold:*

- *The loss function $\ell(\cdot, \cdot)$ is B-bounded, and $\ell(\cdot, y)$ is L-Lipschitz for all $y \in \mathcal{Y}$.*

- *The function $f$ is $L(\mathcal{F})$-Lipschitz with respect to the $\ell_2$ distance, for any $f \in \mathcal{F}$.*

- *The composed function $f \circ \mathbf{h}$ is bounded: $\sup_{\mathbf{x} \in \mathcal{X}} |f \circ \mathbf{h}(\mathbf{x})| \leq D_{\mathcal{X}}$, for any $f \in \mathcal{F}, \mathbf{h} \in \mathcal{H}$.*

We also make the following realizability assumptions, which state that the true underlying task functions and the true representation are contained in the function classes $\mathcal{F}, \mathcal{H}$ over which the two-stage ERM oracle optimizes in (2) and (3).

**Assumption 2** (Realizability). *The true representation $\mathbf{h}^\star$ is contained in $\mathcal{H}$. Additionally, the true task specific functions $f_j^\star$ are contained in $\mathcal{F}$ for both the training tasks and new test task (i.e., for any $j \in [t] \cup \{0\}$).*

### 3.1 Learning shared representations

In order to measure "closeness" between the learned representation and true underlying feature representation, we need to define an appropriate distance measure between arbitrary representations. To this end, we begin by introducing the *task-averaged representation difference*, which captures the extent two representations $\mathbf{h}$ and $\mathbf{h}'$ differ in aggregate over the $t$ training tasks measured by the population train loss.

**Definition 1.** For a function class $\mathcal{F}$, $t$ functions $\mathbf{f} = (f_1, \ldots, f_t)$, and data $(\mathbf{x}_j, y_j) \sim \mathbb{P}_{f_j \circ \mathbf{h}}$ as in (1) for any $j \in [t]$, the **task-averaged representation difference** between representations $\mathbf{h}, \mathbf{h}' \in \mathcal{H}$ is:

$$\bar{d}_{\mathcal{F}, \mathbf{f}}(\mathbf{h}'; \mathbf{h}) = \frac{1}{t} \sum_{j=1}^{t} \inf_{f' \in \mathcal{F}} \mathbb{E}_{\mathbf{x}_j, y_j} \left\{ \ell(f' \circ \mathbf{h}'(\mathbf{x}_j), y_j) - \ell(f_j \circ \mathbf{h}(\mathbf{x}_j), y_j) \right\}.$$

Under this metric, we can show that the distance between a learned representation and the true underlying representation is controlled in the training phase. Our following guarantees also feature the *worst-case Gaussian complexity* over the function class $\mathcal{F}$, which is defined as:[3]

$$\bar{\mathfrak{G}}_n(\mathcal{F}) = \max_{\mathbf{Z} \in \mathcal{Z}} \hat{\mathfrak{G}}_{\mathbf{Z}}(\mathcal{F}), \text{ where } \mathcal{Z} = \{(\mathbf{h}(\mathbf{x}_1), \cdots, \mathbf{h}(\mathbf{x}_n)) \mid \mathbf{h} \in \mathcal{H}, \mathbf{x}_i \in \mathcal{X} \text{ for all } i \in [n]\}. \quad (5)$$

where $\mathcal{Z}$ is the domain induced by any set of $n$ samples in $\mathcal{X}$ and any representation $\mathbf{h} \in \mathcal{H}$. Moreover, we will always use the subscript $nt$, on $\mathfrak{G}_{nt}(\mathcal{Q}) = \mathbb{E}_{\mathbf{X}}[\hat{\mathfrak{G}}_{\mathbf{X}}(\mathcal{Q})]$, to refer to the population Gaussian complexity computed with respect to the data matrix $\mathbf{X}$ formed from the concatentation of the $nt$ training datapoints $\{\mathbf{x}_{ji}\}_{j=1, i=1}^{t, n}$. We can now present our training phase guarantee.

**Theorem 1.** *Let $\hat{\mathbf{h}}$ be an empirical risk minimizer of $\hat{R}_{train}(\cdot, \cdot)$ in (2). Then, if Assumptions 1 and 2 hold, with probability at least $1 - \delta$:*

$$\bar{d}_{\mathcal{F}, \mathbf{f}^\star}(\hat{\mathbf{h}}; \mathbf{h}^\star) \leq 16L\mathfrak{G}_{nt}(\mathcal{F}^{\otimes t} \circ \mathcal{H}) + 8B\sqrt{\frac{\log(2/\delta)}{nt}}$$

$$\leq 4096L \left[ \frac{D_{\mathcal{X}}}{(nt)^2} + \log(nt) \cdot [L(\mathcal{F}) \cdot \mathfrak{G}_{nt}(\mathcal{H}) + \bar{\mathfrak{G}}_n(\mathcal{F})] \right] + 8B\sqrt{\frac{\log(2/\delta)}{nt}}.$$

Theorem 1 asserts that the *task-averaged representation difference* (Definition 1) between our learned representation and the true representation is upper bounded by the population Gaussian complexity of the vector-valued function class $\mathcal{F}^{\otimes t} \circ \mathcal{H} = \{(f_1 \circ \mathbf{h}, \ldots, f_t \circ \mathbf{h}) : (f_1, \ldots, f_t) \in \mathcal{F}^{\otimes t}, \mathbf{h} \in \mathcal{H}\}$, plus a lower-order noise term. Up to logarithmic factors and lower-order terms, this Gaussian complexity can be further decomposed into the complexity of learning a representation in $\mathcal{H}$ with $nt$ samples—$L(\mathcal{F}) \cdot \mathfrak{G}_{nt}(\mathcal{H})$—and the complexity of learning a task-specific function in $\mathcal{F}$ using $n$

samples per task—$\bar{\mathfrak{G}}_n(\mathcal{F})$. For the majority of parametric function classes used in machine learning applications, $\mathfrak{G}_{nt}(\mathcal{H}) \sim \sqrt{C(\mathcal{H})/nt}$ and $\bar{\mathfrak{G}}_n(\mathcal{F}) \sim \sqrt{C(\mathcal{F})/n}$, where the function $C(\cdot)$ measures the intrinsic complexity of the function class (e.g., VC dimension, absolute dimension, or parameter norm [Wainwright, 2019]).

We now make several remarks on this result. First, Theorem 1 differs from standard supervised learning generalization bounds. Theorem 1 provides a bound on the distance between two representations as opposed to the empirical or population training risk, despite the lack of access to a direct signal from the underlying feature representation. Second, the decomposition of $\mathfrak{G}_{nt}(\mathcal{F}^{\otimes t} \circ \mathcal{H})$ into the individual Gaussian complexities of $\mathcal{H}$ and $\mathcal{F}$, leverages a novel chain rule for Gaussian complexities (see Theorem 7), which may be of independent interest. This chain rule (Theorem 7) can be viewed as a generalization of classical Gaussian comparison inequalities and results such as the Ledoux-Talagrand contraction principle [Ledoux and Talagrand, 2013]. Further details and comparisons to the literature for this chain rule can be found in Appendix B.2 (this result also avoids an absolute maxima over $\mathbf{x}_i \in \mathcal{X}$).

## 3.2 Transferring to new tasks

In addition to the *task-averaged representation difference*, we also introduce the *worst-case representation difference*, which captures the distance between two representations $\mathbf{h}', \mathbf{h}$ in the context of an arbitrary worst-case task-specific function $f_0 \in \mathcal{F}_0$.

**Definition 2.** For function classes $\mathcal{F}$ and $\mathcal{F}_0$ such that $f_0 \in \mathcal{F}_0$, and data $(\mathbf{x}, y) \sim \mathbb{P}_{f_0 \circ \mathbf{h}}$ as in (1), the **worst-case representation difference** between representations $\mathbf{h}, \mathbf{h}' \in \mathcal{H}$ is:

$$d_{\mathcal{F},\mathcal{F}_0}(\mathbf{h}'; \mathbf{h}) = \sup_{f_0 \in \mathcal{F}_0} \inf_{f' \in \mathcal{F}} \mathbb{E}_{\mathbf{x},y} \Big\{ \ell(f' \circ \mathbf{h}'(\mathbf{x}), y) - \ell(f_0 \circ \mathbf{h}(\mathbf{x}), y) \Big\}.$$

For flexibility we allow $\mathcal{F}_0$ to be distinct from $\mathcal{F}$ (although in most cases, we choose $\mathcal{F}_0 \subset \mathcal{F}$). The function class $\mathcal{F}_0$ is the set of new tasks on which we hope to generalize. The generalization guarantee for the test phase ERM estimator follows.

**Theorem 2.** *Let $\hat{f}_0$ be an empirical risk minimizer of $\hat{R}_{test}(\cdot, \hat{\mathbf{h}})$ in (3) for any feature representation $\hat{\mathbf{h}}$. Then if Assumptions 1 and 2 hold, and $f_0^\star \in \mathcal{F}_0$ for an unknown class $\mathcal{F}_0$, with probability at least $1 - \delta$:*

$$R_{test}(\hat{f}_0, \hat{\mathbf{h}}) - R_{test}(f_0^\star, \mathbf{h}^\star) \leq d_{\mathcal{F},\mathcal{F}_0}(\hat{\mathbf{h}}; \mathbf{h}^\star) + 16L \cdot \bar{\mathfrak{G}}_m(\mathcal{F}) + 8B\sqrt{\frac{\log(2/\delta)}{m}}$$

Here $\bar{\mathfrak{G}}_m(\mathcal{F})$ is again the worst-case Gaussian complexity[4] as defined in (5). Theorem 2 provides an excess risk bound for prediction on a new task in the test phase with two dominant terms. The first is the worst-case representation difference $d_{\mathcal{F},\mathcal{F}_0}(\hat{\mathbf{h}}; \mathbf{h}^\star)$, which accounts for the error of using a biased feature representation $\hat{\mathbf{h}} \neq \mathbf{h}^\star$ in the test ERM procedure. The second is the difficulty of learning $f_0^\star$ with $m$ samples, which is encapsulated in $\bar{\mathfrak{G}}_m(\mathcal{F})$.

## 3.3 Task diversity and end-to-end transfer learning guarantees

We now introduce the key notion of task diversity. Since the learner does not have direct access to a signal from the representation, they can only observe partial information about the representation channeled through the composite functions $f_j^\star \circ \mathbf{h}^\star$. If a particular direction/component in $\mathbf{h}^\star$ is not seen by a corresponding task $f_j^\star$ in the training phase, that component of the representation $\mathbf{h}^\star$ cannot be distinguished from a corresponding one in a spurious $\mathbf{h}'$. When this component is needed to predict on a new task corresponding to $f_0^\star$ which lies along that particular direction, transfer learning will not be possible. Accordingly, Definition 1 defines a notion of representation distance in terms of information channeled through the training tasks, while Definition 2 defines it in terms of an arbitrary new test task. Task diversity essentially encodes the ratio of these two quantities (i.e. how well the training tasks can cover the space captured by the representation $\mathbf{h}^\star$ needed to predict on new tasks). Intuitively, if all the task-specific functions were quite similar, then we would only expect the training

stage to learn about a narrow slice of the representation—making transferring to a generic new task difficult.

**Definition 3.** For a function class $\mathcal{F}$, we say $t$ functions $\mathbf{f} = (f_1, \ldots, f_t)$ are $(\nu, \epsilon)$-**diverse** over $\mathcal{F}_0$ for a representation $\mathbf{h}$, if uniformly for all $\mathbf{h}' \in \mathcal{H}$,

$$d_{\mathcal{F}, \mathcal{F}_0}(\mathbf{h}'; \mathbf{h}) \leq \bar{d}_{\mathcal{F}, \mathbf{f}}(\mathbf{h}'; \mathbf{h})/\nu + \epsilon.$$

Up to a small additive error $\epsilon$, diverse tasks ensure that the worst-case representation difference for the function class $\mathcal{F}_0$ is controlled when the task-averaged representation difference for a sequence of $t$ tasks $\mathbf{f}$ is small. Despite the abstraction in this definition of task diversity, it *exactly* recovers the notion of task diversity in Tripuraneni et al. [2020] and Du et al. [2020], where it is restricted to the special case of linear functions and quadratic loss. Our general notion allows us to move far beyond the linear-quadratic setting as we show in Section 4 and Appendix A.1.

We now utilize the definition of task diversity to merge our training phase and test phase results into an end-to-end transfer learning guarantee for generalization to the unseen task $f_0^\star \circ \mathbf{h}^\star$.

**Theorem 3.** *Let $(\cdot, \hat{\mathbf{h}})$ be an empirical risk minimizer of $\hat{R}_{train}(\cdot, \cdot)$ in (2), and $\hat{f}_0$ be an empirical risk minimizer of $\hat{R}_{test}(\cdot, \hat{\mathbf{h}})$ in (3) for the learned feature representation $\hat{\mathbf{h}}$. Then if Assumptions 1 and 2 hold, and the training tasks are $(\nu, \epsilon)$-diverse, with probability at least $1 - 2\delta$, the transfer learning risk in (4) is upper-bounded by:*

$$O\left( L \log(nt) \cdot \left[ \frac{L(\mathcal{F}) \cdot \mathfrak{G}_{nt}(\mathcal{H}) + \bar{\mathfrak{G}}_n(\mathcal{F})}{\nu} \right] + L\bar{\mathfrak{G}}_m(\mathcal{F}) + \frac{LD_\mathcal{X}}{\nu(nt)^2} + B\left[ \frac{1}{\nu} \cdot \sqrt{\frac{\log(2/\delta)}{nt}} + \sqrt{\frac{\log(2/\delta)}{m}} \right] + \epsilon \right).$$

Theorem 3 gives an upper bound on the transfer learning risk. The dominant terms in the bound are the three Gaussian complexity terms. For parametric function classes we expect $\mathfrak{G}_{nt}(\mathcal{H}) \sim \sqrt{C(\mathcal{H})/(nt)}$ and $\bar{\mathfrak{G}}_N(\mathcal{F}) \sim \sqrt{C(\mathcal{F})/N}$, where $C(\mathcal{H})$ and $C(\mathcal{F})$ capture the dimension-dependent size of the function classes. Therefore, when $L$ and $L(\mathcal{F})$ are constants, the leading-order terms for the transfer learning risk scale as $\tilde{O}(\sqrt{(C(\mathcal{H}) + t \cdot C(\mathcal{F}))/(nt)} + \sqrt{C(\mathcal{F})/m})$. A naive algorithm which simply learns the new task in isolation, ignoring the training tasks, has an excess risk scaling as $\tilde{O}(\sqrt{C(\mathcal{F} \circ \mathcal{H})/m}) \approx \tilde{O}(\sqrt{(C(\mathcal{H}) + C(\mathcal{F}))/m})$. Therefore, when $n$ and $t$ are sufficiently large, but $m$ is relatively small (i.e., the setting of few-shot learning), the performance of transfer learning is significantly better than the baseline of learning in isolation.

## 4 Applications

We now consider a varied set of applications to instantiate our general transfer learning framework. In each application, we first specify the function classes and data distributions we are considering as well as our assumptions. We then state the task diversity and the Gaussian complexities of the function classes, which together furnish the bounds on the *transfer learning risk*–from (4)–in Theorem 3.

### 4.1 Multitask Logistic Regression

We first instantiate our framework for one of the most frequently used classification methods—logistic regression. Consider the setting where the task-specific functions are linear maps, and the underlying representation is a projection onto a low-dimensional subspace. Formally, let $d \geq r$, and let the function classes $\mathcal{F}$ and $\mathcal{H}$ be:

$$\mathcal{F} = \{ f \mid f(\mathbf{z}) = \boldsymbol{\alpha}^\top \mathbf{z}, \; \boldsymbol{\alpha} \in \mathbb{R}^r, \; \|\boldsymbol{\alpha}\| \leq c_1 \}, \tag{6}$$

$$\mathcal{H} = \{ \mathbf{h} \mid \mathbf{h}(\mathbf{x}) = \mathbf{B}^\top \mathbf{x}, \; \mathbf{B} \in \mathbb{R}^{d \times r}, \; \mathbf{B} \text{ is a matrix with orthonormal columns} \}.$$

Here $\mathcal{X} = \mathbb{R}^d$, $\mathcal{Y} = \{0, 1\}$, and the measure $\mathbb{P}_\mathbf{x}$ is $\boldsymbol{\Sigma}$-sub-gaussian (see Definition 4) and $D$-bounded (i.e., $\|\mathbf{x}\| \leq D$ with probability one). We let the conditional distribution in (1) satisfy:

$$\mathbb{P}_{y|\mathbf{x}}(y = 1 | f \circ \mathbf{h}(\mathbf{x})) = \sigma(\boldsymbol{\alpha}^\top \mathbf{B}^\top \mathbf{x}),$$

where $\sigma(\cdot)$ is the sigmoid function with $\sigma(z) = 1/(1 + \exp(-z))$. We use the logistic loss $\ell(z, y) = -y \log(\sigma(z)) - (1 - y) \log(1 - \sigma(z))$. The true training tasks take the form $f_j^\star(\mathbf{z}) = (\boldsymbol{\alpha}_j^\star)^\top \mathbf{z}$ for all $j \in [t]$, and we let $\mathbf{A} = (\boldsymbol{\alpha}_1^\star, \ldots, \boldsymbol{\alpha}_t^\star)^\top \in \mathbb{R}^{t \times r}$. We make the following assumption on the training tasks being "diverse" and both the training and new task being normalized.

**Assumption 3.** $\sigma_r(\mathbf{A}^\top \mathbf{A}/t) = \tilde{\nu} > 0$ *and* $\|\boldsymbol{\alpha}_j^\star\| \leq O(1)$ *for* $j \in [t] \cup \{0\}$.

In this case where the $\mathcal{F}$ contains underlying linear task functions $\boldsymbol{\alpha}_j^\star \in \mathbb{R}^r$ (as in our examples in Section 4), our task diversity definition reduces to ensuring these task vectors span the entire $r$-dimensional space containing the output of the representation $\mathbf{h}(\cdot) \in \mathbb{R}^r$. This is quantitatively captured by the conditioning parameter $\tilde{\nu} = \sigma_r(\mathbf{A})$ which represents how spread out these vectors are in $\mathbb{R}^r$. The training tasks will be well-conditioned in the sense that $\sigma_1(\mathbf{A}^\top \mathbf{A}/t)/\sigma_r(\mathbf{A}^\top \mathbf{A}/t) \leq O(1)$ (w.h.p.) for example, if each $\boldsymbol{\alpha}_t \sim \mathcal{N}(0, \frac{1}{\sqrt{r}}\boldsymbol{\Sigma})$ i.i.d. with $\sigma_1(\boldsymbol{\Sigma})/\sigma_r(\boldsymbol{\Sigma}) \leq O(1)$.

Assumption 3 with natural choices of $\mathcal{F}_0$ and $\mathcal{F}$ establishes $(\Omega(\tilde{\nu}), 0)$-diversity as defined in Definition 3 (see Lemma 1). Finally, by standard arguments, we can bound the Gaussian complexity of $\mathcal{H}$ in this setting by $\mathfrak{G}_N(\mathcal{H}) \leq \tilde{O}(\sqrt{dr^2/N})$. We can also show that a finer notion of the Gaussian complexity for $\mathcal{F}$, serving as the analog of $\bar{\mathfrak{G}}_N(\mathcal{F})$, is upper bounded by $\tilde{O}(\sqrt{r/N})$. This is used to sharply bound the complexity of learning $\mathcal{F}$ in the training and test phases (see proof of Theorem 4 for more details). Together, these give the following guarantee.

**Theorem 4.** *If Assumption 3 holds,* $\mathbf{h}^\star(\cdot) \in \mathcal{H}$*, and* $\mathcal{F}_0 = \{ f \mid f(\mathbf{x}) = \boldsymbol{\alpha}^\top \mathbf{z}, \ \boldsymbol{\alpha} \in \mathbb{R}^r, \ \|\boldsymbol{\alpha}\| \leq c_2\}$*, then there exist constants* $c_1, c_2$ *such that the training tasks* $f_j^\star$ *are* $(\Omega(\tilde{\nu}), 0)$*-diverse over* $\mathcal{F}_0$*. Furthermore, if for a sufficiently large constant* $c_3$*,* $n \geq c_3(d + \log t)$*,* $m \geq c_3 r$*, and* $D \leq c_3(\min(\sqrt{dr^2}, \sqrt{rm}))$*, then with probability at least* $1 - 2\delta$*:*

$$\text{Transfer Learning Risk} \leq \tilde{O}\left( \frac{1}{\tilde{\nu}} \left( \sqrt{\frac{dr^2}{nt}} + \sqrt{\frac{r}{n}} \right) + \sqrt{\frac{r}{m}} \right).$$

A naive bound for logistic regression ignoring the training task data would have a guarantee $O(\sqrt{d/m})$. For $n$ and $t$ sufficiently large, the bound in Theorem 4 scales as $\tilde{O}(\sqrt{r/m})$, which is a significant improvement over $O(\sqrt{d/m})$ when $r \ll d$. Note that our result in fact holds with the empirical data-dependent quantities $\text{tr}(\boldsymbol{\Sigma}_\mathbf{X})$ and $\sum_{i=1}^r \sigma_i(\boldsymbol{\Sigma}_{\mathbf{X}_j})$ which can be much smaller then their counterparts $d, r$ in Theorem 4, if the data lies on/or close to a low-dimensional subspace[5].

## 4.2  Multitask Deep Neural Network Regression

We now consider the setting of real-valued neural network regression. Here the task-specific functions are linear maps as before, but the underlying representation is specified by a depth-$K$ vector-valued neural network:

$$\mathbf{h}(\mathbf{x}) = \mathbf{W}_K \sigma_{K-1}(\mathbf{W}_{K-1}(\sigma_{K-2}(\ldots \sigma(\mathbf{W}_1\mathbf{x})))). \tag{7}$$

Each $\mathbf{W}_k$ is a parameter matrix, and each $\sigma_k$ is a $\tanh$ activation function. We let $\|\mathbf{W}\|_{1,\infty} = \max_j(\sum_k |\mathbf{W}_{j,k}|)$ and $\|\mathbf{W}\|_{\infty \to 2}$ be the induced $\infty$-to-2 operator norm. Formally, $\mathcal{F}$ and $\mathcal{H}$ are[6]

$$\mathcal{F} = \{ f \mid f(\mathbf{z}) = \boldsymbol{\alpha}^\top \mathbf{z}, \ \boldsymbol{\alpha} \in \mathbb{R}^r, \ \|\boldsymbol{\alpha}\| \leq c_1 M(K)^2\}, \tag{8}$$

$$\mathcal{H} = \{\mathbf{h}(\cdot) \in \mathbb{R}^r \text{ in (7) for } \mathbf{W}_k : \|\mathbf{W}_k\|_{1,\infty} \leq M(k) \text{ for } k \in [K-1],$$

$$\max(\|\mathbf{W}_K\|_{1,\infty}, \|\mathbf{W}_K\|_{\infty \to 2}) \leq M(K), \text{ such that } \sigma_r\left(\mathbb{E}_\mathbf{x}[\mathbf{h}(\mathbf{x})\mathbf{h}(\mathbf{x})^\top]\right) > \Omega(1)\}.$$

We consider the setting where $\mathcal{X} = \mathbb{R}^d$, $\mathcal{Y} = \mathbb{R}$, and the measure $\mathbb{P}_\mathbf{x}$ is $D$-bounded. We also let the conditional distribution in (1) be induced by:

$$y = \boldsymbol{\alpha}^\top \mathbf{h}(\mathbf{x}) + \eta \text{ for } \boldsymbol{\alpha}, \mathbf{h} \text{ as in (8)}, \tag{9}$$

with additive noise $\eta$ bounded almost surely by $O(1)$ and independent of $\mathbf{x}$. We use the standard squared loss $\ell(\boldsymbol{\alpha}^\top \mathbf{h}(\mathbf{x}), y) = (y - \boldsymbol{\alpha}^\top \mathbf{h}(\mathbf{x}))^2$, and let the true training tasks take the form $f_j^\star(\mathbf{z}) = (\boldsymbol{\alpha}_j^\star)^\top \mathbf{z}$ for all $j \in [t]$, and set $\mathbf{A} = (\boldsymbol{\alpha}_1^\star, \ldots, \boldsymbol{\alpha}_t^\star)^\top \in \mathbb{R}^{t \times r}$ as in the previous example. Here we use exactly the same diversity/normalization assumption on the task-specific maps—Assumption 3—as in our logistic regression example.

Choosing $\mathcal{F}_0$ and $\mathcal{F}$ appropriately establishes a $(\Omega(\tilde{\nu}), 0)$-diversity as defined in Definition 3 (see Lemma 6). Standard arguments as well as results in Golowich et al. [2017] allow us to bound the Gaussian complexity terms as follows (see the proof of Theorem 5 for details):

$$\mathfrak{G}_N(\mathcal{H}) \leq \tilde{O}\left(\frac{rM(K) \cdot D\sqrt{K} \cdot \Pi_{k=1}^{K-1}M(k)}{\sqrt{N}}\right); \quad \bar{\mathfrak{G}}_N(\mathcal{F}) \leq \tilde{O}\left(\frac{M(K)^3}{\sqrt{N}}\right).$$

Combining these results yields the following end-to-end transfer learning guarantee.

**Theorem 5.** *If Assumption 3 holds, $\mathbf{h}^\star(\cdot) \in \mathcal{H}$, and $\mathcal{F}_0 = \{ f \mid f(\mathbf{z}) = \boldsymbol{\alpha}^\top \mathbf{z}, \ \boldsymbol{\alpha} \in \mathbb{R}^r, \ \|\boldsymbol{\alpha}\| \leq c_2 \}$, then there exist constants $c_1, c_2$ such that the training tasks $f_j^\star$ are $(\Omega(\tilde{\nu}), 0)$-diverse over $\mathcal{F}_0$. Further, if $M(K) \geq c_3$ for a universal constant $c_3$, then with probability at least $1 - 2\delta$:*

$$\textit{Transfer Learning Risk} \leq \tilde{O}\left(\frac{rM(K)^6 \cdot D\sqrt{K} \cdot \Pi_{k=1}^{K-1}M(k)}{\tilde{\nu}\sqrt{nt}} + \frac{M(K)^6}{\tilde{\nu}\sqrt{n}} + \frac{M(K)^6}{\sqrt{m}}\right).$$

The $\mathrm{poly}(M(K))$ dependence of the guarantee on the final-layer weights can likely be improved, but is dominated by the overhead of learning the complex feature map $\mathbf{h}^\star(\cdot)$ which has complexity $\mathrm{poly}(M(K)) \cdot D\sqrt{K} \cdot \Pi_{k=1}^{K-1}M(k)$. By contrast a naive algorithm which does not leverage the training samples would have a sample complexity of $\tilde{O}\left(\mathrm{poly}(M(K)) \cdot D\sqrt{K} \cdot \Pi_{k=1}^{K-1}M(k)/\sqrt{m}\right)$ via a similar analysis. Such a rate can be much larger than the bound in Theorem 5 when $nt \gg m$: exactly the setting relevant to that of few-shot learning for which ImageNet pretraining is often used.

### 4.3 Multitask Index Models

To illustrate the flexibility of our framework, in our final example, we consider a classical statistical model: the index model, which is often studied from the perspective of semiparametric estimation [Bickel et al., 1993]. As flexible tools for general-purpose, non-linear dimensionality reduction, index models have found broad applications in economics, finance, biology and the social sciences [Bickel et al., 1993, Li and Racine, 2007, Otwinowski et al., 2018]. This class of models has a different flavor then previously considered: the task-specific functions are nonparametric "link" functions, while the underlying representation is a one-dimensional projection. The formal set-up and full transfer generalization guarantees are deferred to Appendix A.1.

## 5 Conclusion

We present a framework for understanding the generalization abilities of generic models which share a common, underlying representation. In particular, our framework introduces a novel notion of task diversity through which we provide guarantees of a fast convergence rate, decaying with *all of the samples* for the transfer learning problem. One interesting direction for future consideration is investigating the effects of relaxing the common design and realizability assumptions on the results presented here. We also believe extending the results herein to accommodate "fine-tuning" of learned representations – that is, mildly adapting the learned representation extracted from training tasks to new, related tasks – is an important direction for future work.

## Broader Impact

As a theoretical paper we do not foresee our work directly having any societal consequences. However, transfer learning is a tool increasingly used in practical machine learning applications. Theoretical explorations related to transfer learning may help provide frameworks through which to reason about, and design, safer and more reliable algorithms.

## Acknowledgments and Disclosure of Funding

The authors thank Yeshwanth Cherapanamjeri for useful discussions. NT thanks the RISELab at U.C. Berkeley for support. In addition, this work was supported by the Army Research Office (ARO)

under contract W911NF-17-1-0304 as part of the collaboration between US DOD, UK MOD and UK Engineering and Physical Research Council (EPSRC) under the Multidisciplinary University Research Initiative (MURI).

## Footnotes

[1]A problem which is also often referred to as learning-to-learn (LTL).

[2]See Theorem 3 and discussion for a formal statement. Note our guarantees also hold for nonparametric function classes, but the scaling with $n$, $t$, $m$ may in general be different.

[3]Note that a stronger version of our results hold with a sharper, data-dependent version of the worst-case Gaussian complexity that eschews the absolute maxima over $\mathbf{x}_i$. See Corollary 1 and Theorem 7 for the formal statements.

[4]As before, a stronger version of this result holds with a sharper data-dependent version of the Gaussian complexity in lieu of $\bar{\mathfrak{G}}_m(\mathcal{F})$ (see Corollary 2).

[5]Here $\boldsymbol{\Sigma}_{\bar{\mathbf{X}}}$ denotes the empirical covariance of the data matrix $\bar{\mathbf{X}}$. See Corollary 3 for the formal statement of this sharper, more general result.

[6]For the following we make the standard assumption each parameter matrix $\mathbf{W}_k$ satisfies $\|\mathbf{W}_k\|_{1,\infty} \leq M(k)$ for each $j$ in the depth-$K$ network [Golowich et al., 2017], and that the feature map is well-conditioned.

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
