[Supplementary Material]

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

[7]Note as $\tilde{\nu}$ is problem-dependent, for a given underlying $\mathbf{f}^\star$, $\mathbf{h}^\star$, $\mathcal{F}_0$ problem instance, $\tilde{\nu}$ may be significantly greater than $\frac{1}{t}$. See the proof of Lemma 7 for details.

[8]note the $\mathbb{E}_\eta[L(\eta)]$ terms cancel in the expressions for $d_{\mathcal{F},\mathcal{F}_0}(\hat{\mathbf{h}}; \mathbf{h}^\star)$ and $\bar{d}_{\mathcal{F},\mathbf{f}^\star}(\hat{\mathbf{h}}; \mathbf{h}^\star)$.

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

# Appendices

**Notation:** Here we introduce several additional pieces of notation we will use throughout.

We use $\mathbb{E}_{\mathbf{x}}[\cdot]$ to refer to the expectation operator taken over the randomness in the vector $\mathbf{x}$ sampled from a distribution $\mathbb{P}_{\mathbf{x}}$. Throughout we will use $\mathcal{F}$ to refer exclusively to a scalar-valued function class of tasks and $\mathcal{H}$ to a vector-valued function class of features. For $\mathcal{F}$, we use $\mathcal{F}^{\otimes t}$ to refer its $t$-fold Cartesian product such that $(f_1, \ldots, f_t) \equiv \mathbf{f} \in \mathcal{F}^{\otimes t}$ for $f_j \in \mathcal{F}, j \in [t]$. We use $f(\mathbf{h})$ as shorthand for the function composition, $f \circ \mathbf{h}$. Similarly, we define the composed function class $\mathcal{F}(\mathcal{H}) = \{f(\mathbf{h}) : f \in \mathcal{F}, \mathbf{h} \in \mathcal{H}\}$ and its vector-valued version $\mathcal{F}^{\otimes t}(\mathcal{H}) = \{(f_1(\mathbf{h}), \ldots, f_t(\mathbf{h})) : f_j \in \mathcal{F}, j \in [t], \mathbf{h} \in \mathcal{H}\}$ with this shorthand. We will use $\gtrsim, \lesssim,$ and $\asymp$ to denote greater than, less than, and equal to up to a universal constant and use $\tilde{O}$ to denote an expression that hides polylogarithmic factors in all problem parameters.

In the context of the two-stage ERM procedure introduced in Section 2 we let the design matrix and responses $y_{ji}$ for the $j$th task be $\mathbf{X}_j$ and $\mathbf{y}_j$ for $j \in [t] \cup \{0\}$, and the entire design matrix and responses concatenated over all $j \in [t]$ tasks as $\mathbf{X}$ and $\mathbf{y}$ respectively. Given a design matrix $\bar{\mathbf{X}} = (\mathbf{x}_1, \ldots, \mathbf{x}_N)^\top$ (comprised of mean-zero random vectors) we will let $\boldsymbol{\Sigma}_{\bar{\mathbf{X}}} = \frac{1}{N}\bar{\mathbf{X}}^\top\bar{\mathbf{X}}$ denote its corresponding empirical covariance.

Recall we define the notions of the empirical and population Gaussian complexity for a generic vector-valued function class $\mathcal{Q}$ containing functions $\mathbf{q}(\cdot) : \mathbb{R}^d \to \mathbb{R}^r$, and data matrix $\mathbf{X}$ with $N$ datapoints as,

$$\hat{\mathfrak{G}}_{\mathbf{X}}(\mathcal{Q}) = \mathbb{E}_{\mathbf{g}}[\sup_{\mathbf{q} \in \mathcal{Q}} \frac{1}{N}\sum_{k=1}^r\sum_{i=1}^N g_{ki}q_k(\mathbf{x}_i)], \qquad \mathfrak{G}_N(\mathcal{Q}) = \mathbb{E}_{\mathbf{X}}[\hat{\mathfrak{G}}_{\mathbf{X}}(\mathcal{Q})] \qquad g_{ki} \sim \mathcal{N}(0,1) \; i.i.d.,$$

where for the latter population Gaussian complexity each its $N$ datapoints are drawn from the $\mathbb{P}_{\mathbf{x}}(\cdot)$ design distribution. Analogously to the above we can define the empirical and population Rademacher complexities for generic vector-valued functions as,

$$\hat{\mathfrak{R}}_{\mathbf{X}}(\mathcal{Q}) = \mathbb{E}_{\boldsymbol{\epsilon}}[\sup_{\mathbf{q} \in \mathcal{Q}} \frac{1}{N}\sum_{k=1}^r\sum_{i=1}^N \epsilon_{ki}q_k(\mathbf{x}_i)], \qquad \mathfrak{R}_N(\mathcal{Q}) = \mathbb{E}_{\mathbf{X}}[\hat{\mathfrak{R}}_{\mathbf{X}}(\mathcal{Q})] \qquad \epsilon_{ki} \sim \text{Rad}(\frac{1}{2}) \; i.i.d.$$

## A  Further Examples

Here we present further examples to exhibit the generality of our framework.

### A.1  Multitask Index Models

To illustrate the flexibility of our framework, in our final example, we consider a classical statistical model: the index model, which is often studied from the perspective of semiparametric estimation [Bickel et al., 1993]. As flexible tools for general-purpose, non-linear dimensionality reduction, index models have found broad applications in economics, finance, biology and the social sciences [Bickel et al., 1993, Li and Racine, 2007, Otwinowski et al., 2018]. This class of models has a different flavor then previously considered: the task-specific functions are nonparametric "link" functions, while the underlying representation is a one-dimensional projection. Formally, let the function classes $\mathcal{F}$ and $\mathcal{H}$ be:

$$\mathcal{F} = \{ f \mid f(\mathbf{z}) \text{ is a 1-Lipschitz, monotonic function bounded in } [0,1]\}, \tag{10}$$
$$\mathcal{H} = \{ \mathbf{h} \mid \mathbf{h}(\mathbf{x}) = \mathbf{b}^\top\mathbf{x}, \mathbf{b} \in \mathbb{R}^d, \|\mathbf{b}\| \le W\}.$$

We consider the setting where $\mathcal{X} = \mathbb{R}^d$, $\mathcal{Y} = \mathbb{R}$, the measure $\mathbb{P}_{\mathbf{x}}$ is $D$-bounded, and $DW \ge 1$. This matches the setting in Kakade et al. [2011]. The conditional distribution in (1) is induced by:

$$y = f(\mathbf{b}^\top\mathbf{x}) + \eta \;\text{ for } f, \mathbf{b} \text{ as in (7)},$$

with additive noise $\eta$ bounded almost surely by $O(1)$ and independent of $\mathbf{x}$. We use the robust $\ell_1$ loss, $\ell(f(\mathbf{b}^\top\mathbf{x}), y) = |y - f(\mathbf{b}^\top\mathbf{x})|$, in this example. Now, define $\mathcal{F}_t = \text{conv}\{f_1^\star, \ldots, f_t^\star\}$ as the

convex hull of the training task-specific functions $f_j^\star$. Given this, we define the $\tilde{\epsilon}$-enlargement of $\mathcal{F}_t$ by $\mathcal{F}_{t,\tilde{\epsilon}} = \{f : \exists \tilde{f} \in \mathcal{F}_t \text{ such that } \sup_z |f(z) - \tilde{f}(z)| \leq \tilde{\epsilon}\}$.

We prove a transfer generalization bound for $\mathcal{F}_0 = \mathcal{F}_{t,\tilde{\epsilon}}$, for which we can establish $(\tilde{\nu}, \tilde{\epsilon})$-diversity with $\tilde{\nu} \geq \frac{1}{t}$ as defined in Definition 3 (see Lemma 7). Standard arguments once again show that $\mathfrak{G}_N(\mathcal{H}) \leq O\left(\sqrt{(W^2\mathbb{E}_{\mathbf{X}}[\mathrm{tr}(\mathbf{\Sigma_X})]/N)}\right)$ and $\bar{\mathfrak{G}}_N(\mathcal{F}) \leq O\left(\sqrt{WD/N}\right)$ (see the proof of Theorem 6 for details). Together these give the following guarantee.

**Theorem 6.** *If $f_j^\star \in \mathcal{F}$ for $j \in [t]$, $\mathbf{h}^\star(\cdot) \in \mathcal{H}$, and $f_0^\star \in \mathcal{F}_0 = \mathcal{F}_{t,\tilde{\epsilon}}$, then the training tasks are $(\tilde{\nu}, \tilde{\epsilon})$-diverse over $\mathcal{F}_0$ where $\tilde{\nu} \geq \frac{1}{t}$. Further, with probability at least $1 - 2\delta$:*

$$\textit{Transfer Learning Risk} \leq \tilde{O}\left(\frac{1}{\tilde{\nu}} \cdot \left(\sqrt{\frac{W^2\mathbb{E}_{\mathbf{X}}[\mathrm{tr}(\mathbf{\Sigma_X})]}{nt}} + \sqrt{\frac{WD}{n}}\right) + \sqrt{\frac{WD}{m}}\right) + \tilde{\epsilon}.$$

As before, the complexity of learning the feature representation decays as $n \to \infty$. Hence if $\mathbb{E}[\mathrm{tr}(\mathbf{\Sigma_X})]$ is large, the aforementioned bound will provide significant savings over the bound which ignores the training phase samples of $O\left(\sqrt{(W^2\mathbb{E}_{\mathbf{X}}[\mathrm{tr}(\mathbf{\Sigma_X})])/m}\right) + O(\sqrt{WD/m})$. In this example, the problem-dependent parameter $\tilde{\nu}$ does not have a simple linear-algebraic interpretation. Indeed, in the worst-case it may seem the aforementioned bound degrades with $t$[7]. However, note that $\mathcal{F}_0 = \mathcal{F}_{t,\tilde{\epsilon}}$, so those unseen tasks which we hope to transfer to itself *grows* with $t$ unlike in the previous examples. The difficulty of the transfer learning problem also increases as $t$ increases. Finally, this example utilizes the full power of $(\nu, \epsilon)$-diversity by permitting robust generalization to tasks outside $\mathcal{F}_t$, at the cost of a bias term $\tilde{\epsilon}$ in the generalization guarantee.

# B   Proofs in Section 3

Here we include the proofs of central generalization guarantees and the Gaussian process chain rule used in its proof.

## B.1   Training Phase/Test Phase Proofs

In all the following definitions $(\mathbf{x}_j, y_j)$ refer to datapoint drawn from the $j$th component of the model in (1). We first include the proof of Theorem 1 which shows that minimizing the training phase ERM objective controls the task-average distance between the underlying feature representation $\mathbf{h}$ and learned feature representation $\hat{\mathbf{h}}$.

*Proof of Theorem 1.* For fixed $\mathbf{f}', \mathbf{h}'$, define the centered training risk as,

$$L(\mathbf{f}', \mathbf{h}', \mathbf{f}^\star, \mathbf{h}^\star) = \frac{1}{t}\sum_{j=1}^{t} \mathbb{E}_{\mathbf{x}_j, y_j}\left\{\ell(f_j' \circ \mathbf{h}'(\mathbf{x}_j), y_j) - \ell(f_j^\star \circ \mathbf{h}^\star(\mathbf{x}_j), y_j)\right\}.$$

and its empirical counterpart,

$$\hat{L}(\mathbf{f}', \mathbf{h}', \mathbf{f}^\star, \mathbf{h}^\star) = \frac{1}{t}\sum_{j=1}^{t}\sum_{i=1}^{n}\left\{\ell(f_j' \circ \mathbf{h}'(\mathbf{x}_{ji}), y_{ji}) - \mathbb{E}_{\mathbf{x},y}[\ell(f_j^\star \circ \mathbf{h}^\star(\mathbf{x}), y)]\right\}$$

Now if $\tilde{\mathbf{f}}$ denotes a minimizer of the former expression for fixed $\hat{\mathbf{h}}$, in the sense that $\tilde{\mathbf{f}} = \frac{1}{t}\sum_{j=1}^{t} \arg\inf_{f_j' \in \mathcal{F}} \mathbb{E}_{\mathbf{x}_j, y_j}\left\{\ell(f_j' \circ \hat{\mathbf{h}}(\mathbf{x}_j), y_j) - \ell(f_j^\star \circ \mathbf{h}^\star(\mathbf{x}_j), y_j)\right\}$, then by definition, we have that $\bar{d}_{\mathcal{F}, \mathbf{f}^\star}(\hat{\mathbf{h}}; \mathbf{h}^\star)$ equals the former expression. We first decompose the average distance using the pair $(\hat{\mathbf{f}}, \hat{\mathbf{h}})$. Recall the pair $(\hat{\mathbf{f}}, \hat{\mathbf{h}})$ refers to the empirical risk minimizer in (2).

$$L(\tilde{\mathbf{f}}, \hat{\mathbf{h}}, \mathbf{f}^\star, \mathbf{h}^\star) - L(\mathbf{f}^\star, \mathbf{h}^\star, \mathbf{f}^\star, \mathbf{h}^\star) = \underbrace{L(\tilde{\mathbf{f}}, \hat{\mathbf{h}}, \mathbf{f}^\star, \mathbf{h}^\star) - L(\hat{\mathbf{f}}, \hat{\mathbf{h}}, \mathbf{f}^\star, \mathbf{h}^\star)}_{a} + L(\hat{\mathbf{f}}, \hat{\mathbf{h}}, \mathbf{f}^\star, \mathbf{h}^\star) - L(\mathbf{f}^\star, \mathbf{h}^\star, \mathbf{f}^\star, \mathbf{h}^\star)$$

Note that by definition of the $\tilde{\mathbf{f}}$, $a \leq 0$. The second pair can be controlled via the canonical risk decomposition,

$$L(\hat{\mathbf{f}}, \hat{\mathbf{h}}, \mathbf{f}^\star, \mathbf{h}^\star) - L(\mathbf{f}^\star, \mathbf{h}^\star, \mathbf{f}^\star, \mathbf{h}^\star) = \underbrace{L(\hat{\mathbf{f}}, \hat{\mathbf{h}}, \mathbf{f}^\star, \mathbf{h}^\star) - \hat{L}(\hat{\mathbf{f}}, \hat{\mathbf{h}}, \mathbf{f}^\star, \mathbf{h}^\star)}_{b} + \underbrace{\hat{L}(\hat{\mathbf{f}}, \hat{\mathbf{h}}, \mathbf{f}^\star, \mathbf{h}^\star) - \hat{L}(\mathbf{f}^\star, \mathbf{h}^\star, \mathbf{f}^\star, \mathbf{h}^\star)}_{c} +$$

$$\underbrace{\hat{L}(\mathbf{f}^\star, \mathbf{h}^\star, \mathbf{f}^\star, \mathbf{h}^\star) - L(\mathbf{f}^\star, \mathbf{h}^\star, \mathbf{f}^\star, \mathbf{h}^\star)}_{d}.$$

By definition $c \leq 0$ (note this inequality uses the realizability in Assumption 2) and $b, d \leq \sup_{\mathbf{f} \in \mathcal{F}^{\otimes t}, \mathbf{h} \in \mathcal{H}} |R_{\text{train}}(\mathbf{f}, \mathbf{h}) - \hat{R}_{\text{train}}(\mathbf{f}, \mathbf{h})|$. By an application of the bounded differences inequality and a standard symmetrization argument (see for example Wainwright [2019, Theorem 4.10] we have that,

$$\sup_{\mathbf{f} \in \mathcal{F}^{\otimes t}, \mathbf{h} \in \mathcal{H}} |R_{\text{train}}(\mathbf{f}, \mathbf{h}) - \hat{R}_{\text{train}}(\mathbf{f}, \mathbf{h})| \leq 2\mathfrak{R}_{nt}(\ell(\mathcal{F}^{\otimes t}(\mathcal{H}))) + 2B\sqrt{\frac{\log(1/\delta)}{nt}}$$

with probability at least $1 - 2\delta$.

It remains to decompose the leading Rademacher complexity term. First we center the functions to $\ell_{ji}(f_j \circ \mathbf{h}(\mathbf{x}_{ji}), y_{ji}) = \ell(f_j \circ \mathbf{h}(\mathbf{x}_{ji}), y_{ji}) - \ell(0, y_{ji})$. Then noting $|\ell_{ji}(0, y_{ji})| \leq B$, the constant-shift property of Rademacher averages Wainwright [2019, Exercise 4.7c] gives,

$$\mathbb{E}_{\boldsymbol{\epsilon}}\Big[\sup_{\mathbf{f} \in \mathcal{F}^{\otimes t}, \mathbf{h} \in \mathcal{H}} \frac{1}{nt} \sum_{j=1}^{t} \sum_{i=1}^{n} \epsilon_{ij} \ell(f_j \circ \mathbf{h}(\mathbf{x}_{ji}), y_{ji})\Big] \leq \mathbb{E}_{\boldsymbol{\epsilon}}\Big[\sup_{\mathbf{f} \in \mathcal{F}, \mathbf{h} \in \mathcal{H}} \frac{1}{nt} \sum_{j=1}^{t} \sum_{i=1}^{n} \epsilon_{ij} \ell_{ij}(f_j \circ \mathbf{h}(\mathbf{x}_{ji}), y_{ji})\Big] + \frac{B}{\sqrt{nt}}$$

Now note each $\ell_{ij}(\cdot, \cdot)$ is $L$-Lipschitz in its first coordinate uniformly for every choice of the second coordinate (and by construction centered in its first coordinate). So, defining the set $S = \{(f_1 \circ \mathbf{h}(\mathbf{x}_{1i}), \ldots, f_j \circ \mathbf{h}(\mathbf{x}_{ji}), \ldots, f_t \circ \mathbf{h}(\mathbf{x}_{ti}))) : j \in [t], f_j \in \mathcal{F}, \mathbf{h} \in \mathcal{H}\} \subseteq \mathbb{R}^{tn}$, and applying the contraction principle Ledoux and Talagrand [2013, Theorem 4.12] over this set shows,

$$\mathbb{E}_{\boldsymbol{\epsilon}}\Big[\sup_{\mathbf{f} \in \mathcal{F}^{\otimes t}, \mathbf{h} \in \mathcal{H}} \frac{1}{nt} \sum_{j=1}^{t} \sum_{i=1}^{n} \epsilon_{ij} \ell_{ij}(f_j \circ \mathbf{h}(\mathbf{x}_{ji}), y_{ji})\Big] \leq 2L \cdot \mathfrak{R}_{nt}(\mathcal{F}^{\otimes t}(\mathcal{H})). \tag{11}$$

Combining gives,

$$\sup_{\mathbf{f} \in \mathcal{F}^{\otimes t}, \mathbf{h} \in \mathcal{H}} |R_{\text{train}}(\mathbf{f}, \mathbf{h}) - \hat{R}_{\text{train}}(\mathbf{f}, \mathbf{h})| \leq 4L \cdot \mathfrak{R}_{nt}(\mathcal{F}^{\otimes t}(\mathcal{H})) + \frac{4B\sqrt{\log(1/\delta)}}{\sqrt{nt}}$$

with probability $1 - 2\delta$. Now note by [Ledoux and Talagrand, 2013, p.97] empirical Rademacher complexity is upper bounded by empirical Gaussian complexity: $\hat{\mathfrak{R}}_{\mathbf{X}}(\mathcal{F}^{\otimes t}(\mathcal{H})) \leq \sqrt{\frac{\pi}{2}}\hat{\mathfrak{G}}_{\mathbf{X}}(\mathcal{F}^{\otimes t}(\mathcal{H}))$. Taking expectations of this and combining with the previous display yields the first inequality in the theorem statement.

The last remaining step hinges on Theorem 7 to decompose the Gaussian complexity over $\mathcal{F}$ and $\mathcal{H}$. A direct application of Theorem 7 gives the conclusion that,

$$\hat{\mathfrak{G}}_{\mathbf{X}}(\mathcal{F}^{\otimes t}(\mathcal{H})) \leq 128\left(\frac{D_{\mathbf{X}}}{(nt)^2} + C(\mathcal{F}^{\otimes t}(\mathcal{H})) \cdot \log(nt)\right)$$

where $C(\mathcal{F}^{\otimes t}(\mathcal{H}); \mathbf{X}) = L(\mathcal{F}) \cdot \hat{\mathfrak{G}}_{\mathbf{X}}(\mathcal{H}) + \max_{\mathbf{Z} \in \mathcal{Z}} \hat{\mathfrak{G}}_{\mathbf{Z}}(\mathcal{F})$ where $\mathcal{Z} = \{\mathbf{h}(\bar{\mathbf{X}}) : \mathbf{h} \in \mathcal{H}, \bar{\mathbf{X}} \in \cup_{j=1}^{t}\{\mathbf{X}_j\}\}$. By definition of $D_{\mathbf{X}}$ we have $D_{\mathbf{X}} \leq 2D_{\mathcal{X}}$ and similarly that $\max_{\mathbf{Z} \in \mathcal{Z}} \hat{\mathfrak{G}}_{\mathbf{Z}}(\mathcal{F}) \leq \max_{\mathbf{Z} \in \mathcal{Z}_1} \hat{\mathfrak{G}}_{\mathbf{Z}}(\mathcal{F})$ for $\mathcal{Z}_1 = \{(\mathbf{h}(\mathbf{x}_1), \cdots, \mathbf{h}(\mathbf{x}_n)) \mid \mathbf{h} \in \mathcal{H}, \mathbf{x}_i \in \mathcal{X} \text{ for all } i \in [n]\}$. Taking expectations over $\mathbf{X}$ in this series of relations and assembling the previous bounds gives the conclusion after rescaling $\delta$. $\qquad\square$

An analogous statement holds both in terms of a sharper notion of the worst-case Gaussian complexity and in terms of empirical Gaussian complexities.

**Corollary 1.** *In the setting of Theorem 1,*

$$\bar{d}_{\mathcal{F},\mathbf{f}^\star}(\hat{\mathbf{h}};\mathbf{h}^\star) \leq 4096L\left[\frac{D_{\mathcal{X}}}{(nt)^2} + \log(nt)\cdot[L(\mathcal{F})\cdot\mathfrak{G}_{\mathbf{X}}(\mathcal{H}) + \mathbb{E}_{\mathbf{X}}[\max_{\mathbf{Z}\in\mathcal{Z}}\hat{\mathfrak{G}}_{\mathbf{Z}}(\mathcal{F})]\right] + 8B\sqrt{\frac{\log(1/\delta)}{n}}$$

*with probability $1-2\delta$ for $\mathcal{Z} = \{\mathbf{h}(\bar{\mathbf{X}}) : \mathbf{h}\in\mathcal{H}, \bar{\mathbf{X}}\in\cup_{j=1}^t\{\mathbf{X}_j\}\}$. Furthermore,*

$$\bar{d}_{\mathcal{F},\mathbf{f}^\star}(\hat{\mathbf{h}};\mathbf{h}^\star) \leq 16\hat{\mathfrak{G}}_{\mathbf{X}}(\mathcal{F}^{\otimes t}(\mathcal{H})) + 16B\sqrt{\frac{\log(1/\delta)}{n}} \leq$$

$$4096L\left[\frac{D_{\mathcal{X}}}{(nt)^2} + \log(nt)\cdot[L(\mathcal{F})\cdot\hat{\mathfrak{G}}_{\mathbf{X}}(\mathcal{H}) + \max_{\mathbf{Z}\in\mathcal{Z}}\hat{\mathfrak{G}}_{\mathbf{Z}}(\mathcal{F})]\right] + 16B\sqrt{\frac{\log(1/\delta)}{n}}$$

*with probability at least $1-4\delta$.*

*Proof.* The argument follows analogously to the proof of Theorem 1. The first statement follows identically by avoiding the relaxation–$\max_{\mathbf{Z}\in\mathcal{Z}}\hat{\mathfrak{G}}_{\mathbf{Z}}(\mathcal{F}) \leq \max_{\mathbf{Z}\in\mathcal{Z}_1}\hat{\mathfrak{G}}_{\mathbf{Z}}(\mathcal{F})$ for $\mathcal{Z}_1 = \{(\mathbf{h}(\mathbf{x}_1),\cdots,\mathbf{h}(\mathbf{x}_n)) \mid \mathbf{h}\in\mathcal{H}, \mathbf{x}_i\in\mathcal{X} \text{ for all } i\in[n]\}$–after applying Theorem 7 in the proof of Theorem 1.

The second statement also follows by a direct modification of the proof of Theorem 1. In the proof another application of the bounded differences inequality would show that $|\mathfrak{R}_{nt}(\mathcal{F}^{\otimes t}(\mathcal{H})) - \hat{\mathfrak{R}}_{\mathbf{X}}((\mathcal{F}^{\otimes t}(\mathcal{H}))| \leq 4B\sqrt{\frac{\log(1/\delta)}{nt}}$ with probability $1-2\delta$. Applying this inequality after (11) and union bounding over this event and the event in the theorem, followed by the steps in Theorem 1, gives the result after an application of Theorem 7. $\square$

We now show how the definition of task diversity in Definition 3 and minimizing the training phase ERM objective allows us to transfer a fixed feature representation $\hat{\mathbf{h}}$ and generalize to a new task-specific mapping $f_0$.

*Proof of Theorem 2.* Note $\tilde{f}_0 = \arg\min_{f\in\mathcal{F}} R_{\text{test}}(f,\hat{\mathbf{h}})$–it is a minimizer of the population test risk loaded with the fixed feature representation $\hat{\mathbf{h}}$. The approach to controlling this term uses the canonical risk decomposition,

$$R_{\text{test}}(\hat{f}_0,\hat{\mathbf{h}}) - R_{\text{test}}(\tilde{f}_0,\hat{\mathbf{h}}) = \underbrace{R_{\text{test}}(\hat{f}_0,\hat{\mathbf{h}}) - \hat{R}_{\text{test}}(\hat{f}_0,\hat{\mathbf{h}})}_{a} + \underbrace{\hat{R}_{\text{test}}(\hat{f}_0,\hat{\mathbf{h}}) - \hat{R}_{\text{test}}(\tilde{f}_0,\hat{\mathbf{h}})}_{b} + \underbrace{\hat{R}_{\text{test}}(\tilde{f}_0,\hat{\mathbf{h}}) - R_{\text{test}}(\tilde{f}_0,\hat{\mathbf{h}})}_{c}$$

First by definition, $b\leq 0$. Now a standard uniform convergence/symmetrization argument which also follows the same steps as in the proof of Theorem 1,

$$a + c \leq 16L\cdot\mathbb{E}_{\mathbf{X}_0}[\hat{\mathfrak{G}}_{\mathbf{Z}_{\hat{\mathbf{h}}}}(\mathcal{F})] + 8B\sqrt{\frac{\log(1/\delta)}{m}} \leq 16L\max_{\hat{\mathbf{h}}\in\mathcal{H}}\mathbb{E}_{\mathbf{X}_0}[\hat{\mathfrak{G}}_{\mathbf{Z}_{\hat{\mathbf{h}}}}(\mathcal{F})] + 8B\sqrt{\frac{\log(1/\delta)}{m}}$$

for $\mathbf{Z}_{\hat{\mathbf{h}}} = \hat{\mathbf{h}}(\mathbf{X}_0)$, with probability at least $1-2\delta$. The second inequality simply uses the fact that the map $\hat{\mathbf{h}}$ is fixed, and independent of the randomness in the test data. The bias from using an imperfect feature representation $\hat{\mathbf{h}}$ in lieu of $\mathbf{h}$ arises in $R_{\text{test}}(\tilde{f}_0,\hat{\mathbf{h}})$. For this term,

$$R_{\text{test}}(\tilde{f}_0,\hat{\mathbf{h}}) - R_{\text{test}}(f_0,\mathbf{h}^\star) = \inf_{\tilde{f}_0\in\mathcal{F}}\{R_{\text{test}}(\tilde{f}_0,\hat{\mathbf{h}}) - R_{\text{test}}(f_0,\mathbf{h}^\star)\} \leq \sup_{f_0\in\mathcal{F}_0}\inf_{\tilde{f}_0\in\mathcal{F}}\{L(\tilde{f}_0,\hat{\mathbf{h}}) - L(f_0,\mathbf{h}^\star)\} =$$

$$d_{\mathcal{F},\mathcal{F}_0}(\mathbf{h};\hat{\mathbf{h}})$$

To obtain the final theorem statement we use an additional relaxation on the Gaussian complexity term for ease of presentation,

$$\max_{\hat{\mathbf{h}}\in\mathcal{H}}\mathbb{E}_{\mathbf{X}_0}[\hat{\mathfrak{G}}_{\mathbf{Z}_{\hat{\mathbf{h}}}}(\mathcal{F})] \leq \bar{\mathfrak{G}}_m(\mathcal{F}).$$

Combining terms gives the conclusion. $\square$

We also present a version of Theorem 2 which can possess better dependence on the boundedness parameter in the noise terms and has data-dependence in the Gaussian complexities. As before our guarantees can be stated both in terms of population or empirical quantities. The result appeals to the functional Bernstein inequality instead of the bounded differences inequality in the concentration step. Although we only state (and use) this guarantee for the test phase generalization an analogous statement can be shown to hold for Theorem 1. Throughout the following, we use $(\mathbf{x}_i, y_i) \sim \mathbb{P}_{f_0 \circ \mathbf{h}}$ for $i \in [m]$ for ease of notation.

**Corollary 2.** *In the setting of Theorem 2, assuming the loss function $\ell$ satisfies the centering $\ell(0, y) = 0$ for all $y \in \mathcal{Y}$,*

$$R_{test}(\hat{f}_0, \hat{\mathbf{h}}) - R_{test}(f_0^\star, \mathbf{h}^\star) \le d_{\mathcal{F}, \mathcal{F}_0}(\hat{\mathbf{h}}; \mathbf{h}^\star) + 16L \cdot \mathbb{E}_{\mathbf{X}_0}[\hat{\mathfrak{G}}_{\mathbf{Z}_{\hat{\mathbf{h}}}}(\mathcal{F})] + 4\sigma\sqrt{\frac{\log(2/\delta)}{m}} + 50B\frac{\log(2/\delta)}{m}$$

*for $\mathbf{Z}_{\hat{\mathbf{h}}} = \hat{\mathbf{h}}(\mathbf{X}_0)$, with probability at least $1 - \delta$. Here the maximal variance $\sigma^2 = \frac{1}{m} \sup_{f \in \mathcal{F}} \sum_{i=1}^m Var(\ell(f \circ \hat{\mathbf{h}}(\mathbf{x}_i), y_i))$. Similarly we have that,*

$$R_{test}(\hat{f}_0, \hat{\mathbf{h}}) - R_{test}(f_0^\star, \mathbf{h}^\star) \le d_{\mathcal{F}, \mathcal{F}_0}(\hat{\mathbf{h}}; \mathbf{h}^\star) + 32L \cdot \hat{\mathfrak{G}}_{\mathbf{Z}_{\hat{\mathbf{h}}}}(\mathcal{F}) + 8\sigma\sqrt{\frac{\log(2/\delta)}{m}} + 100B\frac{\log(2/\delta)}{m}$$

*with probability at least $1 - 2\delta$.*

*Proof of Corollary 2.* The proof is identical to the proof of Theorem 2 save in how the concentration argument is performed. Namely in the notation of Theorem 2, we upper bound,

$$a + c \le 2 \sup_{f \in \mathcal{F}} |\hat{R}_{test}(f, \hat{\mathbf{h}}) - R_{test}(f, \hat{\mathbf{h}})| = 2Z$$

Note by definition $\mathbb{E}_{\mathbf{X}_0, \mathbf{y}_0}[\hat{R}_{test}(f, \hat{\mathbf{h}})] = R_{test}(f, \hat{\mathbf{h}})$, where $\hat{R}_{test}(f, \hat{\mathbf{h}}) = \frac{1}{m} \sum_{i=1}^m \ell(f \circ \hat{\mathbf{h}}(\mathbf{x}_i), y_i)$, and the expectation is taken over the test-phase data. Instead of applying the bounded differences inequality to control the fluctuations of this term we apply a powerful form of the functional Bernstein inequality due to Massart et al. [2000]. Applying Massart et al. [2000, Theorem 3] therein, we can conclude,

$$Z \le (1 + \epsilon)\mathbb{E}[Z] + \frac{\sigma}{\sqrt{n}}\sqrt{2\kappa \log(\frac{1}{\delta})} + \kappa(\epsilon)\frac{B}{m}\log(\frac{1}{\delta})$$

for $\kappa = 2$, $\kappa(\epsilon) = 2.5 + \frac{32}{\epsilon}$ and $\sigma^2 = \frac{1}{m} \sup_{f \in \mathcal{F}} \sum_{i=1}^m Var(\ell(f \circ \hat{\mathbf{h}}(\mathbf{x}_i), y_i))$. We simply take $\epsilon = 1$ for our purposes, which gives the bound,

$$Z \le 2\mathbb{E}[Z] + 4\frac{\sigma}{\sqrt{m}}\sqrt{\log(\frac{1}{\delta})} + 35\frac{B}{m}\log(\frac{1}{\delta})$$

Next note a standard symmetrization argument shows that $\mathbb{E}[Z] \le 2\mathbb{E}_{\mathbf{X}_0, \mathbf{y}_0}[\hat{\mathfrak{R}}_{\mathbf{Z}_{\hat{\mathbf{h}}}}(\ell \circ \mathcal{F})]$ for $\mathbf{Z}_{\hat{\mathbf{h}}} = \hat{\mathbf{h}}(\mathbf{X}_0)$. Following the proof of Theorem 2 but eschewing the unnecessary centering step in the application of the contraction principle shows that, $\hat{\mathfrak{R}}_{\mathbf{Z}_{\hat{\mathbf{h}}}}(\ell \circ \mathcal{F}) \le 2L \cdot \hat{\mathfrak{R}}_{\mathbf{Z}_{\hat{\mathbf{h}}}}(\mathcal{F})$. Upper bounding empirical Rademacher complexity by Gaussian complexity and following the steps of Theorem 2 gives the first statement.

The second statement in terms of empirical quantities follows similarly. First the population Rademacher complexity can be converted into an empirical Rademacher complexity using a similar concentration inequality based result which appears in a convenient form in Bartlett et al. [2005, Lemma A.4 (i)]. Directly applying this result (with $\alpha = \frac{1}{2}$) shows that,

$$\mathbb{E}_{\mathbf{X}_0, \mathbf{y}_0}[\hat{\mathfrak{R}}_{\mathbf{Z}_{\hat{\mathbf{h}}}}(\ell \circ \mathcal{F})] \le 2\hat{\mathfrak{R}}_{\mathbf{Z}_{\hat{\mathbf{h}}}}(\ell \circ \mathcal{F}) + \frac{8B\log(\frac{1}{\delta})}{m}$$

with probability at least $1 - \delta$. The remainder of the argument follows exactly as before and as in the proof of Theorem 2 along with another union bound. $\square$

The proof of Theorem 3 is almost immediate.

*Proof of Theorem 3.* The result follows immediately by combining Theorem 1, Theorem 2, and the definition of task diversity along with a union bound over the two events on which Theorems 1 and 2 hold. $\square$

## B.2 A User-Friendly Chain Rule for Gaussian Complexity

We provide the formal statement and the proof of the chain rule for Gaussian complexity that is used in the main text to decouple the complexity of learning the class $\mathcal{F}^{\otimes t}(\mathcal{H})$ into the complexity of learning each individual class. We believe this result may be a technical tool that is of more general interest for a variety of learning problems where compositions of function classes naturally arise.

Intuitively, the chain rule (Theorem 7) can be viewed as a generalization of the Ledoux-Talagrand contraction principle which shows that for a *fixed*, centered $L$-Lipschitz function $\phi$, $\hat{\mathfrak{G}}_{\mathbf{X}}(\phi(\mathcal{F})) \leq 2L\hat{\mathfrak{G}}_{\mathbf{X}}(\mathcal{F})$. However, as we are learning *both* $\mathbf{f} \in \mathcal{F}^{\otimes t}$ (which is not fixed) and $\mathbf{h} \in \mathcal{H}$, $\hat{\mathfrak{G}}_{\mathbf{X}}(\mathcal{F}^{\otimes t} \circ \mathcal{H})$ features a suprema over both $\mathcal{F}^{\otimes t}$ and $\mathcal{H}$.

A comparable result for Gaussian processes to our Theorem 7 is used in Maurer et al. [2016] for multi-task learning applications, drawing on the chain rule of Maurer [2016]. Although their result is tighter with respect to logarithmic factors, it cannot be written purely in terms of Gaussian complexities. Rather, it includes a worst-case "Gaussian-like" average (Maurer et al. [2016, Eq. 4]) in lieu of $\hat{\mathfrak{G}}_{\mathcal{Z}}(\mathcal{F})$ in Theorem 7. In general, it is not clear how to sharply bound this term beyond the using existing tools in the learning theory literature. The terms appearing in Theorem 7 can be bounded, in a direct and modular fashion, using the wealth of existing results and tools in the learning theory literature.

Our proof technique and that of Maurer [2016] both hinge on several properties of Gaussian processes. Maurer [2016] uses a powerful generalization of the Talagrand majorizing measure theorem to obtain their chain rule. We take a different path. First we use the entropy integral to pass to the space of covering numbers–where the metric properties of the distance are used to decouple the features and tasks. Finally an appeal to Gaussian process lower bounds are used to come back to expression that involves only Gaussian complexities.

We will use the machinery of empirical process theory throughout this section so we introduce several useful definitions we will need. We define the empirical $\ell_2$-norm as, $d_{2,\mathbf{X}}^2(\mathbf{f}(\mathbf{h}), \mathbf{f}'(\mathbf{h}')) = \frac{1}{t \cdot n} \sum_{j=1}^{t} \sum_{i=1}^{n} (f_j(\mathbf{h}(\mathbf{x}_{ji})) - f_j'(\mathbf{h}'(\mathbf{x}_{ji})))^2$, and the corresponding $u$-covering number as $N_{2,\mathbf{X}}(u; d_{2,\mathbf{X}}, \mathcal{F}^{\otimes t}(\mathcal{H}))$. Further, we can define the *worst-case $\ell_2$-covering number* as $N_2(u; \mathcal{F}^{\otimes t}(\mathcal{H})) = \max_{\mathbf{X}} N_{2,\mathbf{X}}(u; d_{2,\mathbf{X}}, \mathcal{F}^{\otimes t}(\mathcal{H}))$. For a vector-valued function class we define the empirical $\ell_2$-norm similarly as $d_{2,\mathbf{X}}^2(\mathbf{h}, \mathbf{h}') = \frac{1}{t \cdot n} \sum_{k=1}^{r} \sum_{j=1}^{t} \sum_{i=1}^{n} (\mathbf{h}_k(\mathbf{x}_{ji}) - \mathbf{h}'_k(\mathbf{x}_{ji}))^2$.

Our goal is to bound the empirical Gaussian complexity of the set $S = \{(f_1(\mathbf{h}(\mathbf{x}_{1i})), \ldots, f_j(\mathbf{h}(\mathbf{x}_{ji})), \ldots, f_t(\mathbf{h}(\mathbf{x}_{ti}))) : j \in [t], f_j \in \mathcal{F}, \mathbf{h} \in \mathcal{H}\} \subseteq \mathbb{R}^{tn}$ or function class,

$$\hat{\mathfrak{G}}_{nt}(S) = \hat{\mathfrak{G}}_{\mathbf{X}}(\mathcal{F}^{\otimes t}(\mathcal{H})) = \frac{1}{nt} \mathbb{E}[\sup_{\mathbf{f} \in \mathcal{F}^{\otimes t}, \mathbf{h} \in \mathcal{H}} \sum_{j=1}^{t} \sum_{i=1}^{n} g_{ji} f_j(\mathbf{h}(\mathbf{x}_{ji}))]; \quad g_{ji} \sim \mathcal{N}(0,1)$$

in a manner that allows for easy application in several problems of interest. To be explicit, we also recall that,

$$\hat{\mathfrak{G}}_{\mathbf{X}}(\mathcal{H}) = \frac{1}{nt} \mathbb{E}_{\mathbf{g}}[\sup_{\mathbf{h} \in \mathcal{H}} \sum_{k=1}^{r} \sum_{j=1}^{t} \sum_{i=1}^{n} g_{kji} \mathbf{h}_k(\mathbf{x}_{ji})]; \quad g_{kji} \sim \mathcal{N}(0,1)$$

We now state the decomposition theorem for Gaussian complexity.

**Theorem 7.** *Let the function class $\mathcal{F}$ consist of functions that are $\ell_2$-Lipschitz with constant $L(\mathcal{F})$, and have boundedness parameter $D_{\mathbf{X}} = \sup_{\mathbf{f}, \mathbf{f}', \mathbf{h}, \mathbf{h}'} d_{2,\mathbf{X}}(\mathbf{f}(\mathbf{h}), \mathbf{f}'(\mathbf{h}'))$. Further, define $\mathcal{Z} = \{\mathbf{h}(\bar{\mathbf{X}}) : \mathbf{h} \in \mathcal{H}, \bar{\mathbf{X}} \in \cup_{j=1}^{t} \{\mathbf{X}_j\}\}$. Then the (empirical) Gaussian complexity of the function class $\mathcal{F}^{\otimes t}(\mathcal{H})$ satisfies,*

$$\hat{\mathfrak{G}}_{\mathbf{X}}(\mathcal{F}^{\otimes t}(\mathcal{H})) \leq \inf_{D_{\mathbf{X}} \geq \delta > 0} \left\{ 4\delta + 64 C(\mathcal{F}^{\otimes t}(\mathcal{H})) \cdot \log\left(\frac{D_{\mathbf{X}}}{\delta}\right) \right\} \leq \frac{4D_{\mathbf{X}}}{(nt)^2} + 128 C(\mathcal{F}^{\otimes t}(\mathcal{H})) \cdot \log(nt)$$

*where $C(\mathcal{F}^{\otimes t}(\mathcal{H})) = L(\mathcal{F}) \cdot \hat{\mathfrak{G}}_{\mathbf{X}}(\mathcal{H}) + \max_{\mathbf{Z} \in \mathcal{Z}} \hat{\mathfrak{G}}_{\mathbf{Z}}(\mathcal{F})$. Further, if $C(\mathcal{F}^{\otimes t}(\mathcal{H})) \leq D_{\mathbf{X}}$ then by computing the exact infima of the expression,*

$$\hat{\mathfrak{G}}_{\mathbf{X}}(\mathcal{F}^{\otimes t}(\mathcal{H})) \leq 64 \left( C(\mathcal{F}^{\otimes t}(\mathcal{H})) + C(\mathcal{F}^{\otimes t}(\mathcal{H})) \cdot \log\left(\frac{D_{\mathbf{X}}}{C(\mathcal{F}^{\otimes t}(\mathcal{H}))}\right) \right)$$

*Proof.* For ease of notation we define $N = nt$ in the following. We can rewrite the Gaussian complexity of the function class $\mathcal{F}^{\otimes t}(\mathcal{H})$ as,

$$\hat{\mathfrak{G}}_{\mathbf{X}}(\mathcal{F}^{\otimes t}(\mathcal{H})) = \mathbb{E}[\frac{1}{nt} \sup_{\mathbf{f}(\mathbf{h}) \in \mathcal{F}^{\otimes t}(\mathcal{H})} \sum_{j=1}^{t} \sum_{i=1}^{n} g_{ji} f_j(\mathbf{h}(\mathbf{x}_{ji}))] = \mathbb{E}[\frac{1}{\sqrt{N}} \cdot \sup_{\mathbf{f}(\mathbf{h}) \in \mathcal{F}^{\otimes t}(\mathcal{H})} Z_{\mathbf{f}(\mathbf{h})}]$$

from which we define the mean-zero stochastic process $Z_{\mathbf{f}(\mathbf{h})} = \frac{1}{\sqrt{N}} \sum_{j=1}^{t} \sum_{i=1}^{n} g_{ji} f_j(\mathbf{h}(\mathbf{x}_{ji}))$ for a fixed sequence of design points $\mathbf{x}_{ji}$, indexed by elements $\{\mathbf{f}(\mathbf{h}) \in \mathcal{F}^{\otimes t}(\mathcal{H})\}$, and for a sequence of independent Gaussian random variables $g_{ji}$. Note the process $Z_{\mathbf{f}(\mathbf{h})}$ has sub-gaussian increments, in the sense that, $Z_{\mathbf{f}(\mathbf{h})} - Z_{\mathbf{f}'(\mathbf{h}')}$ is a sub-gaussian random variable with parameter $d_{2,\mathbf{X}}^2(\mathbf{f}(\mathbf{h}), \mathbf{f}'(\mathbf{h}')) = \frac{1}{N} \sum_{j=1}^{t} \sum_{i=1}^{n} (f_j(\mathbf{h}(\mathbf{x}_{ji})) - f_j'(\mathbf{h}'(\mathbf{x}_{ji}))^2$. Since $Z_{\mathbf{f}(\mathbf{h})}$ is a mean-zero stochastic process we have that, $\mathbb{E}[\sup_{\mathbf{f}(\mathbf{h}) \in \mathcal{F}^{\otimes t}(\mathcal{H})} Z_{\mathbf{f}(\mathbf{h})}] = \mathbb{E}[\sup_{\mathbf{f}(\mathbf{h}) \in \mathcal{F}^{\otimes t}(\mathcal{H})} Z_{\mathbf{f}(\mathbf{h})} - Z_{\mathbf{f}'(\mathbf{h}')}] \leq \mathbb{E}[\sup_{\mathbf{f}(\mathbf{h}), \mathbf{f}'(\mathbf{h}') \in \mathcal{F}^{\otimes t}(\mathcal{H})} Z_{\mathbf{f}(\mathbf{h})} - Z_{\mathbf{f}'(\mathbf{h}')}]$. Now an appeal to the Dudley entropy integral bound, Wainwright [2019, Theorem 5.22] shows that,

$$\mathbb{E}[\sup_{\mathbf{f}(\mathbf{h}), \mathbf{f}'(\mathbf{h}') \in \mathcal{F}^{\otimes t}(\mathbf{h})} Z_{\mathbf{f}(\mathbf{h})} - Z_{\mathbf{f}(\mathbf{h}')}] \leq 4\mathbb{E}[\sup_{d_{2,\mathbf{X}}(\mathbf{f}(\mathbf{h}), \mathbf{f}'(\mathbf{h}')) \leq \delta} Z_{\mathbf{f}(\mathbf{h})} - Z_{\mathbf{f}(\mathbf{h}')}] + 32 \int_{\delta}^{D} \sqrt{\log N_{\mathbf{X}}(u; d_{2,\mathbf{X}}, \mathcal{F}^{\otimes t}(\mathcal{H}))} du.$$

We now turn to bounding each of the above terms. Parametrizing the sequence of i.i.d. gaussian variables as $\mathbf{g}$, it follows that $\sup_{d_{2,\mathbf{X}}(\mathbf{f}(\mathbf{h}), \mathbf{f}'(\mathbf{h}')) \leq \delta} Z_{\mathbf{f}(\mathbf{h})} - Z_{\mathbf{f}(\mathbf{h}')} \leq \sup_{\mathbf{v}: \|\mathbf{v}\|_2 \leq \delta} \mathbf{g} \cdot \mathbf{v} \leq \|\mathbf{g}\| \delta$. The corresponding expectation bound, after an application of Jensen's inequality to the $\sqrt{\cdot}$ function gives $\mathbb{E}[\sup_{d_{2,\mathbf{X}}(\mathbf{f}(\mathbf{h}), \mathbf{f}'(\mathbf{h}')) \leq \delta} Z_{\mathbf{f}(\mathbf{h})} - Z_{\mathbf{f}(\mathbf{h}')}] \leq \mathbb{E}[\|\mathbf{g}\|_2 \delta] \leq \sqrt{N} \delta$.

We now turn to bounding the second term by decomposing the distance metric $d_{2,\mathbf{X}}$ into a distance over $\mathcal{F}^{\otimes t}$ and a distance over $\mathcal{H}$. We then use a covering argument on each of the spaces $\mathcal{F}^{\otimes t}$ and $\mathcal{H}$ to witness a covering of the composed space $\mathcal{F}^{\otimes t}(\mathcal{H})$. Recall we refer to the entire dataset concatenated over the $t$ tasks as $\mathbf{X} \equiv \{\mathbf{x}_{ji}\}_{j=1, i=1}^{t, n}$. First, let $C_{\mathcal{H}_{\mathbf{X}}}$ be a covering of the of function space $\mathcal{H}$ in the empirical $\ell_2$-norm with respect to the inputs $\mathbf{X}$ at scale $\epsilon_1$. Then for each $\mathbf{h} \in C_{\mathcal{H}_{\mathbf{X}}}$, construct an $\epsilon_2$-covering, $C_{\mathcal{F}_{\mathbf{h}(\mathbf{X})}^{\otimes t}}$, of the function space $\mathcal{F}^{\otimes t}$ in the empirical $\ell_2$-norm with respect to the inputs $\mathbf{h}(\mathbf{X})$ at scale $\epsilon_2$. We then claim that set $C_{\mathcal{F}^{\otimes t}(\mathcal{H})} = \cup_{\mathbf{h} \in C_{\mathcal{H}_{\mathbf{X}}}} (C_{\mathcal{F}_{\mathbf{h}(\mathbf{X})}^{\otimes t}})$ is an $\epsilon_1 \cdot L(\mathcal{F}) + \epsilon_2$-cover for the function space $\mathcal{F}^{\otimes t}(\mathcal{H})$ in the empirical $\ell_2$-norm over the inputs $\mathbf{X}$. To see this, let $\mathbf{h} \in \mathcal{H}$ and $\mathbf{f} \in \mathcal{F}^{\otimes t}$ be arbitrary. Now let $\mathbf{h}' \in C_{\mathcal{H}_{\mathbf{X}}}$ be $\epsilon_1$-close to $\mathbf{h}$. Given this $\mathbf{h}'$, there exists $\mathbf{f}' \in C_{\mathcal{F}_{\mathbf{h}'(\mathbf{X})}^{\otimes t}}$ such that $\mathbf{f}'$ is $\epsilon_2$-close to $\mathbf{f}$ with respect to inputs $\mathbf{h}'(\mathbf{X})$. By construction $(\mathbf{h}', \mathbf{f}') \in C_{\mathcal{F}^{\otimes t}(\mathcal{H})}$. Finally, using the triangle inequality, we have that,

$$d_{2,\mathbf{X}}(\mathbf{f}(\mathbf{h}), \mathbf{f}'(\mathbf{h}')) \leq d_{2,\mathbf{X}}(\mathbf{f}(\mathbf{h}), \mathbf{f}(\mathbf{h}')) + d_{2,\mathbf{X}}(\mathbf{f}(\mathbf{h}'), \mathbf{f}'(\mathbf{h}')) =$$

$$\sqrt{\frac{1}{N} \sum_{j=1}^{t} \sum_{i=1}^{n} (f_j(\mathbf{h}(\mathbf{x}_{ji})) - f_j(\mathbf{h}'(\mathbf{x}_{ji})))^2} + \sqrt{\frac{1}{N} \sum_{j=1}^{t} \sum_{i=1}^{n} (f_j(\mathbf{h}'(\mathbf{x}_{ji})) - f_j'(\mathbf{h}'(\mathbf{x}_{ji})))^2} \leq$$

$$L(\mathcal{F}) \sqrt{\frac{1}{N} \sum_{k=1}^{r} \sum_{j=1}^{t} \sum_{i=1}^{n} (\mathbf{h}_k(\mathbf{x}_{ji}) - \mathbf{h}_k'(\mathbf{x}_{ji}))^2} + \sqrt{\frac{1}{N} \sum_{j=1}^{t} \sum_{i=1}^{n} (f_j(\mathbf{h}'(\mathbf{x}_{ji})) - f_j'(\mathbf{h}'(\mathbf{x}_{ji})))^2} =$$

$$L(\mathcal{F}) \cdot d_{2,\mathbf{X}}(\mathbf{h}, \mathbf{h}') + d_{2,\mathbf{h}'(\mathbf{X})}(\mathbf{f}, \mathbf{f}') \leq \epsilon_1 \cdot L(\mathcal{F}) + \epsilon_2$$

appealing to the uniform Lipschitz property of the function class $\mathcal{F}$ in moving from the second to third line, which establishes the claim.

We now bound the cardinality of the covering $C_{\mathcal{F}^{\otimes t}(\mathcal{H})}$. First, note $|C_{\mathcal{F}^{\otimes t}(\mathcal{H})}| = \sum_{\mathbf{h} \in C_{\mathcal{H}_{\mathbf{X}}}} |C_{\mathcal{F}_{\mathbf{h}(\mathbf{X})}^{\otimes t}}| \leq |C_{\mathcal{H}_{\mathbf{X}}}| \cdot \max_{\mathbf{h} \in \mathcal{H}_{\mathbf{X}}} |C_{\mathcal{F}_{\mathbf{h}(\mathbf{X})}^{\otimes t}}|$. To control $\max_{\mathbf{h} \in \mathcal{H}_{\mathbf{X}}} |C_{\mathcal{F}_{\mathbf{h}(\mathbf{X})}^{\otimes t}}|$, note an $\epsilon$-cover of $\mathcal{F}_{\mathbf{h}(\mathbf{X})}^{\otimes t}$ in the empirical $\ell_2$-norm with respect to $\mathbf{h}(\mathbf{X})$ can be obtained from the cover $C_{\mathcal{F}_{\mathbf{h}(\mathbf{X}_1)}} \times \ldots \times C_{\mathcal{F}_{\mathbf{h}(\mathbf{X}_t)}}$ where $C_{\mathcal{F}_{\mathbf{h}(\mathbf{X}_i)}}$ denotes a $\epsilon$-cover of $\mathcal{F}$ in the empirical $\ell_2$-norm with respect to $\mathbf{h}(\mathbf{X}_i)$. Hence $\max_{\mathbf{h} \in \mathcal{H}_{\mathbf{X}}} |C_{\mathcal{F}_{\mathbf{h}(\mathbf{X})}^{\otimes t}}| \leq |C_{\mathcal{F}_{\mathbf{h}(\mathbf{X}_1)}} \times \ldots \times C_{\mathcal{F}_{\mathbf{h}(\mathbf{X}_t)}}| \leq \underbrace{|\max_{\mathbf{z} \in \mathcal{Z}} C_{\mathcal{F}_{\mathbf{z}}} \times \ldots \times \max_{\mathbf{z} \in \mathcal{Z}} C_{\mathcal{F}_{\mathbf{z}}}|}_{t \text{ times}} \leq$

$|\max_{\mathbf{z} \in \mathcal{Z}} C_{\mathcal{F}_{\mathbf{z}}}|^t$. Combining these facts provides a bound on the metric entropy of,

$$\log N_{2,\mathbf{X}}(\epsilon_1 \cdot L(\mathcal{F}) + \epsilon_2, d_{2,\mathbf{X}}, \mathcal{F}^{\otimes t}(\mathcal{H})) \leq \log N_{2,\mathbf{X}}(\epsilon_1, d_{2,\mathbf{X}}, \mathcal{H}) + t \cdot \max_{\mathbf{Z} \in \mathcal{Z}} \log N_{2,\mathbf{Z}}(\epsilon_2, d_{2,\mathbf{Z}}, \mathcal{F}).$$

Using the covering number upper bound with $\epsilon_1 = \frac{\epsilon}{2 \cdot L(\mathcal{F})}$, $\epsilon_2 = \frac{\epsilon}{2}$ and sub-additivity of the $\sqrt{\cdot}$ function then gives a bound on the entropy integral of,

$$\int_\delta^D \sqrt{\log N_2(\epsilon, d_{2,\mathbf{X}}, \mathcal{F}^{\otimes t}(\mathcal{H}))}\, d\epsilon \leq \int_\delta^D \sqrt{\log N_{2,\mathbf{X}}(\epsilon/(2L(\mathcal{F})), d_{2,\mathbf{X}}, \mathcal{H})}\, d\epsilon + \sqrt{t}\int_\delta^D \max_{\mathbf{Z}\in\mathcal{Z}} \sqrt{\log N_{2,\mathbf{z}}(\tfrac{\epsilon}{2}, d_{2,\mathbf{z}}, \mathcal{F})}\, d\epsilon$$

From the Sudakov minoration theorem Wainwright [2019][Theorem 5.30] for Gaussian processes and the fact packing numbers at scale $u$ upper bounds the covering number at scale $u$ we find:

$$\log N_{2,\mathbf{X}}(u; d_{2,\mathbf{X}}, \mathcal{H}) \leq 4\left(\frac{\sqrt{nt}\hat{\mathfrak{G}}_\mathbf{X}(\mathcal{H})}{u}\right)^2 \ \forall u > 0 \quad \text{and} \quad \log N_{2,\mathbf{Z}}(u; d_{2,\mathbf{z}}, \mathcal{F}) \leq 4\left(\frac{\sqrt{n}\hat{\mathfrak{G}}_\mathbf{Z}(\mathcal{F})}{u}\right)^2 \ \forall u > 0.$$

For the $\mathcal{H}$ term we apply the result to the mean-zero Gaussian process $Z_\mathbf{h} = \frac{1}{\sqrt{nt}}\sum_{k=1}^r \sum_{j=1}^t \sum_{i=1}^n g_{kji} h_k(\mathbf{x}_{ji})$, for $g_{kji} \sim \mathcal{N}(0,1)$ i.i.d. and $\mathbf{h} \in \mathcal{H}$. Combining all of the aforementioned upper bounds, shows that

$$\hat{\mathfrak{G}}_\mathbf{X}(\mathcal{F}^{\otimes t}(\mathcal{H})) \leq \frac{1}{\sqrt{nt}}\left(4\delta\sqrt{nt} + 64 L(\mathcal{F}) \cdot \hat{\mathfrak{G}}_\mathbf{X}(\mathcal{H}) \cdot \sqrt{nt}\int_\delta^{D_\mathbf{X}} \frac{1}{u}du + 64\sqrt{nt}\cdot \max_{\mathbf{Z}\in\mathcal{Z}}\hat{\mathfrak{G}}_\mathbf{Z}(\mathcal{F})\int_\delta^{D_\mathbf{X}}\frac{1}{u}du\right) \leq$$

$$4\delta + 64(L(\mathcal{F})\cdot\hat{\mathfrak{G}}_\mathbf{X}(\mathcal{H}) + \max_{\mathbf{Z}\in\mathcal{Z}}\hat{\mathfrak{G}}_\mathbf{Z}(\mathcal{F}))\cdot\log\left(\frac{D_\mathbf{X}}{\delta}\right) = \delta + C(\mathcal{F}^{\otimes t}(\mathcal{H}))\cdot\log\left(\frac{D_\mathbf{X}}{\delta}\right)$$

defining $C(\mathcal{F}^{\otimes t}(\mathcal{H})) = L(\mathcal{F})\cdot\hat{\mathfrak{G}}_\mathbf{X}(\mathcal{H}) + \max_{\mathbf{Z}\in\mathcal{Z}}\hat{\mathfrak{G}}_\mathbf{Z}(\mathcal{F})$. Choosing $\delta = D_\mathbf{X}/(nt)^2$ gives the first inequality. Balancing the first and second term gives the optimal choice $\delta = \frac{1}{C(\mathcal{F}^{\otimes t}(\mathcal{H}))}$ for the second inequality under the stated conditions. $\qquad\square$

## C  Proofs in Section 4

In this section we instantiate our general framework in several concrete examples. This consists of two steps: first verifying a task diversity lower bound for the function classes and losses and then bounding the various complexity terms appearing in the end-to-end LTL guarantee in Theorem 3 or its variants.

### C.1  Logistic Regression

Here we include the proofs of the results which both bound the complexities of the function classes $\mathcal{F}$ and $\mathcal{H}$ in the logistic regression example as well establish the task diversity lower bound in this setting. In this section we use the following definition,

**Definition 4.** We say the covariate distribution $\mathbb{P}_\mathbf{x}(\cdot)$ is $\mathbf{\Sigma}$-sub-gaussian if for all $\mathbf{v}\in\mathbb{R}^d$, $\mathbb{E}[\exp(\mathbf{v}^\top \mathbf{x}_i)] \leq \exp\left(\frac{\|\mathbf{\Sigma}^{1/2}\mathbf{v}\|^2}{2}\right)$ where the covariance $\mathbf{\Sigma}$ further satisfies $\sigma_{\max}(\mathbf{\Sigma}) \leq C$ and $\sigma_{\min}(\mathbf{\Sigma}) \geq c > 0$ for universal constants $c, C$.

We begin by presenting the proof of the Theorem 4 which essentially relies on instantiating a variant of Theorem 3. In order to obtain a sharper dependence in the noise terms in the test learning stage we actually directly combine Theorem 1 and Corollary 2.

Since we are also interested in stating data-dependent guarantees in this section we use the notation $\mathbf{\Sigma}_\mathbf{X} = \frac{1}{nt}\sum_{j=1}^t \sum_{i=1}^n \mathbf{x}_{ji}\mathbf{x}_{ji}^\top$ to refer to the empirical covariance across the the training phase samples and $\mathbf{\Sigma}_{\mathbf{X}_j}$ for corresponding empirical covariances across the per-task samples. Immediately following this result we present the statement of sharp data-dependent guarantee which depends on these empirical quantities for completeness.

*Proof of Theorem 4.* First note due to the task normalization conditions we can choose $c_1, c_2$ sufficiently large so that the realizability assumption in Assumption 2 is satisfied–in particular, we can assume that $c_2$ is chosen large enough to contain all the parameters $\boldsymbol{\alpha}_j^\star$ for $j \in [t]\cup\{0\}$ and $c_1 \geq \frac{C}{c}c_2$. Next note that under the conditions of the result we can use Lemma 1 to verify the task diversity condition is satisfied with parameters $(\tilde{\nu}, 0)$ with $\nu = \sigma_r(\mathbf{A}^\top \mathbf{A}/t) > 0$ with this choice of constants.

Finally, in order to combine Theorem 1 and Corollary 2 we begin by bounding each of the complexity terms in the expression. First,

- For the feature learning complexity in the training phase we obtain,

$$\hat{\mathfrak{S}}_{\mathbf{X}}(\mathcal{H}) = \frac{1}{nt}\mathbb{E}[\sup_{\mathbf{B}\in\mathcal{H}}\sum_{k=1}^{r}\sum_{j=1}^{t}\sum_{i=1}^{n}g_{kji}\mathbf{b}_k^\top\mathbf{x}_{ji}] = \frac{1}{nt}\mathbb{E}[\sup_{(\mathbf{b}_1,...,\mathbf{b}_r)\in\mathcal{H}}\sum_{k=1}^{r}\mathbf{b}_k^\top(\sum_{j=1}^{t}\sum_{i=1}^{n}g_{kji}\mathbf{x}_{ji})] \leq$$

$$\frac{1}{nt}\sum_{k=1}^{r}\mathbb{E}[\|\sum_{j=1}^{t}\sum_{i=1}^{n}g_{kji}\mathbf{x}_{ji}\|] \leq \frac{1}{nt}\sum_{k=1}^{r}\sqrt{\mathbb{E}[\|\sum_{j=1}^{t}\sum_{i=1}^{n}g_{kji}\mathbf{x}_{ji}\|^2]} \leq \frac{1}{nt}\sum_{k=1}^{r}\sqrt{\sum_{j=1}^{t}\sum_{i=1}^{n}\|\mathbf{x}_{ji}\|^2}$$

$$= \frac{r}{\sqrt{nt}}\sqrt{\mathrm{tr}(\mathbf{\Sigma_X})}.$$

Further by definition the class $\mathcal{F}$ as linear maps with parameters $\|\boldsymbol{\alpha}\|_2 \leq O(1)$ we obtain that $L(\mathcal{F}) = O(1)$. We now proceed to convert this to a population quantity by noting that $\mathbb{E}[\sqrt{\mathrm{tr}(\mathbf{\Sigma_X})}] \leq \sqrt{d\cdot\mathbb{E}[\|\mathbf{\Sigma_X}\|]} \leq O(\sqrt{d})$ for $nt \gtrsim d$ by Lemma 4.

- For the complexity of learning $\mathcal{F}$ in the training phase we obtain,

$$\bar{\mathfrak{S}}_n(\mathcal{F}) = \frac{1}{n}\mathbb{E}[\sup_{\|\boldsymbol{\alpha}\|\leq c_1}\sum_{i=1}^{n}g_i\boldsymbol{\alpha}^\top\mathbf{B}^\top\mathbf{x}_{ji}] = \frac{c_1}{n}\mathbb{E}[\|\sum_{i=1}^{n}g_i\mathbf{B}^\top\mathbf{x}_{ji}\|] \leq \frac{c_1}{n}\sqrt{\sum_{i=1}^{n}\|\mathbf{B}^\top\mathbf{x}_{ji}\|^2} =$$

$$\frac{c_1}{\sqrt{n}}\sqrt{\mathrm{tr}(\mathbf{B}\mathbf{B}^\top\mathbf{\Sigma}_{\mathbf{X}_j})} = \frac{c_1}{\sqrt{n}}\sqrt{\mathrm{tr}(\mathbf{B}^\top\mathbf{\Sigma}_{\mathbf{X}_j}\mathbf{B})}.$$

Now by the variational characterization of singular values it follows that $\max_{\mathbf{B}\in\mathcal{H}}\frac{c_1}{\sqrt{n}}\sqrt{\mathrm{tr}(\mathbf{B}^\top\mathbf{\Sigma}_{\mathbf{X}_j}\mathbf{B})} \leq \frac{c_1}{n}\sqrt{\sum_{i=1}^{r}\sigma_i(\mathbf{\Sigma}_{\mathbf{X}_j})}$ Thus it immediately follows that,

$$\max_{\mathbf{Z}\in\mathcal{Z}}\frac{c_1}{\sqrt{n}}\sqrt{\mathrm{tr}(\mathbf{\Sigma}_{\mathbf{X}_j})} = \max_{\mathbf{X}_j}\max_{\mathbf{B}\in\mathcal{H}}\frac{c_1}{\sqrt{n}}\sqrt{\mathrm{tr}(\mathbf{B}^\top\mathbf{\Sigma}_{\mathbf{X}_j}\mathbf{B})} \leq \max_{\mathbf{X}_j}\frac{c_1}{\sqrt{n}}\sqrt{\sum_{i=1}^{r}\sigma_i(\mathbf{\Sigma}_{\mathbf{X}_j})}.$$

for $j \in [t]$. We can convert this to a population quantity again by applying Lemma 4 which shows $\mathbb{E}[\sqrt{\sum_{i=1}^{r}\sigma_i(\mathbf{\Sigma}_{\mathbf{X}_j})}] \leq O(\sqrt{r})$ for $n \gtrsim d + \log t$.

- A nearly identical argument shows the complexity of learning $\mathcal{F}$ in the testing phase is,

$$\hat{\mathfrak{S}}_{\mathbf{Y}_{\hat{\mathbf{h}}}}(\mathcal{F}) = \frac{1}{m}\mathbb{E}[\sup_{\|\boldsymbol{\alpha}\|\leq c_1}\sum_{i=1}^{m}\epsilon_i\boldsymbol{\alpha}^\top\hat{\mathbf{B}}^\top\mathbf{x}_{(0)i}] \leq \frac{c_1}{\sqrt{m}}\sqrt{\sum_{i=1}^{r}\sigma_i(\hat{\mathbf{B}}^\top\mathbf{\Sigma}_{\mathbf{X}_0}\hat{\mathbf{B}})}$$

Crucially, here we can apply the first result in Corollary 2 which allows us to take the expectation over $\mathbf{X}_0$ before maximizing over $\mathbf{B}$. Thus applying Lemma 4 as before gives the result, $\mathbb{E}[\sqrt{\sum_{i=1}^{r}\sigma_i(\mathbf{B}^\top\mathbf{\Sigma}_{\mathbf{X}_0}\mathbf{B})}] \leq O(\sqrt{r})$ for $m \gtrsim r$.

This gives the first series of claims.

Finally we verify that Assumption 1 holds so as to use Theorem 1 and Corollary 2 to instantiate the end-to-end guarantee. First the boundedness parameter becomes,

$$D_{\mathcal{X}} = \sup_{\boldsymbol{\alpha},\mathbf{B}}(\mathbf{x}^\top\mathbf{B}\boldsymbol{\alpha}) \leq O(D)$$

using the assumptions that $\|\mathbf{x}\|_2 \leq D$, $\|\boldsymbol{\alpha}\|_2 \leq O(1)$, $\|\mathbf{B}\|_2 = 1$. For the logistic loss bounds, $\ell(\eta; y) = y\eta - \log(1 + \exp(\eta))$. Since $|\nabla_\eta\ell(\eta; y)| = |y - \frac{\exp(\eta)}{1+\exp(\eta)}| \leq 1$ it is $O(1)$-Lipschitz in its first coordinate uniformly over its second so $L = O(1)$. Moreover as, $|\ell(\eta; y)| \leq O(\eta)$ where $\eta = \mathbf{x}^\top\mathbf{B}\boldsymbol{\alpha} \leq \|\mathbf{x}\| \leq D$ it follows the loss is uniformly bounded with parameter $O(D)$ so $B = O(D)$.

Lastly, to use Corollary 2 to bound the test phase error we need to compute the maximal variance term $\sigma^2 = \frac{1}{m}\sup_{f\in\mathcal{F}}\sum_{i=1}^{m}\mathrm{Var}(\ell(f\circ\hat{\mathbf{h}}(\mathbf{x}_i),y_i))$. Since for the logistic loss $\ell(\cdot,\cdot)$ satisfies the 1-Lipschitz property uniformly we have that, $\mathrm{Var}(\ell(f\circ\hat{\mathbf{h}}(\mathbf{x}_i),y_i))\leq\mathrm{Var}(f\circ\hat{\mathbf{h}}(\mathbf{x}_i))$ for each $i\in[m]$. Collapsing the variance we have that,

$$\frac{1}{m}\sup_{\boldsymbol{\alpha}:\|\boldsymbol{\alpha}\|_2\leq O(1)}\sum_{i=1}^{m}\mathrm{Var}(\mathbf{x}_i^\top\hat{\mathbf{B}}\boldsymbol{\alpha})\leq\frac{1}{m}\sup_{\boldsymbol{\alpha}:\|\boldsymbol{\alpha}\|_2\leq O(1)}\sum_{i=1}^{m}(\boldsymbol{\alpha}\hat{\mathbf{B}})^\top\boldsymbol{\Sigma}\hat{\mathbf{B}}\boldsymbol{\alpha}\leq O(\|\hat{\mathbf{B}}\boldsymbol{\Sigma}\hat{\mathbf{B}}\|_2)\leq$$
$$O(\|\boldsymbol{\Sigma}\|)\leq O(C)=O(1)$$

under our assumptions, which implies that $\sigma\leq O(1)$. Assembling the previous bounds shows the transfer learning risk is bounded by,

$$\lesssim\frac{1}{\tilde{\nu}}\cdot\left(\log(nt)\cdot\left[\sqrt{\frac{dr^2}{nt}}+\sqrt{\frac{r}{n}}\right]\right)+\sqrt{\frac{r}{m}}$$
$$+\left(\frac{D}{\tilde{\nu}}\cdot\max\left(\frac{1}{(nt)^2},\sqrt{\frac{\log(2/\delta)}{nt}}\right)+\sqrt{\frac{\log(2/\delta)}{m}}+D\frac{\log(2/\delta)}{m}\right).$$

with probability at least $1-2\delta$. Suppressing all logarithmic factors and using the additional condition $D\lesssim\min(dr^2,\sqrt{rm})$ guarantees the noise terms are higher-order. $\qquad\square$

Recall, in the context of the two-stage ERM procedure introduced in Section 2 we let the design matrix and responses $y_{ji}$ for the $j$th task be $\mathbf{X}_j$ and $\mathbf{y}_j$ for $j\in[t]\cup\{0\}$, and the entire design matrix and responses concatenated over all $j\in[t]$ tasks as $\bar{\mathbf{X}}$ and $\mathbf{y}$ respectively. $\boldsymbol{\Sigma}_{\bar{\mathbf{X}}}$ denotes the empirical covariance of the design matrix $\bar{\mathbf{X}}$. We now state a sharp, data-dependent guarantee for logistic regression.

**Corollary 3.** *If Assumption 3 holds, $\mathbf{h}^\star(\cdot)\in\mathcal{H}$, and $\mathcal{F}_0=\{\,f\mid f(\mathbf{x})=\boldsymbol{\alpha}^\top\mathbf{z},\ \boldsymbol{\alpha}\in\mathbb{R}^r,\ \|\boldsymbol{\alpha}\|\leq c_2\}$, then there exist constants $c_1,c_2$ such that the training tasks $f_j^\star$ are $(\Omega(\tilde{\nu}),0)$-diverse over $\mathcal{F}_0$. Then with probability at least $1-2\delta$:*

*Transfer Learning Risk* $\leq$

$$O\Big(\frac{1}{\tilde{\nu}}\cdot\Big(\log(nt)\cdot\left[\sqrt{\frac{\mathrm{tr}(\boldsymbol{\Sigma}_{\mathbf{X}})r^2}{nt}}+\max_{j\in[t]}\sqrt{\frac{\sum_{i=1}^{r}\sigma_i(\mathbf{X}_j)}{n}}\right]\Big)+\sqrt{\frac{\sum_{i=1}^{r}\sigma_i(\mathbf{X}_0)}{m}}\Big)$$
$$+O\Big(\frac{D}{\tilde{\nu}}\cdot\max\left(\frac{1}{(nt)^2},\sqrt{\frac{\log(4/\delta)}{nt}}\right)+\sqrt{\frac{\log(4/\delta)}{m}}+D\frac{\log(4/\delta)}{m}\Big).$$

*Proof of Corollary 3.* This follows immediately from the proof of Theorem 4 and applying Corollary 1, Corollary 2. Merging terms and applying a union bound gives the result. $\qquad\square$

The principal remaining challenge is to obtain a sharp lower bound on the task diversity.

**Lemma 1.** *Let Assumption 3 hold in the setting of Theorem 4. Then there exists $c_2$ such that if $c_1\geq\frac{C}{c}c_2$ the problem is task-diverse with parameter $(\Omega(\tilde{\nu}),0)$ in the sense of Definition 3 where $\tilde{\nu}=\sigma_r(\mathbf{A}^\top\mathbf{A}/t)$.*

*Proof.* Our first observation specializes Lemma 2 to the case of logistic regression where $\Phi(\eta)=\log(1+\exp(\eta))$, $s(\sigma)=1$ with $\mathbf{h}(\mathbf{x})=\mathbf{B}\mathbf{x}$ parametrized with $\mathbf{B}\in\mathbb{R}^{d\times r}$ having orthonormal columns and $\mathbf{f}\equiv\boldsymbol{\alpha}$. Throughout we also assume that $c_2$ is chosen large enough to contain all the parameters $\boldsymbol{\alpha}_j^\star$ for $j\in[t]\cup\{0\}$ and $c_1\geq\frac{C}{c}c_2$. These conditions immediately imply the realizability conditions.

This lemma essentially allows us to use smoothness and strong convexity to bound the task-averaged representation distance and worst-case representation difference. By appealing to Lemma 2 and Lemma 3 we have that,

$$\frac{1}{8}\mathbb{E}_{\mathbf{x}_i}[\exp(-\max(|\hat{\mathbf{h}}(\mathbf{x}_i)^\top\hat{\boldsymbol{\alpha}}|,|\mathbf{h}(\mathbf{x}_i)^\top\boldsymbol{\alpha}|))\cdot(\hat{\mathbf{h}}(\mathbf{x}_i)^\top\hat{\boldsymbol{\alpha}}-\mathbf{h}(\mathbf{x}_i)^\top\boldsymbol{\alpha})^2]\leq$$

$$\mathbb{E}_{\mathbf{x}_i,y}[\ell(\hat{f}\circ\hat{\mathbf{h}}(\mathbf{x}_i),y_i)-\ell(f\circ\mathbf{h}(\mathbf{x}_i),y_i)]\leq\frac{1}{8}\mathbb{E}_{\mathbf{x}_i}[(\hat{\mathbf{h}}(\mathbf{x}_i)^\top\hat{\boldsymbol{\alpha}}-\mathbf{h}(\mathbf{x}_i)^\top\boldsymbol{\alpha})^2]$$

We now bound each term in the task diversity,

- We first bound the representation difference where $\mathbf{x}_i, y_i \sim (\mathbb{P}_{\mathbf{x}}(\cdot), \mathbb{P}_{y|\mathbf{x}}(\cdot | f_0^\star \circ \mathbf{h}^\star(\mathbf{x})))$,

$$d_{\mathcal{F}, \mathcal{F}_0}(\hat{\mathbf{h}}; \mathbf{h}^\star) = \sup_{\boldsymbol{\alpha}: \|\boldsymbol{\alpha}\|_2 \leq c_2} \inf_{\hat{\boldsymbol{\alpha}}: \|\hat{\boldsymbol{\alpha}}\| \leq c_1} \mathbb{E}_{\mathbf{x}, y}[\ell(\hat{f} \circ \hat{\mathbf{h}}, \mathbf{x}, y) - \ell(f_0^\star \circ \mathbf{h}^\star(\mathbf{x}), y)]] \leq$$

$$\sup_{\boldsymbol{\alpha}: \|\boldsymbol{\alpha}\|_2 \leq c_2} \inf_{\hat{\boldsymbol{\alpha}}: \|\hat{\boldsymbol{\alpha}}\| \leq c_1} \frac{1}{8} \mathbb{E}_{\mathbf{x}}[(\hat{\mathbf{h}}(\mathbf{x})^\top \hat{\boldsymbol{\alpha}} - \mathbf{h}^\star(\mathbf{x})^\top \boldsymbol{\alpha})^2].$$

Now for sufficiently large $c_1$, by Lagrangian duality the unconstrained minimizer of the inner optimization problem is equivalent to the constrained minimizer. In particular first note that under the assumptions of the problem there is unique unconstrained minimizer given by $\inf_{\hat{\boldsymbol{\alpha}}} \frac{1}{8} \mathbb{E}_{\mathbf{x}_i}[(\hat{\mathbf{h}}(\mathbf{x}_i)^\top \hat{\boldsymbol{\alpha}} - \mathbf{h}^\star(\mathbf{x}_i)^\top \boldsymbol{\alpha})^2] \to \hat{\boldsymbol{\alpha}}_{unconstrained} = -\mathbf{F}_{\hat{\mathbf{h}}\hat{\mathbf{h}}} \mathbf{F}_{\hat{\mathbf{h}}\mathbf{h}} \boldsymbol{\alpha} = (\hat{\mathbf{B}}^\top \boldsymbol{\Sigma} \hat{\mathbf{B}})^{-1}(\hat{\mathbf{B}}^\top \boldsymbol{\Sigma} \hat{\mathbf{B}}) \boldsymbol{\alpha}$ from the proof and preamble of Lemma 6. Note that since $\hat{\mathbf{B}}$ and $\mathbf{B}$ have orthonormal columns it follows that $\|\hat{\boldsymbol{\alpha}}\| \leq \frac{C}{c} c_2$ since $\hat{\mathbf{B}}^\top \boldsymbol{\Sigma} \hat{\mathbf{B}}$ is invertible. Thus if $c_1 \geq \frac{C}{c} c_2$, by appealing to Lagrangian duality for this convex quadratic objective with convex quadratic constraint, the unconstrained minimizer is equivalent to the constrained minimizer (since the unconstrained minimizer is contained in the constraint set). Hence leveraging the proof and result of Lemma 6 we obtain $\sup_{\boldsymbol{\alpha}: \|\boldsymbol{\alpha}\|_2 \leq c_2} \inf_{\hat{\boldsymbol{\alpha}}: \|\hat{\boldsymbol{\alpha}}\| \leq c_1} \frac{1}{8} \mathbb{E}_{\mathbf{x}_i}[(\hat{\mathbf{h}}(\mathbf{x}_i)^\top \hat{\boldsymbol{\alpha}} - \mathbf{h}^\star(\mathbf{x}_i)^\top \boldsymbol{\alpha})^2] \leq \frac{c_2}{8} \sigma_{\max}(\Lambda_{sc}(\mathbf{h}, \hat{\mathbf{h}}))$.

- We now turn out attention to controlling the average distance which we must lower bound. Here $\mathbf{x}_i, y_i \sim (\mathbb{P}_{\mathbf{x}}(\cdot), \mathbb{P}_{y|\mathbf{x}}(\cdot | f_j^\star \circ \mathbf{h}^\star(\mathbf{x}))$

$$\bar{d}_{\mathcal{F}, \mathbf{f}^\star}(\mathbf{h}; \hat{\mathbf{h}}) = \frac{1}{t} \sum_{j=1}^t \inf_{\|\hat{\boldsymbol{\alpha}}\| \leq c_1} \mathbb{E}_{\mathbf{x}, y}[\ell(\hat{f} \circ \hat{\mathbf{h}}(\mathbf{x}_i), y) - \ell(f_j^\star \circ \mathbf{h}^\star(\mathbf{x}_i), y)]] \geq$$

$$\frac{1}{8t} \sum_{j=1}^t \mathbb{E}_{\mathbf{x}}[\exp(-\max(|\hat{\mathbf{h}}(\mathbf{x}_i)^\top \hat{\boldsymbol{\alpha}}|, |\mathbf{h}^\star(\mathbf{x}_i)^\top \boldsymbol{\alpha}_j^\star|)) \cdot (\hat{\mathbf{h}}(\mathbf{x}_i)^\top \hat{\boldsymbol{\alpha}} - \mathbf{h}^\star(\mathbf{x}_i)^\top \boldsymbol{\alpha}_j^\star)^2]$$

We will use the fact that in logistic regression $\mathbf{h}(\mathbf{x}_i) = \mathbf{B}\mathbf{x}_i$; in this case if $\mathbf{x}_i$ is $C$-subgaussian random vector in $d$ dimensions, then $\mathbf{B}\mathbf{x}_i$ is $C$-subgaussian random vector in $r$ dimensions. We lower bound each term in the sum over $j$ identically and suppress the $j$ for ease of notation in the following. For fixed $j$, note the random variables $Z_1 = (\boldsymbol{\alpha}_j^\star)^\top \mathbf{B}\mathbf{x}_i$ and $Z_2 = \hat{\boldsymbol{\alpha}}^\top \hat{\mathbf{B}}\mathbf{x}_i$ are subgaussian with variance parameter at most $\|\boldsymbol{\alpha}_j^\star\|_2^2 C^2$ and $\|\hat{\boldsymbol{\alpha}}\|_2^2 C^2$ respectively. Define the event $\mathbb{1}[E] = \mathbb{1}[|Z_1| \leq Ck\|\boldsymbol{\alpha}_j^\star\| \cap \mathbb{1}\{|Z_2| \leq Ck\|\hat{\boldsymbol{\alpha}}\|]$ for $k$ to be chosen later. We use this event to lower bound the averaged task diversity since it is a non-negative random variable,

$$\mathbb{E}_{\mathbf{x}}[\exp(-\max(|\hat{\mathbf{h}}(\mathbf{x}_i)^\top \hat{\boldsymbol{\alpha}}|, |\mathbf{h}^\star(\mathbf{x}_i)^\top \boldsymbol{\alpha}_j^\star|)) \cdot (\hat{\mathbf{h}}(\mathbf{x}_i)^\top \hat{\boldsymbol{\alpha}} - \mathbf{h}^\star(\mathbf{x}_i)^\top \boldsymbol{\alpha}_j^\star)^2] \geq$$

$$\mathbb{E}_{\mathbf{x}}[\mathbb{1}[E] \exp(-\max(|\hat{\mathbf{h}}(\mathbf{x}_i)^\top \hat{\boldsymbol{\alpha}}|, |\mathbf{h}^\star(\mathbf{x}_i)^\top \boldsymbol{\alpha}_j^\star|)) \cdot (\hat{\mathbf{h}}(\mathbf{x}_i)^\top \hat{\boldsymbol{\alpha}} - \mathbf{h}^\star(\mathbf{x}_i)^\top \boldsymbol{\alpha}_j^\star)^2] \geq$$

$$\exp(-Ck \max(c_1, c_2)) \cdot \mathbb{E}_{\mathbf{x}}[\mathbb{1}[E](\hat{\mathbf{h}}(\mathbf{x}_i)^\top \hat{\boldsymbol{\alpha}} - \mathbf{h}^\star(\mathbf{x}_i)^\top \boldsymbol{\alpha}_j^\star)^2]$$

We now show that for appropriate choice of $k$, $\mathbb{E}_{\mathbf{x}}[\mathbb{1}[E](\hat{\mathbf{h}}(\mathbf{x}_i)^\top \hat{\boldsymbol{\alpha}} - \mathbf{h}^\star(\mathbf{x}_i)^\top \boldsymbol{\alpha}_j^\star)^2]$ is lower bounded by $\mathbb{E}_{\mathbf{x}}[(\hat{\mathbf{h}}(\mathbf{x}_i)^\top \hat{\boldsymbol{\alpha}} - \mathbf{h}^\star(\mathbf{x}_i)^\top \boldsymbol{\alpha}_j^\star)^2]$ modulo a constant factor. First write $\mathbb{E}_{\mathbf{x}}[\mathbb{1}[E](\hat{\mathbf{h}}(\mathbf{x}_i)^\top \hat{\boldsymbol{\alpha}} - \mathbf{h}^\star(\mathbf{x}_i)^\top \boldsymbol{\alpha}_j^\star)^2] = \mathbb{E}_{\mathbf{x}}[(\hat{\mathbf{h}}(\mathbf{x}_i)^\top \hat{\boldsymbol{\alpha}} - \mathbf{h}^\star(\mathbf{x}_i)^\top \boldsymbol{\alpha}_j^\star)^2] - \mathbb{E}_{\mathbf{x}}[\mathbb{1}[E^c](\hat{\mathbf{h}}(\mathbf{x}_i)^\top \hat{\boldsymbol{\alpha}} - \mathbf{h}^\star(\mathbf{x}_i)^\top \boldsymbol{\alpha}_j^\star)^2]$.

We upper bound the second term first using Cauchy-Schwarz,

$$\mathbb{E}_{\mathbf{x}}[\mathbb{1}[E^c](\hat{\mathbf{h}}(\mathbf{x}_i)^\top \hat{\boldsymbol{\alpha}} - \mathbf{h}^\star(\mathbf{x}_i)^\top \boldsymbol{\alpha}_j^\star)^2] \leq \sqrt{\mathbb{P}[E^c]} \sqrt{\mathbb{E}_{\mathbf{x}}(\hat{\mathbf{h}}(\mathbf{x}_i)^\top \hat{\boldsymbol{\alpha}} - \mathbf{h}^\star(\mathbf{x}_i)^\top \boldsymbol{\alpha}_j^\star)^4}$$

Define $Z_3 = \mathbf{x}_i^\top((\mathbf{B}^\star)^\top \boldsymbol{\alpha}_j^\star - \hat{\mathbf{B}}^\top \hat{\boldsymbol{\alpha}})$ which by definition is subgaussian with parameter at most $((\mathbf{B}^\star)^\top \boldsymbol{\alpha}_j^\star - \hat{\mathbf{B}}^\top \hat{\boldsymbol{\alpha}})\boldsymbol{\Sigma}((\mathbf{B}^\star)^\top \boldsymbol{\alpha}_j^\star - \hat{\mathbf{B}}^\top \hat{\boldsymbol{\alpha}}) = \sigma^2$; since this condition implies L4-L2 hypercontractivity (see for example Wainwright [2019, Theorem 2.6]) we can also conclude that,

$$\sqrt{\mathbb{E}_{\mathbf{x}}(\hat{\mathbf{h}}(\mathbf{x}_i)^\top \hat{\boldsymbol{\alpha}} - \mathbf{h}^\star(\mathbf{x}_i)^\top \boldsymbol{\alpha}_j^\star)^4} \leq 10\sigma^2 = 10 \cdot \mathbb{E}_{\mathbf{x}}(\hat{\mathbf{h}}(\mathbf{x}_i)^\top \hat{\boldsymbol{\alpha}} - \mathbf{h}^\star(\mathbf{x}_i)^\top \boldsymbol{\alpha}_j^\star)^2.$$

Recalling the subgaussianity of $Z_1$ and $Z_2$, from an application of Markov and Jensen's inequality,

$$\mathbb{P}[|Z_1| \geq k \cdot C \|\boldsymbol{\alpha}_j^\star\|_2] \leq \frac{\mathbb{E}[Z^2]}{k^2 \cdot C^2 \|\boldsymbol{\alpha}_j^\star\|_2} \leq \frac{1}{k^2}$$

with an identical statement true for $Z_2$. Finally by using a union bound we have that $\sqrt{\mathbb{P}[E^c]} \leq \frac{\sqrt{2}}{k}$ using these probability bounds. Hence by taking $k = 30$ we can ensure that $\mathbb{E}_{\mathbf{x}}[\mathbb{1}[E](\hat{\mathbf{h}}(\mathbf{x}_i)^\top \hat{\boldsymbol{\alpha}} - \mathbf{h}^\star(\mathbf{x}_i)^\top \boldsymbol{\alpha}_j^\star)^2] \geq \frac{1}{2}\mathbb{E}_{\mathbf{x}}[(\hat{\mathbf{h}}(\mathbf{x}_i)^\top \hat{\boldsymbol{\alpha}} - \mathbf{h}^\star(\mathbf{x}_i)^\top \boldsymbol{\alpha}_j^\star)^2]$ by assembling the previous bounds. Finally since $c_1, c_2, C, k$ are universal constants, by definition the conclusion that,

$$\mathbb{E}_{\mathbf{x}}[\exp(-\max(|\hat{\mathbf{h}}(\mathbf{x}_i)^\top \hat{\boldsymbol{\alpha}}|, |\mathbf{h}^\star(\mathbf{x}_i)^\top \boldsymbol{\alpha}_j^\star|)) \cdot (\hat{\mathbf{h}}(\mathbf{x}_i)^\top \hat{\boldsymbol{\alpha}} - \mathbf{h}^\star(\mathbf{x}_i)^\top \boldsymbol{\alpha}_j^\star)^2] \geq$$
$$\Omega(\mathbb{E}_{\mathbf{x}}(\hat{\mathbf{h}}(\mathbf{x}_i)^\top \hat{\boldsymbol{\alpha}} - \mathbf{h}^\star(\mathbf{x}_i)^\top \boldsymbol{\alpha}_j^\star)^2)$$

follows for each $j$. Hence the average over the $t$ tasks is identically lower bounded as,

$$\Omega\left(\frac{1}{t}\sum_{j=1}^{t} \mathbb{E}_{\mathbf{x}}(\hat{\mathbf{h}}(\mathbf{x}_i)^\top \hat{\boldsymbol{\alpha}} - \mathbf{h}^\star(\mathbf{x}_i)^\top \boldsymbol{\alpha}_j^\star)^2\right)$$

Now using the argument from the upper bound to compute the infima since all the $\|\boldsymbol{\alpha}_j^\star\| \leq c_2$ (and hence the constrained minimizers identical to the unconstrained minimizers for each of the $j$ terms for $c_1 \geq \frac{C}{c}c_2$) and using the proof of Lemma 6 we conclude that,

$$\bar{d}_{\mathcal{F},\mathbf{f}^\star}(\hat{\mathbf{h}}; \mathbf{h}^\star) \geq \Omega(\mathrm{tr}(\Lambda_{sc}(\mathbf{h}^\star, \hat{\mathbf{h}})\mathbf{C})).$$

Combining these upper and lower bounds and concluding as in the proof of Lemma 6 shows

$$d_{\mathcal{F},\mathcal{F}_0}(\hat{\mathbf{h}}; \mathbf{h}^\star) \leq \frac{1}{\Omega(\tilde{\nu})}\bar{d}_{\mathcal{F},\mathbf{f}^\star}(\hat{\mathbf{h}}; \mathbf{h}^\star)$$

$\square$

Before showing the convexity-based lemmas used to control the representation differences in the loss we make a brief remark to interpret the logistic loss in the well-specified model.

**Remark 1.** If the data generating model satisfies the logistic model conditional likelihood as in Section 4.1, for the logistic loss $\ell$ we have that,

$$\mathbb{E}_{y \sim f \circ \mathbf{h}(\mathbf{x})}[\ell(\hat{f} \circ \hat{\mathbf{h}}(\mathbf{x}), y) - \ell(f \circ \mathbf{h}(\mathbf{x}), y)]] = \mathbb{E}_{\mathbf{x}}[\mathrm{KL}[\mathrm{Bern}(\sigma(f \circ \mathbf{h}(\mathbf{x})) \mid \mathrm{Bern}(\sigma(\hat{f} \circ \hat{\mathbf{h}}(\mathbf{x}))]].$$

simply using the fact the data is generated from the model $y \sim \mathbb{P}_{y|\mathbf{x}}(\cdot | f \circ \mathbf{h}(\mathbf{x}))$.

To bound the task diversity we show a convexity-based lemma for general GLM/nonlinear models,

**Lemma 2.** *Consider the generalized linear model for which the $\mathbb{P}_{y|\mathbf{x}}(\cdot)$ distribution is,*

$$\mathbb{P}_{y|\mathbf{x}}(y_i | \boldsymbol{\alpha}^\top \mathbf{h}(\mathbf{x}_i)) = b(y_i) \exp\left(\frac{y_i \boldsymbol{\alpha}^\top \mathbf{h}(\mathbf{x}_i) - \Phi(\boldsymbol{\alpha}^\top \mathbf{h}(\mathbf{x}_i))}{s(\sigma)}\right).$$

*Then if $\sup_{c \in S} \Phi''(c) = L(\mathbf{x}_i)$ and $\inf_{c \in S} \Phi''(c) = \mu(\mathbf{x}_i)$ where $c \in S = [\hat{\mathbf{h}}(\mathbf{x}_i)^\top \hat{\boldsymbol{\alpha}}, \mathbf{h}(\mathbf{x}_i)^\top \boldsymbol{\alpha}]$,*

$$\frac{\mu(\mathbf{x}_i)}{2s(\sigma)}(\hat{\mathbf{h}}(\mathbf{x}_i)^\top \hat{\boldsymbol{\alpha}} - \mathbf{h}(\mathbf{x}_i)^\top \boldsymbol{\alpha})^2 \leq KL[\mathbb{P}_{y|\mathbf{x}}(\cdot | \boldsymbol{\alpha}^\top \mathbf{h}(\mathbf{x}_i)), \mathbb{P}_{y|\mathbf{x}}(\cdot | \hat{\boldsymbol{\alpha}}^\top \mathbf{h}^\star(\mathbf{x}_i))] \leq \frac{L(\mathbf{x}_i)}{2s(\sigma)}(\hat{\mathbf{h}}(\mathbf{x}_i)^\top \hat{\boldsymbol{\alpha}} - \mathbf{h}(\mathbf{x}_i)^\top \boldsymbol{\alpha})^2$$

*where the KL is taken with respect to a fixed design point $\mathbf{x}_i$, and fixed feature functions $\mathbf{h}$, and $\hat{\mathbf{h}}$.*

*Proof.*

$$\mathrm{KL}[\mathbb{P}_{y|\mathbf{x}}(\cdot | \boldsymbol{\alpha}^\top \mathbf{h}(\mathbf{x}_i)), \mathbb{P}_{y|\mathbf{x}}(\cdot | \hat{\boldsymbol{\alpha}}^\top \hat{\mathbf{h}}(\mathbf{x}_i))] =$$

$$\int dy_i \, \mathbb{P}_{y|\mathbf{x}}(y_i | \boldsymbol{\alpha}^\top \mathbf{h}(\mathbf{x}_i))\left(\frac{y_i(\mathbf{h}(\mathbf{x}_i)^\top \boldsymbol{\alpha} - \hat{\mathbf{h}}(\mathbf{x}_i)^\top \hat{\boldsymbol{\alpha}})}{s(\sigma)} + \frac{-\Phi(\mathbf{h}(\mathbf{x}_i)^\top \boldsymbol{\alpha}) + \Phi(\hat{\mathbf{h}}(\mathbf{x}_i)^\top \hat{\boldsymbol{\alpha}}))}{s(\sigma)}\right) =$$

$$\frac{1}{s(\sigma)}\left[\Phi'(\mathbf{h}(\mathbf{x}_i)^\top \boldsymbol{\alpha})(\mathbf{h}(\mathbf{x}_i)^\top \boldsymbol{\alpha} - \hat{\mathbf{h}}(\mathbf{x}_i)^\top \hat{\boldsymbol{\alpha}}) - \Phi(\mathbf{h}(\mathbf{x}_i)^\top \boldsymbol{\alpha}) + \Phi(\hat{\mathbf{h}}(\mathbf{x}_i)^\top \hat{\boldsymbol{\alpha}})\right]$$

since we have that $\frac{\Phi(\mathbf{h}(\mathbf{x}_i)^\top \boldsymbol{\alpha})}{s(\sigma)} = \log \int dy_i \, b(y_i) \exp\left(\frac{y_i \mathbf{h}(\mathbf{x}_i)^\top \boldsymbol{\alpha}}{s(\sigma)}\right) \implies \frac{\Phi'(\mathbf{h}(\mathbf{x}_i)^\top \boldsymbol{\alpha})}{s(\sigma)} = \frac{\int dy_i \, \mathbb{P}_{y|\mathbf{x}}(y_i|\boldsymbol{\alpha}^\top \mathbf{h}(\mathbf{x}_i))y_i}{s(\sigma)}$ as it is the log-normalizer. Using Taylor's theorem we have that

$$\Phi(\hat{\mathbf{h}}(\mathbf{x}_i)^\top \hat{\boldsymbol{\alpha}}) = \Phi\left(\mathbf{h}(\mathbf{x}_i)^\top \boldsymbol{\alpha}\right) + \Phi'(\mathbf{h}(\mathbf{x}_i)^\top \boldsymbol{\alpha})(\hat{\mathbf{h}}(\mathbf{x}_i)^\top \hat{\boldsymbol{\alpha}} - \mathbf{h}(\mathbf{x}_i)^\top \boldsymbol{\alpha}) + \frac{\Phi''(c)}{2}(\hat{\mathbf{h}}(\mathbf{x}_i)^\top \hat{\boldsymbol{\alpha}} - \mathbf{h}(\mathbf{x}_i)^\top \boldsymbol{\alpha})^2$$

for some intermediate $c \in [\hat{\mathbf{h}}(\mathbf{x}_i)^\top \hat{\boldsymbol{\alpha}}, \mathbf{h}(\mathbf{x}_i)^\top \boldsymbol{\alpha}]$. Combining the previous displays we obtain that:

$$\mathrm{KL}[\mathbb{P}_{y|\mathbf{x}}(\cdot | \boldsymbol{\alpha}^\top \mathbf{h}(\mathbf{x}_i)), \mathbb{P}_{y|\mathbf{x}}(\cdot | \hat{\boldsymbol{\alpha}}^\top \mathbf{h}(\mathbf{x}_i))] = \frac{1}{2s(\sigma)}\left[\Phi''(c)(\hat{\mathbf{h}}(\mathbf{x}_i)^\top \hat{\boldsymbol{\alpha}} - \mathbf{h}(\mathbf{x}_i)^\top \boldsymbol{\alpha})^2\right]$$

Now using the assumptions on the second derivative $\Phi''$ gives,

$$\frac{\mu}{2s(\sigma)}(\hat{\mathbf{h}}(\mathbf{x}_i)^\top \hat{\boldsymbol{\alpha}} - \mathbf{h}(\mathbf{x}_i)^\top \boldsymbol{\alpha})^2 \le \frac{1}{2s(\sigma)}\left[\Phi''(c)(\hat{\mathbf{h}}(\mathbf{x}_i)^\top \hat{\boldsymbol{\alpha}} - \mathbf{h}(\mathbf{x}_i)^\top \boldsymbol{\alpha})^2\right] \le \frac{L}{2s(\sigma)}(\hat{\mathbf{h}}(\mathbf{x}_i)^\top \hat{\boldsymbol{\alpha}} - \mathbf{h}(\mathbf{x}_i)^\top \boldsymbol{\alpha})^2$$

$\square$

We now instantiate the aforementioned lemma in the setting of logistic regression.

**Lemma 3.** *Consider the $\mathbb{P}_{y|\mathbf{x}}(\cdot)$ logistic generative model defined in Section 4.1 for a general feature map $\mathbf{h}(\mathbf{x})$. Then for this conditional generative model in the setting of Lemma 2, where $\Phi(\eta) = \log(1 + \exp(\eta))$, $s(\sigma) = 1$, $b(y_i) = 1$,*

$$\sup_{c(\mathbf{x}) \in S(\mathbf{x})} \Phi''(c(\mathbf{x})) \le \frac{1}{4}$$

*and*

$$\inf_{c(\mathbf{x}) \in S(\mathbf{x})} \Phi''(c(\mathbf{x})) \ge \frac{1}{4} \exp(-\max(|\hat{\mathbf{h}}(\mathbf{x}_i)^\top \hat{\boldsymbol{\alpha}}|, |\mathbf{h}(\mathbf{x}_i)^\top \boldsymbol{\alpha}|)).$$

*for fixed $\mathbf{x}$.*

*Proof.* A short computation shows $\Phi''(t) = \frac{e^t}{(e^t+1)^2}$. Note that the maxima of $\Phi''(t)$ over all $\mathbb{R}$ occurs at $t = 0$. Hence we have that, $\mathbb{E}_\mathbf{x}[\sup_{c \in S} \Phi''(c)] \le \frac{1}{4}$ using a uniform upper bound. The lower bound follows by noting that

$$\inf_{c \in S} \Phi''(t) = \min(\Phi''(|\hat{\mathbf{h}}(\mathbf{x}_i)^\top \hat{\boldsymbol{\alpha}}|), \Phi''(|\mathbf{h}(\mathbf{x}_i)^\top \boldsymbol{\alpha})|)).$$

For the lower bound note that for $t > 0$ that $e^{2t} \ge e^t \ge 1$ implies that $\frac{e^t}{(1+e^t)^2} \ge \frac{1}{4}e^{-t}$. Since $\Phi''(t) = \Phi''(-t)$ it follows that $\Phi''(t) \ge \frac{1}{4}e^{-|t|}$ for all $t \in \mathbb{R}$. $\square$

Finally we include a simple auxiliary lemma to help upper bound the averages in our data-dependent bounds which relies on a simple tail bound for covariance matrices drawn from sub-gaussian ensembles (Vershynin [2010, Theorem 4.7.3, Exercise 4.7.1] or Wainwright [2019, Theorem 6.5]). Further recall that in Definition 4 our covariate distribution is $O(1)$-sub-gaussian.

**Lemma 4.** *Let the common covariate distribution $\mathbb{P}_\mathbf{x}(\cdot)$ satisfy Definition 4. Then if $nt \gtrsim d$,*

$$\mathbb{E}[\|\boldsymbol{\Sigma}_\mathbf{X}\|] \le O(1),$$

*if $n \gtrsim d + \log t$,*

$$\mathbb{E}[\max_{j \in [t]} \|\boldsymbol{\Sigma}_{\mathbf{X}_j}\|] \le O(1),$$

*and if $m \gtrsim r$,*

$$\max_{\mathbf{B} \in \mathcal{H}} \mathbb{E}[\|\mathbf{B}^\top \boldsymbol{\Sigma}_{\mathbf{X}_0} \mathbf{B}\|] \le O(1),$$

*where $\mathcal{H}$ is the set of $d \times r$ orthonormal matrices.*

*Proof.* All of these statement essentially following by integrating a tail bound and applying the triangle inequality. For the first statement since $\mathbb{E}[\|\boldsymbol{\Sigma}_{\mathbf{X}}\|] = \mathbb{E}[\|\boldsymbol{\Sigma}_{\mathbf{X}} - \boldsymbol{\Sigma}\|] + \|\boldsymbol{\Sigma}\| \leq O(1)$ under the conditions $nt \gtrsim d$ of the result directly by Vershynin [2010, Theorem 4.7.3].

For the second by Wainwright [2019, Theorem 6.5], $\mathbb{E}[\exp(\lambda\|\boldsymbol{\Sigma} - \boldsymbol{\Sigma}\|)] \leq \exp(c_0(\lambda^2/N) + 4d)]$ for all $|\lambda| \leq \frac{N}{c_2}$, for a sample covariance averaged over $N$ datapoints. So using a union bound alongside a tail integration since the data is i.i.d. across tasks we have that,

$$\mathbb{E}[\max_{j \in [t]} \|\boldsymbol{\Sigma}_{\mathbf{X}_j} - \boldsymbol{\Sigma}\|] \leq \int_0^\infty \min(1, t\mathbb{P}[\|\boldsymbol{\Sigma}_{\mathbf{X}_1} - \boldsymbol{\Sigma}\| > \delta])d\delta \leq \int_0^\infty \min(1, \exp(c_0(\lambda^2/n) + 4d + \log t - \lambda\delta)] \leq$$

$$\int_0^\infty \min(1, \exp(4d + \log t) \cdot \exp(-c_1 \cdot n \min(\delta^2, \delta))d\delta \leq O\left(\sqrt{\frac{d + \log t}{n}} + \frac{d + \log t}{n}\right) \lesssim O(1)$$

via a Chernoff argument. The final inequality follows by bounding the tail integral and using the precondition $n \gtrsim d + \log t$. Centering the expectation and using the triangle inequality gives the conclusion.

For the last statement the crucial observation that allows the condition $m \gtrsim r$, is that $\mathbf{B}^\top \mathbf{x}_{0i}$, for all $i \in [m]$, is by definition an $r$-dimensional $O(1)$-sub-Gaussian random vector since $\mathbf{B}$ is an orthonormal projection matrix. Thus an identical argument to the first statement gives the result. $\square$

## C.2 Deep Neural Network Regression

We first begin by assembling the results necessary to bound the Gaussian complexity of our deep neural network example. To begin we introduce a representative result from the which bounds the empirical Rademacher complexity of a deep neural network.

**Theorem 8** (Theorem 2 adapted from Golowich et al. [2017])**.** *Let $\sigma$ be a 1-Lipschitz activation function with $\sigma(0) = 0$, applied element-wise. Let $\mathcal{N}$ be the class of real-valued networks of depth $K$ over the domain $\mathcal{X}$ with bounded data $\|\mathbf{x}_{ji}\| \leq D$, where $\|\mathbf{W}_k\|_{1,\infty} \leq M(k)$ for all $k \in [K]$. Then,*

$$\mathfrak{R}_n(\mathcal{N}; \mathbf{X}) \leq \left(\frac{2}{n}\Pi_{k=1}^K M(k)\right)\sqrt{(K + 1 + \log d) \cdot \max_{j \in [d]}\sum_{i=1}^n \mathbf{x}_{i,j}^2} \leq \frac{2D\sqrt{K + 1 + \log d} \cdot \Pi_{k=1}^K M(k)}{\sqrt{n}}.$$

*where $\mathbf{x}_{i,j}$ denotes the $j$-th coordinate of the vector $\mathbf{x}_i$ and $\mathbf{X}$ is an $n \times d$ design matrix (with $n$ datapoints).*

With this result in hand we proceed to bound the Gaussian complexities for our deep neural network and prove Theorem 5. Note that we make use of the result $\hat{\mathfrak{R}}_{\mathbf{X}}(\mathcal{N}) \leq \sqrt{\frac{\pi}{2}} \cdot \hat{\mathfrak{G}}_{\mathbf{X}}(\mathcal{N})$ and that $\hat{\mathfrak{G}}_{\mathbf{X}}(\mathcal{N}) \leq 2\sqrt{\log N} \cdot \hat{\mathfrak{R}}_{\mathbf{X}}(\mathcal{N})$ for any function class $\mathcal{N}$ when $\mathbf{X}$ has $N$ datapoints [Ledoux and Talagrand, 2013, p. 97].

*Proof of Theorem 5.* First note due to the task normalization conditions we can choose $c_1, c_2$ sufficiently large so that the realizability assumption in Assumption 2 is satisfied–in particular, we can assume that $c_2$ is chosen large enough to contain all the parameters $\boldsymbol{\alpha}_0^\star$ and $c_1$ large enough so that $c_1(1 \vee M(K)^2)$ is larger then the norms of the parameters $\boldsymbol{\alpha}_j^\star$ for $j \in [t]$.

Next recall that under the conditions of the result we can use Lemma 6 to verify the task diversity condition is satisfied with parameters $(\tilde{\nu}, 0)$ with $\tilde{\nu} = \sigma_r(\mathbf{A}^\top \mathbf{A}/t) > 0$. In particular under the conditions of the theorem we can verify the well-conditioning of the feature representation with $c = \Omega(1)$ which follows by definition of the set $\mathcal{H}$ and we can see that $\|\mathbb{E}_{\mathbf{x}}[\hat{\mathbf{h}}(\mathbf{x})\mathbf{h}^\star(\mathbf{x})^\top]\|_2 \leq \mathbb{E}_{\mathbf{x}}[\|\hat{\mathbf{h}}(\mathbf{x})\|\|\mathbf{h}^\star(\mathbf{x})\|] \leq O(M(K)^2)$ using the norm bound from Lemma 5. Hence under this setting we can choose $c_1$ sufficiently large so that $c_1 M(K)^2 \gtrsim O(M(K)^2)$. The condition $M(K) \gtrsim 1$ in the theorem statement is simply used to clean up the final bound.

In order to instantiate Theorem 3 we begin by bounding each of the complexity terms in the expression. First,

- For the feature learning complexity in the training phase we leverage Theorem 8 from Golowich et al. [2017] (which holds for scalar-valued outputs) in a modular fashion. For convenience let

nn $= \frac{2D\sqrt{K+1+\log d}\cdot\Pi_{k=1}^{K}M(k)}{\sqrt{nt}}$. To bound this term we simply pull the summation over the rank $r$ outside the complexity and apply Theorem 8, so

$$\hat{\mathfrak{G}}_{\mathbf{X}}(\mathcal{H}) = \frac{1}{nt}\mathbb{E}[\sup_{\mathcal{W}_K}\sum_{l=1}^{r}\sum_{j=1}^{t}\sum_{i=1}^{n}g_{kji}\mathbf{h}_k(\mathbf{x}_{ji})] \le \sum_{k=1}^{r}\hat{\mathfrak{G}}_{\mathbf{X}}(\mathbf{h}_k(\mathbf{x}_{ji})) \le \log(nt)\cdot\sum_{k=1}^{r}\hat{\mathfrak{R}}_{\mathbf{X}}(h_k(\mathbf{x}_{ji})) \le$$

$\log(nt)\cdot r\cdot$ nn

since under the weight norm constraints (i.e. the max $\ell_1$ row norms are bounded) each component of the feature can be identically bounded. This immediately implies the population Gaussian complexity bound as the expectation over $\mathbf{X}$ is trivial. Further by definition the class $\mathcal{F}$ as linear maps with parameters $\|\boldsymbol{\alpha}\|_2 \le M(K)^2$ we obtain that $L(\mathcal{F}) = O(M(K)^2)$.

- For the complexity of learning $\mathcal{F}$ we use the fact that for in the training phase we obtain,

$$\hat{\mathfrak{G}}_{\mathbf{X}_j}(\mathcal{F}) = \frac{1}{n}\mathbb{E}_{\mathbf{g}}[\sup_{\boldsymbol{\alpha}\in\mathcal{F}}\sum_{i=1}^{n}g_{ji}\boldsymbol{\alpha}^\top\mathbf{h}(\mathbf{x}_{ji})] = O\left(\frac{M(K)^2}{n}\mathbb{E}_{\mathbf{g}}[\|\sum_{i=1}^{n}g_{ji}\mathbf{h}(\mathbf{x}_{ji})\|]\right)$$

$$\le O\left(\frac{M(K)^2}{n}\sqrt{\sum_{i=1}^{n}\|\mathbf{h}(\mathbf{x}_{ji})\|^2}\right) \le O\left(\frac{M(K)^2}{\sqrt{n}}\max_i\|\mathbf{h}(\mathbf{x}_{ji})\|\right).$$

Now by appealing to the norm bounds on the feature map from Lemma 5 we have that $\max_{\mathbf{h}\in\mathcal{H}}\max_{\mathbf{X}_j}\max_i\|\mathbf{h}(\mathbf{x}_{ji})\| \lesssim M(K)$. Hence in conclusion we obtain the bound,

$$\bar{\mathfrak{G}}_n(\mathcal{F}) \le O\left(\frac{M(K)^3}{\sqrt{n}}\right)$$

since the expectation is once again trivial.

- A nearly identical argument shows the complexity of learning $\mathcal{F}$ in the testing phase is,

$$\hat{\mathfrak{G}}_{\mathbf{X}_0}(\mathcal{F}) = \frac{1}{m}\mathbb{E}_{\mathbf{g}}\left[\sup_{\boldsymbol{\alpha}:\|\boldsymbol{\alpha}\|\le c_1}\sum_{i=1}^{m}g_i\boldsymbol{\alpha}^\top\mathbf{h}(\mathbf{x}_{(0)i})\right] \le \frac{c_1 M(K)^3}{\sqrt{m}}$$

from which the conclusion follows.

Finally we verify that Assumption 1 holds so as to use Theorem 3 to instantiate the end-to-end guarantee. The boundedness parameter is,

$$D_{\mathcal{X}} \le M(K)^3$$

by Lemma 5 since it must be instantiated with $\boldsymbol{\alpha}\in\mathcal{F}$. For the $\ell_2$ loss bounds, $\ell(\eta; y) = (y - \eta)^2$. Since $\nabla_\eta\ell(\eta; y) = 2(y - \eta) \le O(N + |\eta|) = O(M(K)^3)$ where $|\eta| \le |\boldsymbol{\alpha}^\top\mathbf{h}(\mathbf{x})| \le O(M(K)^3)$ for $\boldsymbol{\alpha}\in\mathcal{F}, \mathbf{h}\in\mathcal{H}$ and $N = O(1)$ by Lemma 5. So it follows the loss is Lipschitz with $L = O(M(K)^3)$. Moreover by an analogous argument, $|\ell(\eta; y)| \le O(M(K)^6)$ so it follows the loss is uniformly bounded with parameter $B = O(M(K)^6)$.

Assembling the previous bounds shows the transfer learning risk is bounded by.

$$\lesssim \frac{L}{\tilde{\nu}}\cdot\left(\log(nt)\cdot\left[\log(nt)\cdot r\cdot M(K)^2\cdot \text{nn} + \frac{M(K)^3}{\sqrt{n}}\right]\right) + \frac{LM(K)^3}{\sqrt{m}}$$

$$+ \left(\frac{1}{\tilde{\nu}}\cdot\max\left(L\cdot\frac{M(K)^3}{(nt)^2}, B\sqrt{\frac{\log(1/\delta)}{nt}}\right) + B\sqrt{\frac{\log(1/\delta)}{m}}\right).$$

where nn $= \frac{2D\sqrt{K+1+\log d}\cdot\Pi_{k=1}^{K}M(k)}{\sqrt{nt}}$. Under the conditions of the result, the risk simplifies as in the theorem statement. $\qquad\square$

We now state a simple result which allows us to bound the suprema of the empirical $\ell_2$ norm (i.e. the $D_{\bar{\mathbf{X}}}$ parameter in Theorem 1) and activation outputs for various neural networks.

**Lemma 5.** *Let $\hat{\mathbf{h}}(\mathbf{x})$ be a vector-valued neural network of depth $K$ taking the form in (7) with each $f_j \equiv \boldsymbol{\alpha}_j$ satisfying $\|\boldsymbol{\alpha}_j\| \leq A$ with bounded data $\|\mathbf{x}\| \leq D$. Then the boundedness parameter in the setting of Theorem 1 satisfies,*

$$D_{\mathcal{X}} \lesssim AD \cdot \Pi_{k=1}^K \|\mathbf{W}_k\|_2$$

*If we further assume that $\sigma(z) = \frac{e^z - e^{-z}}{e^z + e^{-z}}$ which is centered and $1$-Lipschitz (i.e. the tanh activation function), then we obtain the further bounds that,*

$$\|\mathbf{h}(\mathbf{x})\| \leq \|\mathbf{W}_K\|_{\infty \to 2}$$

*and*

$$D_{\mathcal{X}} \lesssim A \cdot \|\mathbf{W}_K\|_{\infty \to 2}$$

*which holds without requiring boundedness of $\mathbf{x}$. Note $\|\mathbf{W}_K\|_{\infty \to 2}$ is the induced $\infty$ to $2$ operator norm.*

*Proof.* For the purposes of induction let $\mathbf{r}_k(\cdot)$ denote the vector-valued output of the $k$th layer for $k \in [K]$. First note that the bound

$$D_{\mathcal{X}} \lesssim \sup_{\boldsymbol{\alpha}, \mathbf{h}, \mathbf{x}} (\boldsymbol{\alpha}^\top \mathbf{h}(\mathbf{x}))^2 \leq \sup_{\mathbf{W}_k, \mathbf{x}} A^2 \|\mathbf{r}_K\|^2$$

Now, for the inductive step, $\|\mathbf{r}_K\|^2 = \|\mathbf{W}_K \sigma(\mathbf{W}_{K-1} \mathbf{r}_{K-1})\|^2 \leq \|\mathbf{W}_K\|_2^2 \|\sigma(\mathbf{W}_{K-1}\mathbf{r}_{K-1})\|^2 \leq \|\mathbf{W}_K\|_2^2 \|\mathbf{W}_{K-1}\mathbf{r}_{K-1}\|^2 \leq \|\mathbf{W}_K\|_2^2 \|\mathbf{W}_{K-1}\|_2^2 \|\mathbf{r}_{K-1}\|^2$ where the first inequality follows because $\sigma(\cdot)$ is element-wise Lipschitz and zero-centered. Recursively applying this inequality to the base case where $\mathbf{r}_0 = \mathbf{x}$ gives the conclusion after taking square roots.

If we further assume that $\sigma(z) = \frac{e^z - e^{-z}}{e^z + e^{-z}}$ which is centered and $1$-Lipschitz (i.e. the tanh activation function) then we can obtain the following result by simply bounding the last layer by noting that $\|\mathbf{r}_{K-1}\|_\infty \leq 1$. Then,

$$\|\mathbf{h}(\mathbf{x})\|^2 = \|\mathbf{r}_K\|_2^2 = \|\mathbf{W}_K \mathbf{r}_{K-1}\|_2^2 \leq \|\mathbf{W}_K\|_{\infty \to 2}^2$$

where $\|\mathbf{W}_K\|_{\infty \to 2}$ is the induced $\infty$ to $2$ operator norm $\qquad \square$

We now turn to proving a task diversity lower bound applicable to general $\ell_2$ regression with general feature maps $\mathbf{h}(\cdot)$ under the assumptions of the $\mathbb{P}_{y|\mathbf{x}}$ of the generative model specified in (8). As our result holds only requiring $f_j^\star \equiv \boldsymbol{\alpha}_j^\star$ and applies to more then neural network features we define some generic notation.

We assume the data generating model takes the form,

$$y_{ji} = (\boldsymbol{\alpha}_j^\star)^\top \mathbf{h}^\star(\mathbf{x}_{ji}) + \eta_{ji} \text{ for } j \in \{1, \ldots, t\}, i \in \{1, \ldots, n\} \tag{12}$$

for $\eta_{ji}$ with bounded second moments and independent of $\mathbf{x}_{ji}$. Here the shared feature representation $\mathbf{h}^\star(\cdot) \in \mathbb{R}^r$ is given by a generic function. In our generic framework we can identify $f_j^\star \equiv \boldsymbol{\alpha}_j^\star$ for $j \in \{1, \ldots, t\}$. As before we define the population task diversity matrix as $\mathbf{A} = (\boldsymbol{\alpha}_1^\star, \ldots, \boldsymbol{\alpha}_t^\star)^\top \in \mathbb{R}^{t \times r}$, $\mathbf{C} = \mathbf{A}^\top \mathbf{A}/t$ and $\tilde{\nu} = \sigma_r(\frac{\mathbf{A}^\top \mathbf{A}}{t})$. Given two feature representations $\hat{\mathbf{h}}(\cdot)$ and $\mathbf{h}^\star(\cdot)$, we can define their population covariance as,

$$\Lambda(\hat{\mathbf{h}}, \mathbf{h}^\star) = \begin{bmatrix} \mathbb{E}_{\mathbf{x}}[\hat{\mathbf{h}}(\mathbf{x})\hat{\mathbf{h}}(\mathbf{x})^\top] & \mathbb{E}_{\mathbf{x}}[\hat{\mathbf{h}}(\mathbf{x})\mathbf{h}^\star(\mathbf{x})^\top] \\ \mathbb{E}_{\mathbf{x}}[\mathbf{h}^\star(\mathbf{x})\hat{\mathbf{h}}(\mathbf{x})^\top] & \mathbb{E}_{\mathbf{x}}[\mathbf{h}^\star(\mathbf{x})\mathbf{h}^\star(\mathbf{x})^\top] \end{bmatrix} \equiv \begin{bmatrix} \mathbf{F}_{\hat{\mathbf{h}}\hat{\mathbf{h}}} & \mathbf{F}_{\hat{\mathbf{h}}\mathbf{h}^\star} \\ \mathbf{F}_{\mathbf{h}^\star\hat{\mathbf{h}}} & \mathbf{F}_{\mathbf{h}^\star\mathbf{h}^\star} \end{bmatrix} \succeq 0$$

and the generalized Schur complement of the representation of $\mathbf{h}^\star$ with respect to $\hat{\mathbf{h}}$ as,

$$\Lambda_{Sc}(\hat{\mathbf{h}}, \mathbf{h}^\star) = \mathbf{F}_{\mathbf{h}^\star\mathbf{h}^\star} - \mathbf{F}_{\mathbf{h}^\star\hat{\mathbf{h}}}(\mathbf{F}_{\hat{\mathbf{h}}\hat{\mathbf{h}}})^\dagger \mathbf{F}_{\hat{\mathbf{h}}\mathbf{h}^\star} \succeq 0.$$

We now instantiate the definition of task diversity in this setting. We assume that the universal constants $c_2$ and $c_1$ are large-enough to contain the true parameters $\boldsymbol{\alpha}_0^\star$ and $\boldsymbol{\alpha}_j^\star$ respectively for the following.

**Lemma 6.** *Consider the $\mathbb{P}_{y|\mathbf{x}}(\cdot)$ regression model defined in (12) and the loss function $\ell(\cdot,\cdot)$ taken as the squared $\ell_2$ loss. Then for this conditional generative model with $\mathcal{F} = \{\boldsymbol{\alpha} : \boldsymbol{\alpha} \in \mathbb{R}^r\}$ and $\mathcal{F}_0 = \{\boldsymbol{\alpha} : \|\boldsymbol{\alpha}\|_2 \leq c_2\}$ the model is $(\frac{\tilde{\nu}}{c_2}, 0)$ diverse in the sense of Definition 3 and,*

$$d_{\mathcal{F},\mathcal{F}_0}(\hat{\mathbf{h}}; \mathbf{h}^\star) = c_2 \cdot \sigma_{\max}(\Lambda_{sc}(\hat{\mathbf{h}}, \mathbf{h}^\star)); \quad \bar{d}_{\mathcal{F},\mathbf{f}^\star}(\hat{\mathbf{h}}; \mathbf{h}^\star) = \operatorname{tr}(\Lambda_{sc}(\hat{\mathbf{h}}, \mathbf{h}^\star)\mathbf{C}).$$

*Moreover, if we assume the set of feature representations $\hat{\mathbf{h}} \in \mathcal{H}$ in the infima over $\hat{\mathbf{h}}$ are well-conditioned in the sense that $\sigma_r(\mathbb{E}_{\mathbf{x}}[\hat{\mathbf{h}}(\mathbf{x})\hat{\mathbf{h}}(\mathbf{x})^\top]) \geq c > 0$ and $\|\mathbb{E}_{\mathbf{x}}[\hat{\mathbf{h}}(\mathbf{x})\mathbf{h}^\star(\mathbf{x})^\top]\|_2 \leq C$, then if $\mathcal{F} = \{\boldsymbol{\alpha} : \|\boldsymbol{\alpha}\| \leq c_1\}$, $\mathcal{F}_0 = \{\boldsymbol{\alpha} : \|\boldsymbol{\alpha}\|_2 \leq c_2\}$ and $c_1 \geq \frac{C}{c}c_2$ the same conclusions hold.*

*Proof.* We first bound the worst-case representation difference and then the task-averaged representation metric. For convenience we let $\mathbf{v}(\hat{\boldsymbol{\alpha}}, \boldsymbol{\alpha}) = \begin{bmatrix} \hat{\boldsymbol{\alpha}} \\ \boldsymbol{\alpha} \end{bmatrix}$ in the following. First, note that under the regression model defined with the squared $\ell_2$ loss we have that,

$$\mathbb{E}_{\mathbf{x},y \sim f \circ \mathbf{h}(\mathbf{x})}\left\{\ell(\hat{f} \circ \hat{\mathbf{h}}(\mathbf{x}), y) - \ell(f \circ \mathbf{h}(\mathbf{x}), y)\right\} = \mathbb{E}_{\mathbf{x}}[|\hat{\boldsymbol{\alpha}}^\top \hat{\mathbf{h}}(\mathbf{x}) - \boldsymbol{\alpha}^\top \mathbf{h}(\mathbf{x})|^2]$$

- the worst-case representation difference between two distinct feature representations $\mathbf{h}$ and $\mathbf{h}'$ becomes

$$d_{\mathcal{F},\mathcal{F}_0}(\hat{\mathbf{h}}; \mathbf{h}^\star) = \sup_{\boldsymbol{\alpha}:\|\boldsymbol{\alpha}\|_2 \leq c_2} \inf_{\hat{\boldsymbol{\alpha}}} \mathbb{E}_{\mathbf{x}}|\hat{\mathbf{h}}(\mathbf{x})^\top \hat{\boldsymbol{\alpha}} - \mathbf{h}^\star(\mathbf{x})^\top \boldsymbol{\alpha}_0|^2 =$$

$$\sup_{\boldsymbol{\alpha}_0:\|\boldsymbol{\alpha}_0\|_2 \leq c_2} \inf_{\hat{\boldsymbol{\alpha}}} \{\mathbf{v}(\hat{\boldsymbol{\alpha}}, -\boldsymbol{\alpha})^\top \Lambda(\hat{\mathbf{h}}, \mathbf{h}^\star)\mathbf{v}(\hat{\boldsymbol{\alpha}}, -\boldsymbol{\alpha})\} = \sup_{\boldsymbol{\alpha}_0:\|\boldsymbol{\alpha}_0\|_2 \leq c_2} \inf_{\hat{\boldsymbol{\alpha}}} \{\mathbf{v}(\hat{\boldsymbol{\alpha}}, \boldsymbol{\alpha}_0)^\top \Lambda(\hat{\mathbf{h}}, \mathbf{h}^\star)\mathbf{v}(\hat{\boldsymbol{\alpha}}, \boldsymbol{\alpha}_0)\}.$$

Recognizing the inner infima as the partial minimization of a convex quadratic form (see for example Boyd and Vandenberghe [2004, Example 3.15, Appendix A.5.4]), we find that,

$$\inf_{\hat{\boldsymbol{\alpha}}} \{\mathbf{v}(\hat{\boldsymbol{\alpha}}, \boldsymbol{\alpha}_0)^\top \Lambda(\hat{\mathbf{h}}, \mathbf{h}^\star)\mathbf{v}(\hat{\boldsymbol{\alpha}}, \boldsymbol{\alpha}_0)\} = \boldsymbol{\alpha}_0^\top \Lambda_{sc}(\hat{\mathbf{h}}, \mathbf{h}^\star)\boldsymbol{\alpha}_0$$

Note that in order for the minimization be finite we require $\mathbf{F}_{\hat{\mathbf{h}}\hat{\mathbf{h}}} \succeq 0$ and that $\mathbf{F}_{\hat{\mathbf{h}}\mathbf{h}^\star}\boldsymbol{\alpha} \in$ range$(\mathbf{F}_{\hat{\mathbf{h}}\hat{\mathbf{h}}})$ – which are both satisfied in our since they are constructed as expectations over appropriate rank-one operators. In this case, a sufficient condition for $\hat{\boldsymbol{\alpha}}$ to be an minimizer is that $\hat{\boldsymbol{\alpha}} = -\mathbf{F}_{\hat{\mathbf{h}}\hat{\mathbf{h}}}^\dagger \mathbf{F}_{\hat{\mathbf{h}}\mathbf{h}^\star}\boldsymbol{\alpha}$. Finally the suprema over $\boldsymbol{\alpha}$ can be computed using the variational characterization of the singular values.

$$\sup_{\boldsymbol{\alpha}_0:\|\boldsymbol{\alpha}_0\|_2 \leq c_2} \boldsymbol{\alpha}_0^\top \Lambda_{sc}(\hat{\mathbf{h}}, \mathbf{h}^\star)\boldsymbol{\alpha}_0 = c_2 \cdot \sigma_{\max}(\Lambda_{sc}(\hat{\mathbf{h}}, \mathbf{h}^\star))$$

- The task-averaged representation difference can be computed by similar means

$$\bar{d}_{\mathcal{F},\mathbf{f}^\star}(\hat{\mathbf{h}}; \mathbf{h}^\star) = \frac{1}{t}\sum_{j=1}^{t} \inf_{\hat{\boldsymbol{\alpha}}} \mathbb{E}_{\mathbf{x}}|\hat{\mathbf{h}}(\mathbf{x})^\top \hat{\boldsymbol{\alpha}} - \mathbf{h}^\star(\mathbf{x})^\top \boldsymbol{\alpha}_j^\star|^2 = \frac{1}{t}\sum_{j=1}^{t} (\boldsymbol{\alpha}_j^\star)^\top \Lambda_{sc}(\hat{\mathbf{h}}, \mathbf{h}^\star)\boldsymbol{\alpha}_j^\star$$

$$= \operatorname{tr}(\Lambda_{sc}(\hat{\mathbf{h}}, \mathbf{h}^\star)\mathbf{C})$$

Note that since $\Lambda_{sc}(\hat{\mathbf{h}}, \mathbf{h}^\star) \succeq 0$, and $\mathbf{C} \succeq 0$, by a corollary of the Von-Neumann trace inequality, we have that $\operatorname{tr}(\Lambda_{sc}(\hat{\mathbf{h}}, \mathbf{h}^\star)\mathbf{C}) \geq \sum_{i=1}^{r} \sigma_i(\Lambda_{sc}(\hat{\mathbf{h}}, \mathbf{h}^\star))\sigma_{r-i+1}(\mathbf{C}) \geq \operatorname{tr}(\Lambda_{sc}(\hat{\mathbf{h}}, \mathbf{h}^\star))\sigma_r(\mathbf{C}) \geq \sigma_{\max}(\Lambda_{sc}(\hat{\mathbf{h}}, \mathbf{h}^\star))\sigma_r(\mathbf{C}).$

Combining the above two results we can immediately conclude that,

$$d_{\mathcal{F},\mathcal{F}_0}(\hat{\mathbf{h}}; \mathbf{h}^\star) = c_2\sigma_{\max}(\Lambda_{sc}(\hat{\mathbf{h}}, \mathbf{h}^\star)) \leq \frac{1}{\tilde{\nu}/c_2}\bar{d}_{\mathcal{F},\mathbf{f}^\star}(\hat{\mathbf{h}}; \mathbf{h}^\star)$$

The second conclusion uses Lagrangian duality for the infima in both optimization problems for the worst-case and task-averaged representation differences. In particular, since the $\inf_{\hat{\boldsymbol{\alpha}}} \mathbb{E}_{\mathbf{x}}|\hat{\mathbf{h}}(\mathbf{x})^\top \hat{\boldsymbol{\alpha}} - \mathbf{h}^\star(\mathbf{x})^\top \boldsymbol{\alpha}|^2$ is a strongly-convex under the well-conditioned assumption, we have its unique minimizer is given by $\hat{\boldsymbol{\alpha}} = -(\mathbf{F}_{\hat{\mathbf{h}}\hat{\mathbf{h}}})^{-1}\mathbf{F}_{\hat{\mathbf{h}}\mathbf{h}^\star}\boldsymbol{\alpha}$; hence $\|\hat{\boldsymbol{\alpha}}\| \leq$

$\frac{C}{c}\|\boldsymbol{\alpha}\|$. Thus, if we consider the convex quadratically-constrained quadratic optimization problem $\inf_{\hat{\boldsymbol{\alpha}}:\|\hat{\boldsymbol{\alpha}}\|_2 \leq c_1} \mathbb{E}_{\mathbf{x}}|\hat{\mathbf{h}}(\mathbf{x})^\top \hat{\boldsymbol{\alpha}} - \mathbf{h}^\star(\mathbf{x})^\top \boldsymbol{\alpha}|^2$ and $c_1 \geq \frac{C}{c}\|\boldsymbol{\alpha}\|$ the constraint is inactive, and the constrained optimization problem is equivalent to the unconstrained optimization problem. Hence for the choice of $\mathcal{F} = \{\boldsymbol{\alpha} : \|\boldsymbol{\alpha}\| \leq c_1\}$ the infima in both the computation of the task-averaged distance and worst-case representation difference can be taken to be unconstrained. The second conclusion follows. $\qquad\square$

## C.3   Index Models

We prove the general result which provides the end-to-end learning guarantee. Recall that we will use $\boldsymbol{\Sigma}_{\mathbf{X}}$ to refer the sample covariance over the the training phase data.

*Proof of Theorem 6.* First by definition of the sets $\mathcal{F}_0$ and $\mathcal{F}$ the realizability assumption holds true. Next recall that under the conditions of the result we can use Lemma 7 to verify the task diversity condition is satisfied with parameters $(\tilde{\nu}, \epsilon)$ with $\tilde{\nu} \geq \frac{1}{t}$. Note in fact we have the stronger guarantee $\tilde{\nu} \geq \frac{\|\mathbf{v}\|_1}{\|\mathbf{v}\|_\infty}\frac{1}{t}$ for $\mathbf{v}_j = \inf_{\hat{f}\in\mathcal{F}} \mathbb{E}_{\mathbf{x},\eta}[L(f_j^\star(\mathbf{b}^\star(\mathbf{x})) - \hat{f}(\hat{\mathbf{b}}(\mathbf{x})) + \eta)]$. So if $\mathbf{v}$ is well spread-out given a particular learned representation $\hat{\mathbf{b}}$, the quantity $\tilde{\nu}$ could be much larger in practice and the transfer more sample-efficient then the worst-case bound suggests.

In order to instantiate Theorem 3 we begin by bounding each of the complexity terms in the expression. First,

- For the feature learning complexity in the training phase standard manipulations give,

$$\hat{\mathfrak{G}}_{\mathbf{X}}(\mathcal{H}) \leq \frac{1}{nt}\mathbb{E}\left[\sup_{\mathbf{b}:\|\mathbf{b}\|_2 \leq W} \sum_{j=1}^{t}\sum_{i=1}^{n} g_{ji}\mathbf{b}^\top \mathbf{x}_{ji}\right] \leq \frac{W}{nt}\sqrt{\mathbb{E}[\|\sum_{j=1}^{t}\sum_{i=1}^{n} g_{ji}\mathbf{x}_{ji}\|_2^2]}$$

$$\leq \frac{W}{nt}\sqrt{\sum_{j=1}^{t}\sum_{i=1}^{n}\|\mathbf{x}_{ji}\|^2} = \sqrt{\frac{W^2\mathrm{tr}(\boldsymbol{\Sigma}_{\mathbf{X}})}{nt}}$$

  Further by definition the class $\mathcal{F}$ is 1-Lipschitz so $L(\mathcal{F}) = 1$. Taking expectations and using concavity of the $\sqrt{\cdot}$ yields the first term.

- For the complexity of learning $\mathcal{F}$ in the training phase we appeal to the Dudley entropy integral (see [Wainwright, 2019, Theorem 5.22]) and the metric entropy estimate from Kakade et al. [2011, Lemma 6(i)]. First note that $N(\mathcal{F}, d_{2,\mathbf{b}\mathbf{X}_j}, \epsilon) \leq N(\mathcal{F}, \|\cdot\|_\infty, \epsilon)$. By Kakade et al. [2011, Lemma 6(i)] $N(\mathcal{F}, \|\cdot\|_\infty, \epsilon) \leq \frac{1}{\epsilon}2^{2DW/\epsilon}$. So for all $0 \leq \epsilon \leq 1$,

$$\hat{\mathfrak{G}}_{\mathbf{Y}}(\mathcal{F}) \lesssim 4\epsilon + \frac{32}{\sqrt{n}}\int_{\epsilon/4}^{1}\sqrt{\log N(\mathcal{F}, \|\cdot\|_\infty, u)}du \lesssim \epsilon + \frac{1}{\sqrt{n}}\int_{\epsilon/4}^{1}\sqrt{\log\left(\frac{1}{u}\right) + \frac{2WD}{u}}du$$

$$\lesssim \epsilon + \frac{\sqrt{WD}}{\sqrt{n}}\int_{\epsilon/4}^{1}\frac{1}{u^{1/2}}du \lesssim \epsilon + \sqrt{\frac{WD}{n}}\cdot(2-\epsilon) \leq O\left(\sqrt{\frac{WD}{n}}\right)$$

  using the inequality that $\log(\frac{1}{u}) \leq 2\frac{WD}{u}$ and taking $\epsilon = 0$. This expression has no dependence on the input data or feature map so it immediately follows that,

$$\bar{\mathfrak{G}}_n(\mathcal{F}) \leq O\left(\sqrt{\frac{WD}{n}}\right)$$

- A nearly identical argument shows the complexity of learning $\mathcal{F}$ in the testing phase is,

$$\bar{\mathfrak{G}}_m(\mathcal{F}) \leq O\left(\sqrt{\frac{WD}{m}}\right)$$

Finally we verify that Assumption 1 holds so as to use Theorem 3 to instantiate the end-to-end guarantee. First the boundedness parameter becomes,

$$D_{\mathcal{X}} = 1$$

by definition since all the functions $f$ are bounded between $[0, 1]$. Again, simply by definition the $\ell_1$ norm is 1-Lipschitz in its first coordinate uniformly over the choice of its second coordinate. Moreover as the noise $\eta_{ij} = O(1)$, the loss is uniformly bounded by $O(1)$ so $B = O(1)$. Assembling the previous bounds and simplifying shows the transfer learning risk is bounded by,

$$\lesssim \frac{\log(nt)}{\tilde{\nu}} \cdot \left( \sqrt{\frac{W^2 \mathbb{E}_{\mathbf{X}}[\text{tr}(\boldsymbol{\Sigma}_{\mathbf{x}})]}{nt}} + \sqrt{\frac{WD}{n}} \right) + \sqrt{\frac{WD}{m}} + \frac{1}{(nt)^2} + \frac{1}{\tilde{\nu}}\sqrt{\frac{\log(1/\delta)}{nt}} + \sqrt{\frac{\log(1/\delta)}{m}} + \epsilon$$

If we hide all logarithmic factors, we can verify the noise-terms are all higher-order to get the simplified statement in the lemma. □

We now introduce a generic bound to control the task diversity in a general setting. In the following recall $\mathcal{F}_t = \text{conv}\{f_1, \ldots, f_t\}$ where $f_j \in \mathcal{F}$ for $j \in [t]$ where $\mathcal{F}$ is a convex function class. Further, we define the $\epsilon$-enlargement of $\mathcal{F}_t$ with respect to the sup-norm by $\mathcal{F}_{t,\epsilon} = \{f : \exists \tilde{f} \in \mathcal{F}_t \text{ such that } \sup_z |f(z) - f'(z)| \leq \epsilon\}$. We also assume the loss function $\ell(a, b) = L(a - b)$ for a positive, increasing function obeying a triangle inequality (i.e. a norm) for the following.

Our next results is generic and holds for all regression models of the form,

$$y = f(\mathbf{h}(\mathbf{x})) + \eta. \tag{13}$$

which encompasses the class of multi-index models.

**Lemma 7.** *In the aforementioned setting and consider the $\mathbb{P}_{y|\mathbf{x}}(\cdot)$ regression model defined in* (13). *If $\mathcal{F} = \mathcal{F}$ for a base, convex function class $\mathcal{F}$, and $\mathcal{F}_0 = \mathcal{F}_{t,\tilde{\epsilon}}$ the model is $(\tilde{\nu}, \tilde{\epsilon})$ diverse in the sense of Definition 3 for $\tilde{\nu} \geq \frac{1}{t}$.*

*Proof.* This result follows quickly from several properties of convex functions. First the mapping

$$(f, \hat{f}) \rightarrow \mathbb{E}_{\mathbf{x} \sim \mathbb{P}_{\mathbf{x}}(\cdot), y \sim \mathbb{P}_{y|\mathbf{x}}(f \circ \mathbf{h}(\mathbf{x}))} \left[ \ell(\hat{f} \circ \hat{\mathbf{h}}(\mathbf{x}), y) - \ell(f \circ \mathbf{h}(\mathbf{x}), y) \right] =$$

$$\mathbb{E}_{\mathbf{x},n}[L(f(\mathbf{h}(\mathbf{x})) - \hat{f}(\hat{\mathbf{h}}(\mathbf{x})) + \eta)] - \mathbb{E}_n[L(n)]$$

is a jointly convex function of $(f, \hat{f})$. This follows since as an affine precomposition of a convex function, $L(f(\mathbf{h}(\mathbf{x})) - \hat{f}(\hat{\mathbf{h}}(\mathbf{x})) + \eta)$ is convex for all $\mathbf{x}, \eta$ and the expectation operator preserves convexity. Now by definition of $\mathcal{F}_{t,\tilde{\epsilon}}$, for all $f \in \mathcal{F}_{t,\tilde{\epsilon}}$ there exists $\tilde{f} \in \mathcal{F}_t$ such $\sup_z |f(z) - \tilde{f}(z)| \leq \tilde{\epsilon}$. Thus for all $f$ we have that,

$$\mathbb{E}_{\mathbf{x},\eta}[L(f(\mathbf{h}(\mathbf{x})) - \hat{f}(\hat{\mathbf{h}}(\mathbf{x})) + \eta)] - \mathbb{E}_\eta[L(\eta)] \leq \mathbb{E}_{\mathbf{x},\eta}[L(\tilde{f}(\mathbf{h}(\mathbf{x})) - \hat{f}(\hat{\mathbf{h}}(\mathbf{x})) + \eta)] - \mathbb{E}_\eta[L(\eta)] + \tilde{\epsilon}$$

for $\tilde{f} \in \mathcal{F}_t$. Then since partial minimization of $\hat{f}$ over the convex set $\mathcal{F}$ of this jointly convex upper bound preserves convexity, we have that the mapping from $f$ to $\inf_{\hat{f} \in \mathcal{F}} \mathbb{E}_{\mathbf{x},n}[L(f(\mathbf{h}(\mathbf{x})) - \hat{f}(\hat{\mathbf{h}}(\mathbf{x})) + \epsilon)] - \mathbb{E}_\eta[L(\eta)]$ is a convex function of $f$. Thus,

$$\inf_{\hat{f} \in \mathcal{F}} \mathbb{E}_{\mathbf{x},\eta}[L(f(\mathbf{h}(\mathbf{x})) - \hat{f}(\hat{\mathbf{h}}(\mathbf{x})) + \eta)] - \mathbb{E}_\eta[L(\eta)] \leq \inf_{\hat{f} \in \mathcal{F}} \mathbb{E}_{\mathbf{x},\eta}[L(\tilde{f}(\mathbf{h}(\mathbf{x})) - \hat{f}(\hat{\mathbf{h}}(\mathbf{x})) + \eta)] - \mathbb{E}_\eta[L(\eta)] + \tilde{\epsilon}$$

Now taking the suprema over $f \in \mathcal{F}_{t,\tilde{\epsilon}}$ gives,

$$\sup_{f \in \mathcal{F}_{t,\tilde{\epsilon}}} \inf_{\hat{f} \in \mathcal{F}} \mathbb{E}_{\mathbf{x},\eta}[L(f(\mathbf{h}(\mathbf{x})) - \hat{f}(\hat{\mathbf{h}}(\mathbf{x})) + \eta)] - \mathbb{E}_\eta[L(\eta)] \leq$$

$$\sup_{\tilde{f} \in \mathcal{F}_t} \inf_{\hat{f} \in \mathcal{F}} \mathbb{E}_{\mathbf{x},\eta}[L(\tilde{f}(\mathbf{h}(\mathbf{x})) - \hat{f}(\hat{\mathbf{h}}(\mathbf{x})) + \eta)] - \mathbb{E}_\eta[L(\eta)] + \tilde{\epsilon}$$

Finally, since the suprema of a a convex function over a convex hull generated by a finite set of points can be taken to occur at the generating set,

$$\sup_{\tilde{f} \in \mathcal{F}_t} \inf_{\hat{f} \in \mathcal{F}} \mathbb{E}_{\mathbf{x},\eta}[L(\tilde{f}(\mathbf{h}(\mathbf{x})) - \hat{f}(\hat{\mathbf{h}}(\mathbf{x})) + \eta)] = \max_{j \in [t]} \inf_{\hat{f} \in \mathcal{F}} \mathbb{E}_{\mathbf{x},\eta}[L(f_j(\mathbf{h}(\mathbf{x})) - \hat{f}(\hat{\mathbf{h}}(\mathbf{x})) + \eta)]$$

Finally, for a $t$-dimensional vector $\mathbf{v}$, $\|\mathbf{v}\|_\infty \leq \|\mathbf{v}\|_1$. Instantiating this with the vector with components $\mathbf{v}_j = \inf_{\hat{f} \in \mathcal{F}} \mathbb{E}_{\mathbf{x},\eta}[L(f_j^\star(\mathbf{h}^\star(\mathbf{x})) - \hat{f}(\hat{\mathbf{h}}(\mathbf{x})) + \eta)]$ and combining with the above shows that[8],

$$d_{\mathcal{F},\mathcal{F}_0}(\hat{\mathbf{h}}; \mathbf{h}^\star) \leq \bar{d}_{\mathcal{F},\mathbf{f}^\star}(\hat{\mathbf{h}}; \mathbf{h}^\star) \cdot \frac{1}{\tilde{\nu}} + \epsilon$$

where $\tilde{\nu} \geq \frac{1}{t}$ (but might potentially be larger). Explicitly $\tilde{\nu} \geq \frac{1}{t}\frac{\|\mathbf{v}\|_1}{\|\mathbf{v}\|_\infty}$. In the case the vector $\mathbf{v}$ is well-spread out over its coordinates we expect the bound $\|\mathbf{v}\|_1 \geq \|\mathbf{v}\|_\infty$ to be quite loose and $\tilde{\nu}$ could be potentially much greater. $\qquad\square$

Note if $\mathbf{v}$ is well-spread out – intuitively the problem possesses a problem-dependent "uniformity" and the bound $\tilde{\nu} \geq \frac{1}{t}$ is likely pessimistic. However, formalizing this notion in a clean way for nonparametric function classes considered herein seems quite difficult.

Also note the diversity bound of Lemma 7 is valid for *generic* functions and representations in addition to applying to a wide class of regression losses. In particular, all $p$-norms such $L(a, b) = \|a - b\|_p$. Further only mild moments boundedness conditions are required on $\epsilon$ to ensure finiteness of the objective.