[Reviews · NeurIPS 2020]

Review 1

Summary and Contributions: 1. This work proves a new generalization bound for transfer learning in the realizable setting via first learning a common representation and then training a classification head for the new task. 2. Examples are given for logistic regression, deep neural network regression, and robust regression for single-index models. 3. A novel chain rule for Gaussian complexities is derived as an additional technical contribution.

Strengths: 1. The framework generalizes several previous works, and the bound works for general losses and model families. 2. Theoretical derivations are thorough and rigorous. Interpretations of theorems are clearly stated.

Weaknesses: 1. The framework assumes realizability. Maybe the author can discuss what happens with moderate model misspecification. 2. The training tasks are assumed to be homogenous in sample size and complexity. In realistic scenarios, practitioners pre-train on only one or two complex tasks (such as ImageNet) before transferring to many downstream tasks. The theory does not explain why transfer learning works when training tasks are not diverse. 3. In all three examples, the 'classifier head' hypothesis class F is linear. I wonder what task-diversity constants (definition 3) can be derived for more complex family F such as a multi-layer neural network. 4. Question: Does the analysis for neural networks only work for squared loss? How about logistic loss, or classification? 5. Question: Can more refined bounds than [1] be applied to deep neural networks? For example, how about data-dependent bounds, such as margin bounds, or bounds based on layer-wise Lipschitzness of the network on the training data [2]? 6. More experiments can be done to explore the applicability of the theory and the tightness of the bound. [1] Golowich, Noah, Alexander Rakhlin, and Ohad Shamir. "Size-independent sample complexity of neural networks." Conference On Learning Theory. 2018. [2] Wei, Colin, and Tengyu Ma. "Improved sample complexities for deep neural networks and robust classification via an all-layer margin." International Conference on Learning Representations. 2019.

Correctness: The theoretical derivations seem correct, although I did not fully read the proof in the supplementary materials.

Clarity: Yes. The notation about eigenvalue in assumption 3 is not defined.

Relation to Prior Work: Yes.

Reproducibility: Yes

Additional Feedback: Update after author feedback: I thank the authors for their response to my questions. I think it would be interesting to explain why transfer learning works when training tasks are not diverse (e.g. pre-train on one hard task can lead to good performance on many easier tasks).


Review 2

Summary and Contributions: The paper studies the problem of transfer learning to a task with few samples via learning a shared representation from many related tasks. The authors propose a two stage ERM procedure and show that, under certain assumptions, that the performance of the final predictor on the new task benefits from the representation learning step. Furthermore, among these assumptions, is a notion of task diversity that the authors propose which captures that the excess statistical risk decreases as the tasks (to learn the representation) are more and more diverse. The authors use this general result to instantiate bounds on Logistic regression, deep neural networks and multi-index models. Towards proving these results, the authors develop a new chain rule for computing Gaussian complexity of classes with a composite function structure. --------------- Post author feedback comments --------------- I thank the authors for the response. I encourage the authors to add this discussion on task diversity provided in the feedback to the paper itself. Based on this, I increase my score from 6 to 7.

Strengths: 1. The paper studies problem/method which in recent years, has been empirically very successful and is being widely used, but has little theory. 2. The results are good, in the sense that the final bounds that the authors obtain are intuitively expected, and indeed showcase improvements via the representation learning step (unlike previous work). 3. The paper is mostly written well, the related work is discused well and the proofs are described rigorously. 4. The develop a new technical tool: chain rule for Gaussian complexity, which could be applicable/useful in such composite function learning settings.

Weaknesses: To me, the theory felt abstract, in the sense that the quantities seemed conveniently defined so as the theoretical results follow. If the authors think that these are indeed natural quantities, then (in my opinion), their intuitive meaning is not sufficiently discussed. For example, the title and asbstract of the paper suggests that a key contribution is the notion of task diversity. This quantity first appears in page 6 (section 3.3, definition 3) and besides explaining the formula very generally in a few lines, there is hardly any discussion on what it means. Furthermore, in the applications considered, the diversity assumption takes specific forms (like assumption 3), which is also not (sufficiently) discussed.

Correctness: Yes, I skimmed through the proofs, which are mostly based on standard arguments, and look sound.

Clarity: Mostly well-written. Some more discussion on the proposed task diversity notion could be useful.

Relation to Prior Work: Yes.

Reproducibility: Yes

Additional Feedback: 1. Why state results in terms of Gaussian complexity rather than Rademacher complexity which is more common in learning theory literature? Especially when the proofs in the paper start with using these standard Rademacher Complexity bounds, and then these are switched to Gaussian complexity. Is there a technical reason (though I don't see that any place in the proofs which crucially uses some fact about Gaussian random variables), or it is arbitrary choice? 2. Typos: 1. In line 480 of supplementary, What is $\hat f$? (it doesn't seem to be defined anywhere before) 2. In line 626 of supplementary, "we claim the set..", shouldn't the union be $h \in C_{F_{h(x)}}^{\times t}$ 3. In line 629 of supplementary, "By construction..", shouldn't it be $(f',h')$ instead of $(f,h')$?


Review 3

Summary and Contributions: This work presents new statistical guarantees in the context of transfer learning using shared learning representations. The decomposition of the Gaussian complexity of the end-to-end transfer learning pipeline into Gaussian complexity of learning representation H and task-specific function F is based on a novel chain rule for Gaussian complexities contributed by this work. A fundamental task-diversity measure, (v, epsilon)-diverse, is also defined so that the statistical guarantees can be provided in terms of the diversity of a family of task.

Strengths: The new guarantees are provided on decoupled representation complexity and task-specific function complexity. This enables a better understanding of the dynamics due to the sample size of the pre-training tasks, the sample size of the target task, the complexity of the representation model and the complexity of the task-specific model. Indeed, we can see from Theorem 3 that the Gaussian complexity of the representation function must be greater than the Gaussian complexity of the task-specific function for the transfer learning risk to scale slower than the empirical risk of naive algorithm learning each task in isolation, along with large enough sample size of pre-training tasks. In order to provide the statistical guarantees, this work provided new chain rule for Gaussian complexities and a task diversity definition more general than previous works.

Weaknesses: The Gaussian complexity for H and F are likely providing loose bounds. I do not see this as a strong limitation, but rather as future work.

Correctness: As stated earlier the assumptions behind the claims are reasonable. I did not verify the proofs in the appendix.

Clarity: As someone of limited familiarity with theoretical works on statistical guarantees, the paper is written well enough for me to follow and understand overall.

Relation to Prior Work: The related section gives a great overview of the topic and clear comparisons with the current work.

Reproducibility: Yes

Additional Feedback: I am no expert in theoretical work and therefore could only provide limited feedback. I apologize for this.

[Author Response · NeurIPS 2020]

We thank all the reviewers for their helpful corrections and feedback. We will incorporate them into the revised version.

**R1:** We thank R1 for pointing out the missing eigenvalue notation. With regards to comments in weakness section,

1. In the fully agnostic setting (where the model is completely misspecified) we don't believe our results apply
because the identity of the underlying shared representation becomes ambiguous. But for small degrees of model
misspecification, we believe our results can be extended there at the price of additional error terms in our guarantees.

2. Our current results assume homogeneous sample size across different tasks for a clean presentation. We believe
our techniques/results can be easily extended to non-homogeneous sample sizes across different tasks (with additional
notational modifications in our results).

3. The "classifier head" of Example 3 (in Appendix A) is actually a nonlinear function. We agree that further
investigating the task diversity definition for different examples is an important direction for future research.

4. We believe the neural network example will generalize to the logistic loss (by combining with the contents in Section
4.1). We opted for our current choice of different losses/functions to exhibit the utility of our general framework.

5. We believe that other bounds on the Gaussian/Rademacher complexity can directly be applied in our framework
with little modification (such as [1, Theorem 5] for example). However, our framework does not directly accommodate
margin bounds–such as in [2]–and extending our results to include these is an interesting future direction for research.

6: We agree that investigating the experimental implications of our bounds is an important direction for future work.

**R2:** We thank R2 for pointing out the typos, and suggestions for clarification. With regards to the typos in the
"Additional Feedback" Section. For 1: $\hat{\mathbf{f}}$ refers to an ERM solution in the task variables of the training risk in Eq.
2–the pair $(\hat{\mathbf{f}}, \hat{\mathbf{h}})$ refers an entire ERM solution in Eq.2; we will define/clarify this. For 2: The union should be written
$\mathbf{h} \in C_{\mathcal{H}_X}$ (referring to the $\mathcal{H}$ covering with respect to inputs $\mathbf{X}$). For 3: Yes, both terms should be primed (and the
statement should be more clearly written as, "given this $\mathbf{h}'$, $\exists \mathbf{f}' \in C_{\mathcal{F}^{\otimes t}_{\mathbf{h}'(\mathbf{X})}}$ that is $\epsilon_2$-close to $\mathbf{f}$ with respect to inputs
$\mathbf{h}'(\mathbf{X})$. By construction, $\mathbf{h}' \in C_{\mathcal{H}_X}$ and $\mathbf{f}' \in C_{\mathcal{F}^{\otimes t}(\mathcal{H})}$."). We will correct/clarify the notation here.

With regards to Gaussian complexity vs Rademacher complexity, we chose to include only Gaussian complexities in the
main paper to simplify the presentation, since the chain rule is most naturally stated in terms of Gaussian complexities.
Since Gaussian/Rademacher complexities are equivalent up to logarithmic factors [3, p.97] we could also rewrite all our
results in terms of Rademacher complexities at the cost of only logarithmic factors.

With regards to the comments in the weakness section, we believe our notion of task diversity can be understood in a nat-
ural way, and will provide further discussion and intuition to interpret it—see the following. Our framework/arguments
(which hold for general $\mathcal{F}$, $\mathcal{H}$ and $\ell$) do require abstraction, but we also believe this abstraction is a strength that allows
our guarantees to be applied to a wide class of problems.

Definition 1 and Definition 2 seek to define two notions of distance between two representations $\mathbf{h}, \mathbf{h}'$. In our
framework, information about the representations is only observed through the composite functions $f \circ \mathbf{h}$. For any
direction/component in $\mathbf{h}$ that is not seen by a corresponding task $f$, that component of the representation $\mathbf{h}$ cannot
be distinguished from a corresponding one in a spurious $\mathbf{h}'$. When this component is needed to predict on a new task
$f_0$ which lies along that direction, transfer learning will not be possible. Therefore, Definition 1 defines a notion of
representation distance in terms of information channeled through the training tasks while Definition 2 defines it in
terms of an arbitrary new test task. Task diversity essentially encodes the ratio of these two quantities (i.e. how well
the training tasks can observe relevant parts of the representations useful for the new task). In the case where the $\mathcal{F}$
contains underlying linear task functions $\boldsymbol{\alpha}_j^\star \in \mathbb{R}^r$ (as in our examples in Section 4), our task diversity definition
reduces to ensuring these task vectors span the entire $r$-dimensional space containing the output of the representation
$\mathbf{h}(\cdot) \in \mathbb{R}^r$. This is quantitatively captured by the conditioning parameter $\tilde{\nu} = \sigma_r(\mathbf{A})$ for $\mathbf{A} = (\boldsymbol{\alpha}_1^\star, \ldots, \boldsymbol{\alpha}_t^\star)^\top \in \mathbb{R}^{t \times r}$
which represents how correlated these vectors are in $\mathbb{R}^r$. Appendix A gives a further task diversity example when $\mathcal{F}$
contains nonparametric functions. We will provide further explanation of these definitions and their relationship to each
specific example in the final version.

**R3:** We thank R3 for their comments and agree studying transfer learning in frameworks other then those using
Gaussian complexities (i.e. with more refined data-dependent bounds as R1 mentions), is an interesting future direction.

[1] Golowich, Noah, Alexander Rakhlin, and Ohad Shamir. "Size-independent sample complexity of neural networks."
Conference On Learning Theory. 2018.

[2] Wei, Colin, and Tengyu Ma. "Improved sample complexities for deep neural networks and robust classification via
an all-layer margin." International Conference on Learning Representations. 2019.

[3] Ledoux, Michel, and Talagrand, Michel. Probability in Banach Spaces.


[Meta-Review · NeurIPS 2020]

The reviewers reached a consensus that the paper can be accepted to NeurIPS. One additional comment from the meta-reviewer is that because the paper doesn't have any experimental component, it's unclear whether the message in the title "The Importance of Task Diversity" is sufficiently justified. It's true that the theory need the assumption of diverse task, but it's unclear why that's the most important one amont many of the assumptions.